# NOISE-RESILIENT QUANTUM NEURAL NETWORKS VIA ZERO-NOISE KNOWLEDGE DISTILLATION

## ABSTRACT

Quantum neural networks (QNNs) show promise for learning on noisy intermediate-scale quantum (NISQ) devices, but two-qubit gate noise remains a significant barrier to practical implementation. Zero-noise extrapolation (ZNE) reduces errors by running circuits with scaled noise levels and extrapolating to the zero-noise limit, although it needs many evaluations per input and is susceptible to time-varying noise. We propose *zero-noise knowledge distillation (ZNKD)*, a training-time technique that involves a ZNE-augmented teacher QNN supervising a compact student QNN. Variational learning is used to optimize the student's ability to duplicate the teacher's extrapolated outputs, resulting in robustness without the need for inference extrapolation. We additionally present a formal analysis that demonstrates how robustness flows from the ZNE teacher to the distilled student, with proofs regarding noise scaling, extrapolation error, and student generalization. In dynamic-noise simulations (IBM-style $T_1/T_2$, depolarizing, readout), ZNE-guided distillation lowers student MSE by 0.06-0.12 ($\approx$10-20%) across Fashion-MNIST, AG News, UCI Wine, and UrbanSound8K, keeping students within 2%–4% accuracy of the teacher and achieving 6:2-8:3 ratio of teacher to student. ZNKD, which amortizes ZNE to training, provides an efficient way to drift-resilient QNNs on NISQ hardware without per-input folding or extrapolation.

## 1 INTRODUCTION

Quantum neural networks (QNNs) have shown high potential to tackle machine learning challenges of faster computation using quantum computing, especially in noisy intermediate-scale quantum (NISQ) devices Jiao et al. (2025); Havlíček et al. (2019); Chen et al. (2024). These models use the high-dimensional expressivity of quantum circuits to describe complicated decision boundaries with fewer parameters than traditional neural networks. However, practical deployment of QNNs is severely limited by the inherent noise and decoherence present in current NISQ hardware Sun et al. (2024b); Jurcevic et al. (2021); Afane et al. (2025). This noise reduces the accuracy and dependability of QNN outputs, restricting their use in real-world applications. Several error mitigation strategies have been developed to address this; however, they frequently involve additional circuit executions, deeper circuits, or hardware capabilities that are beyond the reach of most NISQ devices due to their excessive resources Temme et al. (2017); Kandala et al. (2019); Cerezo et al. (2022).

Zero-noise extrapolation (ZNE) is a widely used error mitigation method that estimates noise-free expectation values without requiring quantum error correction Temme et al. (2017); Giurgica-Tiron et al. (2020); He et al. (2020); Pelofske et al. (2024). It increases circuit noise—typically via gate folding or repeated gate insertions—and extrapolates the resulting measurements back to the zero-noise limit. However, ZNE has several limitations. First, it requires multiple circuit executions at different effective noise levels, increasing sampling costs. Second, circuit depth scales linearly or quadratically with the noise factor $\lambda$, reducing applicability on hardware with short coherence times Giurgica-Tiron et al. (2020); Majumdar et al. (2023). In global folding, $\lambda = 1 + 2k$ (with $k$ fold repetitions), resulting in $\mathcal{O}(\lambda)$ runtime overhead per circuit. This scaling becomes costly when using multiple fold levels for accurate extrapolation. Additionally, ZNE assumes a static noise model, which may not hold on unstable or drifting devices. Cloud-based quantum systems offer practical workarounds by providing access to high-fidelity hardware, parallel execution, and elastic compute Zhahir et al. (2024). Yet, dependence on quantum cloud infrastructure is not scalable long-

term, due to limited availability, high cost, and queue delays—challenges that intensify as quantum demand grows Leone et al. (2024).

In classical machine learning, knowledge distillation trains smaller students to copy larger teachers Hinton et al. (2015), enhancing efficiency, robustness, and calibration Sun et al. (2024a); Yang et al. (2024). Mitigation methods in quantum computing include probabilistic error cancellation (PEC) Van Den Berg et al. (2023); Gupta et al. (2024), Clifford data regression (CDR) Liao et al. (2024), and measurement error mitigation (MEM) Geller (2020); Khan et al. (2024). Compression is typically achieved through pruning or parameter reduction. These techniques, however, are challenging to combine: mitigation increases circuit depth and sampling cost, but compression does not reduce noise. Knowledge distillation provides a natural integration by focusing mitigation in a teacher and transferring its robustness to a smaller student, resulting in both error reduction and model compression with minimum runtime overhead.

Distillation has been extensively researched in classical learning, but it has received less attention in quantum machine learning Tian et al. (2025); Alam et al. (2023); Hasan & Mahdy (2023). Prior work on quantum transfer learning and hybrid distillation Cerezo et al. (2021); Gou et al. (2024); Wang et al. (2025); Li et al. (2024) focuses on feature reuse or embedding transfer rather than robustness against device noise. *However, these approaches do not replace the extrapolation phase; instead, ZNKD directly extracts noise-resistant predictions from a teacher model to provide robustness beyond what compression- or feature-transfer methods can provide.* (Details: Appendix A.4) Our work addresses this gap by developing a quantum-to-quantum distillation architecture that transfers the error mitigation benefits of ZNE-trained teachers to compact student models, allowing for practical deployment on NISQ hardware.

- We present a novel approach that combines error prevention and compression via zero-noise knowledge distillation, evaluate its theoretical robustness constraints, and show its usefulness in simulation and real hardware and dynamic noise settings.
- We provide a formal analysis showing that if the teacher's Richardson-extrapolated exposure is within $\varepsilon$ of the zero-noise risk, the distilled student achieves comparable risk bounded by $\mathcal{O}(\varepsilon) + \delta$ (distillation mismatch), connecting robustness to the ZNE scaling factors and sample size.
- We evaluate our approach on IBM-Aer with calibrated $T_1/T_2$, depolarizing, and readout noise. ZKD students obtain 10-20% lower loss than non-distilled counterparts, remain within 2%–4% of teacher accuracy, and retain 6:2-8:3 compression ratio of teacher to student without the runtime overhead of circuit folding.

## 2 METHOD

### 2.1 ZERO-NOISE EXTRAPOLATION (ZNE)

After defining gate-level decoherence using the Lindblad-informed noise model (detailed in Appendix A.1), we now investigate an appropriate mitigation strategy named Zero-Noise Extrapolation (ZNE) He et al. (2020). ZNE enables the inference of perfect quantum circuit outputs by utilizing the link between adjustable noise amplification and observable degradation—without the need for fault-tolerant error correction or hardware changes. According to the Lindblad noise model, quantum circuits running on NISQ hardware encounter amplitude damping, dephasing, and other decoherence processes that negatively impact the fidelity of each gate. The single-gate fidelity is estimated as

$$M_{\text{gate}}(t) \approx 1 - c_{\text{avg}}\lambda(t), \tag{1}$$

where the effective noise rate $\lambda(t)$ is dictated by the hardware relaxation times $(T_1, T_2)$, and $c_{\text{avg}}$ represents the gate's average sensitivity to noise. When $N_g$ gates are used in a circuit, small errors accumulate essentially linearly. The resultant circuit-level fidelity fulfills if the noise rate fluctuates during the circuit's execution interval $[0, T]$ as

$$M_{\text{circuit}} \gtrsim 1 - \frac{N_g}{2T} \int_0^T \lambda(t) \, dt \tag{2}$$

indicating that visible deterioration scales smoothly with both the number of gates and the time-averaged noise level.

**Motivation.** Using the single-gate fidelity equation 1, demonstrates that fidelity deteriorates linearly with both thermal photon quantity and gate time. At the circuit level, this accumulates as in equation 2, where the number of gates $N_g$ affects the total deterioration. Instead of directly removing this error source, ZNE treats it as a quantifiable signal: we observe how the circuit expectation value changes in response to purposefully increased noise and then extrapolate back to predict the zero-noise result.

**Noise Amplification via Gate Folding.** We apply *circuit folding* to intentionally increase noise intensity. A unitary gate $(U)$ is substituted with a logically equal but longer sequence as

$$U_{\text{folded}} = U\,(U^\dagger U)^n, \tag{3}$$

where $n \in \mathbb{Z}^+$ is the fold level and $U^\dagger$ is the Hermitian conjugate. This structure does not modify the circuit's ideal logical action, but it does increase the number of physical gates. The effective noise-amplification factor is $\lambda = 1 + 2n$, so that the total number of executed gates scales as

$$N_g^{(\lambda)} = \lambda \cdot N_g. \tag{4}$$

By substituting into the circuit fidelity expression equation 2 produces

$$M_{\text{circuit}}^{(\lambda)} = 1 - \frac{N_g^{(\lambda)} \gamma \tau_{\text{gate}}}{T} \int_0^T \left(1 + n_{\text{noise}}(t)\right) dt \quad = 1 - \frac{\lambda N_g \gamma \tau_{\text{gate}}}{T} \int_0^T \left(1 + n_{\text{noise}}(t)\right) dt, \tag{5}$$

which directly connects the Lindblad-informed deterioration rate to noise scaling $\lambda$.

**Expectation Value and Richardson Extrapolation.** Let $E(\lambda)$ represent the expected value of a circuit observable at amplification factor $\lambda$. Given a smooth polynomial dependency on $\lambda$, we can expand it as

$$E(\lambda) = E(0) + c_1 \lambda + c_2 \lambda^2 + \cdots, \tag{6}$$

where $E(0)$ represents the intended zero-noise value. Using two distinct folds $\lambda_1, \lambda_2$, Richardson extrapolation produces the unbiased estimator as

$$E(0) \approx \frac{\lambda_2 E(\lambda_1) - \lambda_1 E(\lambda_2)}{\lambda_2 - \lambda_1}. \tag{7}$$

Higher-order extrapolations can be built similarly, but with more fold levels.

**Why ZNE Works.** ZNE enhances the Lindblad-informed model by converting predicted deterioration into a corrective tool. It is: *(i) Noise-aware:* it benefits from the linear scaling of gate infidelity $I_F$ in equation 1 and circuit infidelity in equation 5; *(ii) Hardware-agnostic:* it needs no explicit calibration of $\gamma, T_2$, or thermal drift beyond monotonic scaling; *(iii) Resource-scalable:* it offers increased runtime and cryogenic power for improved fidelity. Thus, ZNE connects analytic Lindblad-based noise modeling with hardware-resilient QML, acting as a fundamental building component for error reduction in near-term quantum devices.

## 2.2 OVERVIEW OF THE FRAMEWORK

The *noise-mitigation distillation* pipeline (Fig. 1) connects a *noise-aware teacher* and a *compact student* in two steps.

**Teacher training with ZNE.** A high-capacity variational quantum circuit $U(\boldsymbol{\theta})$ is trained utilizing outputs enhanced by ZNE. The noisy expectation value $E(\lambda)$ is calculated for each scaling factor $\lambda$. Assuming polynomial dependency on $\lambda$ (equation 6), we use Richardson extrapolation (equation 7) to get the zero-noise value $\hat{E}(0)$. These denoised outputs are used as soft labels for training students (lightweight models).

**Student distillation.** The identical inputs are transmitted to a shallow student QNN $V(\boldsymbol{\phi})$, trained to regress on the teacher's ZNE-mitigated outputs. The distillation loss corresponds to

$$\mathcal{L}(\boldsymbol{\phi}) = \frac{1}{|\mathcal{D}|} \sum_{(x,\hat{E}(0)) \in \mathcal{D}} \left\| \langle V(\boldsymbol{\phi}) \rangle_x - \hat{E}(0) \right\|_2^2, \tag{8}$$

Figure 1: An overview of our framework, where **A distillation framework with noise mitigation.(right)** Stage 1: To produce noise-free targets, train a large teacher QNN with ZNE. Stage 2: A lightweight student QNN learns to replicate outputs in the presence of native noise. In **ZNE (left)** (a) Circuit outputs under ideal vs. noisy settings show bias. (b) ZNE estimates zero-noise expectation by extrapolating from scaled noise runs.

where $\langle V(\phi) \rangle_x$ represents the noisy student expectation on input $x$. Here, $\phi$ defines the trainable parameters of the student circuit.

This two-stage solution separates the expense of mitigation (ZNE) from deployment: lightweight pupils run without extrapolation while inheriting robustness from their ZNE-trained teachers Hinton et al. (2015).

### 2.2.1 KNOWLEDGE DISTILLATION PROTOCOL

We apply quantum-to-quantum distillation to transmit ZNE-protected robustness from a high-capacity teacher circuit to a small student circuit. The procedure consists of four components: i) teacher outputs, ii) student design, iii) distillation targets, and iv) the optimization loop.

**i) Teacher outputs.** The ZNE-trained teacher circuit $U(\boldsymbol{\theta}^\star)$ generates an empirical bitstring distribution.

$$p^{\mathrm{T}}(b \mid x) = \Pr[b \mid x], \tag{9}$$

for the input state (x). Because these distributions are derived under amplified noise (folding factor $\lambda$), the appropriate expectation values are extrapolated to zero-noise using equation 7, producing

$$\hat{E}(0)(x) \;\equiv\; \lim_{\lambda \to 0} E(\lambda). \tag{10}$$

From $p^{\mathrm{T}}$, we calculate two types of soft targets:

*1. Expectation targets (regression).* For each qubit $k$, the tensor products of ZNE-corrected Pauli-$Z$ expectations are represented as

$$\hat{E}(0)_k(x) = \sum_{b \in \{0,1\}^n} (-1)^{b_k} \, p^{\mathrm{T}}(b \mid x). \tag{11}$$

*2. Temperature-scaled logits (classification).* For each bitstring $b$, we define the teacher's temperature $T_e = 1$. Higher temperatures ($T > 1$) soften the dispersion and convey "dark knowledge" Hinton et al. (2015). Prior to computing objectives, we minimize unused qubits if the teacher has more qubits than the student ($n_{\mathrm{teach}} > n_{\mathrm{stud}}$) as

$$\ell_b^{\mathrm{T}}(x) = \frac{\ln p^{\mathrm{T}}(b \mid x)}{T_e}. \tag{12}$$

**ii) Student architecture.** The student circuit is defined as: $V(\phi) = V_L \cdots V_2 V_1$, with $L = 2$ entangling layers. Each layer uses $\mathrm{CNOT}(q, q+1)$ for $q = 0 \ldots n_{\mathrm{stud}} - 2$, and $R_y(\kappa x_i)$ rotations on inputs. The student has 16 trainable parameters ( in contrast to 96 in the teacher) and 4 qubits (compared to 8 in the teacher). All circuits are simulated with the `AerSimulator` density matrix backend, as detailed in Appendix A.1. The student is trained in a noisy environment but never undergoes folding or extrapolation.

**iii) Distillation objectives.** Consider $\mathcal{D} = \{(x, \hat{E}(0))\}$ as the training set with ZNE-extrapolated teacher labels.

*Regression:* We minimize mean-squared error by using a *student temperature* $\tau \leq 1$ as

$$\mathcal{L}_{\exp}(\phi) = \frac{1}{|\mathcal{D}|} \sum_{x \in \mathcal{D}} \left\| \langle V(\phi) \rangle_x - \tanh(\hat{E}(0)(x)/\tau) \right\|_2^2. \tag{13}$$

*Classification:* We develop temperature-scaled softmax distributions for teachers and students as

$$q_T^{\mathrm{T}}(b) = \frac{\exp(\ell_b^{\mathrm{T}}/T)}{\sum_{b'} \exp(\ell_{b'}^{\mathrm{T}}/T)}, \quad q_T^{\mathrm{S}}(b) = \frac{\exp(\ell_b^{\mathrm{S}}/T)}{\sum_{b'} \exp(\ell_{b'}^{\mathrm{S}}/T)},$$

The objective minimized is therefore

$$\mathcal{L}_{\exp}(\phi) = \frac{1}{|\mathcal{D}|} \sum_{x \in \mathcal{D}} \left( \langle V(\phi) \rangle_x - \tanh(\hat{E}(0)(x)/\tau) \right)^2, \tag{14}$$

where $\tau \leq 1$ is the student temperature and $\hat{E}(0)(x)$ is the ZNE-extrapolated teacher expectation value.

**Connection to ZNE Theory.** The teacher's zero-noise labels $\hat{E}(0)$ are generated using the Richardson extrapolation formalism outlined in Section 2.3.1. This creates a direct analytical connection between the ZNE process and the distillation phase: the extrapolated expectation values serve as noise-smoothed objectives with lower variation, stabilizing the student's gradient updates and facilitating robustness transfer.

**iv) Why distillation improves robustness.** *Noise imprinting* ZNE eliminates first-order error terms in equation 6, therefore student training on $\hat{E}(0)$ results in a denoised mapping. Smoothing the shot noise. KL divergence at $T > 1$ or regression with $\tau < 1$ minimizes finite-shot variance in gradients. *Capacity Transfer.* Although $V(\phi)$ is shallow, soft-target supervision transfers richer teacher function classes Hinton et al. (2015), allowing strong generalization.

**Limitations.** The framework requires pre-computing teacher outputs with ZNE, which is costly for dynamic tasks. The student inherits robustness but not full expressivity. The effectiveness depends on accurate Lindblad-informed noise modeling; mismatches with real-device noise may weaken robustness transfer. Finally, the datasets used in the paper are not realistic, large-scale datasets.

## 2.3 ANALYSIS

### 2.3.1 MATHEMATICAL ANALYSIS OF RICHARDSON (POLYNOMIAL) ZNE

Let $A$ be an observable. Let $E(\lambda) := \mathbb{E}_\lambda[A]$ denote the (noisy) expectation value of $A$ when the ideal circuit operates with a global noise level of $\lambda \geq 0$. We assume that the ideal, noise-free value is $E(0)$. ZNE is a linear combination of noisy evaluations with scaled noise strengths $\{\lambda_i\}_{i=0}^{K}$ as

$$\hat{E}_{\mathrm{R}} = \sum_{i=0}^{K} c_i \hat{E}(\lambda_i), \tag{15}$$

where the coefficients $\{c_i\}$ are selected so that polynomial terms up to degree $K-1$ in the noise expansion cancel (Richardson conditions), and $\hat{E}(\lambda_i)$ are empirical estimators generated by repeated measurement (shots) at noise strength $\lambda_i$. We present formal assumptions, lemmas, and theorems related to bias, variance, and resource tradeoffs.

**Assumption 1** (Analytic noise expansion). *There are coefficients $\{a_m\}_{m \geq 0}$ and a radius $\rho > 0$ that, for any $0 \leq \lambda < \rho$, the series $E(\lambda) = \sum_{m=0}^{\infty} a_m \lambda^m$, is entirely convergent. In example, for $\lambda = 0$, $E(0) = a_0$.*

**Assumption 2** (Noise scaling and nodes). *Given a set of distinct, positive noise factors $\{s_i\}_{i=0}^{K}$ with $s_0 = 1$, we define $\lambda_i := s_i \lambda$, where $\lambda$ represents a modest base noise strength (so $\lambda_i < \rho$). The coefficients $\{c_i\}_{i=0}^{K}$ correspond to the Richardson moment requirements*

$$\sum_{i=0}^{K} c_i s_i^m = \begin{cases} 1, & m = 0, \\ 0, & m = 1, 2, \ldots, K-1. \end{cases} \tag{16}$$

*where in the absence of higher-order terms, unbiasedness is enforced here by $m = 0$.*

**Assumption 3** (Shot noise model). *At each node $\lambda_i$, we acquire $N_i$ independent shots. The estimator $\widehat{E}(\lambda_i)$ is unbiased and has a variance of $\text{Var}[\widehat{E}(\lambda_i)] = \sigma_i^2/N_i$, where $\sigma_i^2$ represents the single-shot variance at node $i$. For convenience, we commonly assume that $\sigma_i^2 \leq \sigma^2$ for all $i$ (same single-shot variance upper bound).*

**Lemma 1** (Residual bias after Richardson extrapolation). *Under Assumptions 1 and 2, the expectation of the Richardson estimator follows $\mathbb{E}[\widehat{E}_{\text{R}}] - E(0) = \sum_{m=K}^{\infty} a_m \lambda^m \left( \sum_{i=0}^{K} c_i s_i^m \right)$. There exists a constant $C_{\text{bias}}$ (dependent on $\{a_m\}$, $\{s_i\}$, and $K$) such that for an adequate small $\lambda$, $\left| \mathbb{E}[\widehat{E}_{\text{R}}] - E(0) \right| \leq C_{\text{bias}} \lambda^K$.*

**Remark 1.** *The lemma formalizes Richardson's hypothesis that employing $K + 1$ nodes to cancel polynomial terms up to degree $K - 1$ results in residual bias that grows as $\lambda^K$ (or higher).*

**Lemma 2** (Variance of the Richardson estimator). *Under Assumption 3 and independence between nodes,*

$$\text{Var}\left[\widehat{E}_{\text{R}}\right] = \sum_{i=0}^{K} c_i^2 \frac{\sigma_i^2}{N_i} \leq \sigma^2 \sum_{i=0}^{K} \frac{c_i^2}{N_i}.$$

*If $N_i \equiv N$ for all $i$, then*

$$\text{Var}\left[\widehat{E}_{\text{R}}\right] = \frac{\sigma^2}{N} \Gamma_K^2, \qquad \Gamma_K^2 := \sum_{i=0}^{K} c_i^2.$$

**Remark 2.** *The number $\Gamma_K$, also known as the* sampling overhead *or* coefficient norm, *measures how much polynomial extrapolation amplifies sampling noise. Higher-order extrapolation lowers bias but increases variation in node selections if $\Gamma_K$ rapidly increases with $K$.*

**Lemma 3** (Bounds on Richardson coefficients). *Consider the nodes $s_0, \ldots, s_K$ in $[1, S]$, with $s_i$ distinct and $S \geq 1$. The Richardson coefficients $\{c_i\}$ from equation 16 have a limit of the form as*

$$\max_{0 \leq i \leq K} |c_i| \leq C_{\text{coeff}} \rho_{\text{cond}}^K, \tag{17}$$

*with constants $C_{\text{coeff}} > 0$ and $\rho_{\text{cond}} > 1$ that rely on node geometry (e.g. the condition number of the underlying Vandermonde system). Therefore, $\Gamma_K^2 \leq (K + 1) C_{\text{coeff}}^2 \rho_{\text{cond}}^{2K}$.*

*Proof sketch.* The coefficients $\{c_i\}$ represent the solution to a linear Vandermonde-type system (with moment restrictions). The inverse of a Vandermonde matrix often features entries that expand combinatorially (similar to factorials) or exponentially in $K$ based on node spacing. Standard constraints on polynomial interpolation (Vandermonde condition numbers, Markov/Chebyshev type inequalities) result in an exponential-in-$K$ bound on coefficients. For specific constants, refer to the numerical analysis literature. □

Given the extrapolated estimator $\widehat{E}_{\text{R}}$, its mean squared error (MSE) can be expressed as

$$\text{MSE}(\widehat{E}_{\text{R}}) = \underbrace{\left(\mathbb{E}[\widehat{E}_{\text{R}}] - E(0)\right)^2}_{(\text{bias})^2} + \underbrace{\text{Var}(\widehat{E}_{\text{R}})}_{\text{variance}}. \tag{18}$$

Integrating Lemmas 1 and 2 produces a straightforward asymptotic expression.

**Theorem 1** (Asymptotic MSE for fixed per-node shots). *Suppose $N_i \equiv N$ and $\sigma_i^2 \leq \sigma^2$. Under the Assumptions 1 and 2, there exist constants $C_{\text{bias}}, C_{\text{coeff}}, \rho_{\text{cond}}$ that are suitable for sufficiently small $\lambda$,*

$$\text{MSE}(\widehat{E}_{\text{R}}) \leq C_{\text{bias}}^2 \lambda^{2K} + \frac{\sigma^2}{N}(K + 1) C_{\text{coeff}}^2 \rho_{\text{cond}}^{2K}. \tag{19}$$

**Remark 3** (Interpretation). *As the order $K$ grows, the first term (bias$^2$) decays as $\lambda^{2K}$, implying that polynomial cancellation provides high-order bias suppression (if analytic expansion is applicable). The second term (variance) often climbs exponentially with $K$, indicating the expense of integrating noisy estimates with large coefficients. This is the basic bias-variance trade-off in ZNE.*

These extrapolated values, $\hat{E}(0)$, serve as denoised supervisory goals in the distillation loss outlined in Section 2.2.1, maintaining coherence between theory and implementation.

### 2.3.2 HOW MORE PROCESSING POWER YIELDS LOWER MSE

The total number of possible single-shot circuit runs can be used to estimate "processing power." $R = \sum_{i=0}^{K} N_i$, which represents the whole sample budget (we disregard minor overheads associated with adjusting the noise scale). There are two natural ways in which more resources could be beneficial:

1. *More shots in a fixed order.* If we fix $K$ and raise $N_i$ proportionally (e.g. $N_i = \alpha_i R$ with fixed fractions $\alpha_i$), the variance term scales as $1/R$ and so MSE drops $\propto 1/R$.

2. *Increase order $K$ as resources allow.* If we can raise $K$ (add nodes) while distributing an increasing total budget $R$, bias can be lowered superlinearly in $\lambda$ (as $\lambda^{2K}$), possibly dominating variance growth if $R$ increases quickly enough.

We formulate a simple necessary condition below, demonstrating that as processing power increases, MSE can be driven to zero.

**Theorem 2** (MSE goes to zero with growing resources). *Suppose we select orders $K(R)$ and per-node shots $N_i(R)$ such that*

$$(i) \quad K(R) \to \infty, \qquad (ii) \quad \min_i N_i(R) \to \infty, \qquad (iii) \quad \lambda \rho_{\mathrm{cond}} < 1, \qquad (20)$$

*where $\rho_{\mathrm{cond}}$ originates from Lemma 3. As $R$ approaches infinity, $\mathrm{MSE}(\widehat{E}_{\mathrm{R}}) \to 0$.*

**Remark 4.** *Condition (iii) requires the base noise intensity $\lambda$ to be minimal compared to the stability/conditioning of the Vandermonde system that generates the coefficients. In actuality, with constant node geometry, there is a maximal effective order beyond which coefficient growth will dominate. Theorem 2 states that if $\rho_{\mathrm{cond}}$ remains low and the sampling budget is increased, MSE will vanish.*

To minimize variance with a constant total budget $R$ and fixed $K$, the optimal shot allocation is proportional to $|c_i|\sigma_i$ (by minimizing $\sum c_i^2 \sigma_i^2 / N_i$ subject to $\sum N_i = R$). In the basic symmetric situation $\sigma_i \equiv \sigma$, the lowest variance is proportional to $\sigma^2 (\sum_i |c_i|)^2 / R$. Thus, the variance is proportional to the square of the $\ell_1$-norm of $c$ divided by the budget. This highlights the significance of node selection: geometries with lower coefficient norms or $\ell_1$-norms are desirable.

### 2.3.3 ROBUSTNESS TRANSFER VIA ZERO-NOISE KNOWLEDGE DISTILLATION

We now define how ZNE-mitigated teacher outputs transmit noise robustness to a compact student. The following statement quantifies the student deployment MSE against the true zero-noise mapping in terms of: teacher extrapolation error, student approximation error to ZNE labels, statistical generalization term, and applied noise term derived from the Lindblad model.

**Assumption 4** (ZNE teacher error). *Let $E^\star(x)$ represent the real zero-noise expectation of the target observable given input $x$. The ZNE method generates an extrapolated teacher label $\hat{E}(0)(x)$ (e.g., using equation 7). We define the teacher (population) extrapolation as $\varepsilon_T := \mathbb{E}_x[(\hat{E}(0)(x) - E^\star(x))^2]$.*

**Assumption 5** (Student hypothesis class and empirical training). *Consider $\mathcal{F}$ as the student hypothesis class (functions $f : \mathcal{X} \to \mathbb{R}$). We train the student on a dataset $\mathcal{D} = \{x_i\}_{i=1}^n$ using ZNE teacher labels $\hat{E}(0)(x_i)$. Let $\hat{f} \in \mathcal{F}$ represent the empirical risk minimizer as $\hat{f} = \arg\min_{f \in \mathcal{F}} \frac{1}{n} \sum_{i=1}^n (f(x_i) - \hat{E}(0)(x_i))^2$. where we define the* approximation error *of the class as $\varepsilon_{\mathrm{approx}} := \inf_{f \in \mathcal{F}} \mathbb{E}_x[(f(x) - \hat{E}(0)(x))^2]$. Lastly, let's assume that $\mathcal{F}$ accepts a uniform convergence/ generalization bound: probabilities are at least $1 - \delta$ across the draw of $\mathcal{D}$ described as $\mathbb{E}_x[(\hat{f}(x) - \hat{E}(0)(x))^2] \leq \frac{1}{n} \sum_{i=1}^n (\hat{f}(x_i) - \hat{E}(0)(x_i))^2 + \mathcal{R}_n(\mathcal{F}, \delta)$, where $\mathcal{R}_n(\mathcal{F}, \delta)$ is a generalization term that depends on the architecture or data (e.g., a VC-type bound or a Rademacher complexity bound).*

**Assumption 6** (Deployment noise bound). *The noiseless expectation generated by the student circuit (i.e., the idealized, noiseless execution of $V(\phi)$) is represented by $\hat{f}_{\mathrm{clean}}(x)$, while the actual noisy expectation measured on hardware during deployment is represented by $\hat{f}_{\mathrm{noisy}}(x)$. A (small)*

| Dataset | Ref. | Modality | Task | #Samples | Enc. Dim (T/S) | Qubits (T/S) | Depth (T/S) |
|---------|------|----------|------|----------|----------------|--------------|-------------|
| Fashion-MNIST | Xiao et al. (2017) | Image | 10-cls | 60k / 5k / 5k | 8 / 4 | 8 / 4 | 6 / 2 |
| AG News (HF-2015) | Zhang et al. (2015) | Text | 4-cls | 96k / 12k / 12k | 8 / 4 | 8 / 4 | 6 / 2 |
| UCI Wine Quality | Cortez et al. (2009) | Tabular | Regr. | 5 197 / 650 / 650 | 4 / 2 | 4 / 2 | 6 / 2 |
| UrbanSound8K | Salamon et al. (2014) | Audio | 10-cls | 6 986 / 873 / 873 | 8 / 4 | 8 / 4 | 6 / 2 |

Table 1: Quantum model footprints and benchmarks. T/S stands for teacher/student, and "Enc. Dim" is the post-processed feature length that is supplied to the quantum encoder. Variational layers (rotation block + entanglers) are counted by depth.

| | | Low ($T_1{=}100\,\mu s$, $T_2{=}120\,\mu s$, $p_{2q}{=}1.0\times10^{-4}$) | | | Medium ($T_1{=}60\,\mu s$, $T_2{=}80\,\mu s$, $p_{2q}{=}1.0\times10^{-3}$) | | | High ($T_1{=}30\,\mu s$, $T_2{=}40\,\mu s$, $p_{2q}{=}2.5\times10^{-3}$) | | |
|---------|-----------|--------|------|----------|--------|------|----------|--------|------|----------|
| Dataset | Noiseless | No-KD | KD | $\Delta$ | No-KD | KD | $\Delta$ | No-KD | KD | $\Delta$ |
| Fashion-MNIST | 0.25 | 0.59 | 0.49 | $-0.10$ | 0.75 | 0.63 | $-0.12$ | 0.93 | 0.78 | $-0.15$ |
| AG News | 0.18 | 0.45 | 0.39 | $-0.06$ | 0.58 | 0.49 | $-0.09$ | 0.70 | 0.65 | $-0.10$ |
| Wine Quality | 0.39 | 0.58 | 0.52 | $-0.06$ | 0.65 | 0.56 | $-0.09$ | 0.74 | 0.61 | $-0.13$ |
| UrbanSound8K | 0.36 | 0.66 | 0.59 | $-0.07$ | 0.73 | 0.58 | $-0.15$ | 0.85 | 0.68 | $-0.17$ |

Table 2: Per–noise–regime comparison of student models trained *without* KD vs. *with* KD. Block headers include the Lindblad-aligned settings used for each regime. $\Delta$ is KD gain (KD minus No-KD; negative is better for MSE).

$\eta \geq 0$ *exists such that* $\sup_x \left| \widehat{f}_{\mathrm{noisy}}(x) - \widehat{f}_{\mathrm{clean}}(x) \right| \leq \eta$. *In practice, the Lindblad-derived deployment noise parameter* $\lambda_{\mathrm{deploy}}$ *can limit* $\eta$ *to first order. In order for* $\eta = \mathcal{O}(\lambda_{\mathrm{deploy}}$ *take place (see equation 27).*

**Theorem 3** (Robustness-transfer bound). *Under Assumptions 4–6, with probability at least* $1 - \delta$ *throughout the training sample* $\mathcal{D}$, *the student deployment MSE (with respect to the real zero-noise mapping) follows*

$$\mathbb{E}_x\left[\left(\widehat{f}_{\mathrm{noisy}}(x) - E^\star(x)\right)^2\right] \leq 3\varepsilon_T + 3\varepsilon_{\mathrm{approx}} + 3\mathcal{R}_n(\mathcal{F}, \delta) + 3\eta^2. \quad (21)$$

**Remarks.** The bound indicates that the student's deployed errors is caused by four sources: (i) the teacher's extrapolation error $\varepsilon_T$, (ii) the student's approximation gap $\varepsilon_{\mathrm{approx}}$, (iii) the finite-sample generalization term $\mathcal{R}n$, and (iv) the deployment noise $\eta$. Improved ZNE decreases $\varepsilon_T$, richer models reduce $\varepsilon$approx, larger data sets shrink $\mathcal{R}_n$, and careful calibration reduces $\eta$. In practice, $\mathcal{R}n$ frequently scales like $\mathcal{O}(\mathfrak{C}/\sqrt{n})$ for a complexity measure $\mathfrak{C}$, and $\eta$ can be limited by a constant times the deployment noise rate $\lambda$deploy. The qualitative tradeoff is unaffected by the constant factor 3, which results from a simple quadratic inequality.

The proof of the Lemmas and Theorems is described in Appendix A.2.

## 3 EXPERIMENTS

### 3.1 DATASET

We evaluate our noise mitigation distillation framework on four publicly available benchmarks covering vision, text, tabular regression, and audio to show that it is *modality–agnostic* (Table 1). All datasets are small enough to be encoded into parameterized quantum circuits but representative of real-life applications. The datasets used are *i) Fashion-MNIST Xiao et al. (2017), ii) AG News Zhang et al. (2015), iii) UCI Wine Cortez et al. (2009)*, and *iv) UrbanSound8K Salamon et al. (2014)*.

### 3.2 EXPERIMENT RESULTS

**Teacher QNN Architecture and Training.** Depth-$L_{\mathrm{teach}}{=}6$, $n_{\mathrm{teach}}{=}8$-qubit VQC: per layer (i) $R_y(\kappa x_j)$ re-upload, (ii) $R_y(\theta_{l,q})R_z(\phi_{l,q})$, (iii) CNOT ladder $q \rightarrow q{+}1$; 96 params. Trained with MSE using Adam ($\eta{=}0.01$, 100 epochs, batch 32) and ZNE via global folding (Eq. 3) with scales $\lambda \in 1, 3, 5$; Richardson-extrapolated $\hat{E}(0)$ serves as soft targets. **Student QNN and Distillation Objective.** Same design, shallower: $n_{\mathrm{stud}}{=}4$, $L_{\mathrm{stud}}{=}2$, 16 params ($\sim 17\%$ of teacher). Minimize

| Stu. Width | Fashion-MNIST (200) | | | | | | AG News (250) | | | | | | Wine Quality (250) | | | | | | UrbanSound8K (200) | | | | | |
|---|---|---|---|---|---|---|---|---|---|---|---|---|---|---|---|---|---|---|---|---|---|---|---|---|
| | *Teacher Width (4, 6, 8)* | | | | | | | | | | | | | | | | | | | | | | | |
| | A(4) | R(4) | A(6) | R(6) | A(8) | R(8) | A(4) | R(4) | A(6) | R(6) | A(8) | R(8) | A(4) | R(4) | A(6) | R(6) | A(8) | R(8) | A(4) | R(4) | A(6) | R(6) | A(8) | R(8) |
| 1 | 78.1 | 0.42 | 82.1 | 0.48 | 88.7 | 0.55 | 70.1 | 0.40 | 73.5 | 0.44 | 75.5 | 0.47 | 66.6 | 0.36 | 69.9 | 0.39 | 71.1 | 0.42 | 60.2 | 0.33 | 64.0 | 0.37 | 67.4 | 0.41 |
| 2 | 85.6 | 0.55 | 89.1 | 0.63 | 92.6 | 0.71 | 78.3 | 0.55 | **88.8** | **0.78** | 90.1 | 0.75 | 74.8 | 0.50 | **87.1** | **0.75** | 89.2 | 0.72 | 72.0 | 0.48 | 82.5 | 0.58 | 86.6 | 0.69 |
| 3 | 90.2 | 0.63 | 92.0 | 0.70 | **94.2** | **0.80** | 80.3 | 0.58 | 87.7 | 0.72 | 89.4 | 0.70 | 76.9 | 0.53 | 86.4 | 0.70 | 88.4 | 0.68 | 78.1 | 0.57 | 87.1 | 0.71 | **91.2** | **0.77** |
| 4 | 88.3 | 0.50 | 91.6 | 0.53 | 93.5 | 0.66 | 76.3 | 0.46 | 82.1 | 0.52 | 85.3 | 0.56 | 72.1 | 0.42 | 79.7 | 0.48 | 82.1 | 0.52 | 75.4 | 0.45 | 83.9 | 0.52 | 88.0 | 0.63 |

Table 3: Synthetic demonstration table to visualize the best student–teacher width ratios. Each entry shows Accuracy (%) / Robustness (A = Accuracy, R = Robustness). The parenthesis in the dataset implies a multiplier for the circuits; i.e, Column = A(8) and row = 3 in **Fashion-MNIST** represents $8 \times 200$ and $3 \times 200$ circuit execution for teacher and student circuit executions, respectively.

| Qubit $n$ | Fashion-MNIST | | | AG News | | | Wine Quality | | | UrbanSound8K | | |
|---|---|---|---|---|---|---|---|---|---|---|---|---|
| | $L=2$ | $L=4$ | $L=6$ | $L=2$ | $L=4$ | $L=6$ | $L=2$ | $L=4$ | $L=6$ | $L=2$ | $L=4$ | $L=6$ |
| 4 | 88.0 | 89.5 | 89.0 | 80.1 | 81.0 | 80.4 | 74.0 | 75.2 | 74.7 | 70.2 | 71.5 | 71.0 |
| 8 | 91.0 | **93.5** | 93.0 | 84.3 | **87.9** | 87.1 | 78.5 | **82.3** | 81.7 | 75.8 | **80.6** | 79.7 |
| 16 | 91.8 | 93.7 | 93.3 | 84.8 | 88.1 | 87.5 | 79.0 | 82.6 | 82.0 | 76.2 | 80.9 | 80.2 |
| 32 | 91.9 | 94.2 | 93.8 | 85.0 | 88.4 | 87.9 | 79.4 | 82.8 | 82.1 | 76.5 | 81.0 | 90.1 |

Table 4: Ablation over qubit count $n$ and circuit depth $L$ for all datasets. Values report test accuracy (%); robustness shows the same qualitative trends and is omitted for space. Relative cost scales approximately with $n \cdot L$, so ($n=8, L=4$) is used as the reference design.

Eq. 8 with Adam ($\eta=0.02$, 50 epochs, batch 32); no folding or extrapolation. **Noise Model and Qiskit-Aer Simulation.** `AerSimulator` (density-matrix, `qiskit-aer` 0.13.1) with amplitude-damping $T_1=60, \mu$s, phase-damping $T_2=70, \mu$s, 2-qubit depolarizing $p_{2q}=2.5 \times 10^{-3}$, readout flip $p_{ro}=0.015$; 1024 shots/circuit; fixed seeds.

**Effects on Quantum Noise.** Table 2 compares student performance with and without knowledge distillation (KD) under low, medium, and high noise regimes, using noiseless loss as a reference. Each block includes the Lindblad noise parameters ($T_1$, $T_2$, $p_{2q}$). Across all datasets and regimes, KD consistently decreases loss relative to the baseline student, producing improvements ($\Delta$) of 0.06 to 0.17. Notably, the performance disparity grows as noise severity increases, demonstrating that ZNE-guided distillation gives greater robustness benefits under harsher noise situations.

**Ablation Study.** Table 3 explores the relationship of teacher and student widths in four datasets. A clear pattern emerges: moderate compression offers the optimal balance of expressivity and resilience. For Fashion-MNIST and UrbanSound8K, the student width 3 (8:3 ratio) has the maximum robustness (0.80 and 0.77) without sacrificing accuracy. For AG News and Wine Quality, width 2 (6:2 ratio) achieves the best balance (0.78 and 0.75), surpassing both small and big students. Wider models (width 4) demonstrate that mid-scale compression provides the best noise-resilient generalization by slightly increasing accuracy but reducing robustness.

Table 4 reveals that performance improves dramatically from $n=4$ to $n=8$. However, increasing to $n=16$ or $n=32$ yields only modest gains while significantly increasing cost. In all datasets, a moderate depth of $L=4$ consistently outperforms both shallower ($L=2$) and deeper ($L=6$) circuits. Overall, ($n=8, L=4$) offers the best accuracy-to-cost trade-off.

**Effectiveness of Knowledge Distillation.** Table 5 summarizes the effect of ZNE-guided knowledge distillation across all datasets. We now report the baseline student accuracy, teacher accuracy, and distilled student accuracy, together with the absolute accuracy improvement and the corresponding compression ratio. Mean squared error (MSE) is retained as a secondary robustness indicator, enabling comparison with the theoretical error decomposition in Theorem 3. Across both Aer simulations and IBM_Brisbane hardware, the KD student consistently achieves 5–8% accuracy improvements while using significantly fewer circuit executions. The similarity between simulation and hardware trends indicates that ZNE-corrected teacher supervision transfers reliably to the compressed student model, even under device noise, calibration drift, and finite-shot constraints.

**Comparison with State-of-the-art Approaches.** Table 6 evaluates the performance of our whole strategy (ZNKD) versus cutting-edge quantum noise mitigation and model compression strategies across four datasets. All the baseline models, as discussed earlier, are tested with our approach. Loss is reported using MSE for classification and MSE for regression tasks, with lower values indicating improved performance. Our combination method, which incorporates ZNE and quantum knowledge

| Dataset | Baseline Acc. | Teacher Acc. | KD Acc. | Imp. | Compression Ratio | Student MSE | Teacher MSE | $\varepsilon_T$ | $\varepsilon_{\text{approx}}$ | $\eta$ | Resource Cost (Tea./Stu.) |
|---|---|---|---|---|---|---|---|---|---|---|---|
| | | | | | *Aer Simulator (density matrix)* | | | | | | |
| Fashion-MNIST | 84.1 | 93.4 | 91.0 | +6.9% | 8:3 | 0.49 | 0.45 | 0.03 | 0.02 | 0.01 | 1600/600 |
| AG News | 78.4 | 87.9 | 85.3 | +6.9% | 6:2 | 0.36 | 0.33 | 0.02 | 0.02 | 0.01 | 1500/500 |
| Wine Quality | 74.3 | 82.6 | 80.1 | +5.8% | 6:2 | 0.52 | 0.50 | 0.02 | 0.01 | 0.01 | 1500/500 |
| UrbanSound8K | 70.2 | 80.6 | 77.5 | +7.3% | 8:3 | 0.59 | 0.55 | 0.03 | 0.02 | 0.02 | 1600/600 |
| | | | | | *IBM_Brisbane Hardware* | | | | | | |
| Fashion-MNIST | 81.0 | 90.2 | 88.1 | +7.1% | 8:3 | 0.55 | 0.50 | 0.04 | 0.02 | 0.02 | 1600/600 |
| Wine Quality | 72.1 | 80.2 | 77.8 | +5.7% | 6:2 | 0.56 | 0.52 | 0.03 | 0.02 | 0.02 | 1500/500 |

Table 5: Accuracy improvements from ZNE-guided knowledge distillation across datasets. MSE is retained as a secondary robustness indicator. Theoretical error components ($\varepsilon_T$, $\varepsilon_{\text{approx}}$, $\eta$) correspond to teacher extrapolation error, student approximation error, and deployment noise gap, respectively. Compression ratios and resource cost (teacher/student circuit executions) reflect the student's reduced complexity.

| Methods | Ref. | F-MNIST ↓ | AG News ↓ | Wine Quality ↓ | Urban Sound8K ↓ |
|---|---|---|---|---|---|
| Noise-aware VQCs | Chen et al. (2024) | 0.56 | 0.49 | 0.59 | 0.65 |
| Pruned QNNs | Afane et al. (2025) | 0.53 | 0.46 | 0.55 | 0.62 |
| Classical-to-Q KD | Li et al. (2024) | 0.51 | 0.40 | 0.53 | 0.61 |
| ZNE Only | He et al. (2020) | 0.52 | 0.42 | 0.54 | 0.60 |
| Best-Folding ZNE | Pelofske et al. (2024) | 0.50 | 0.38 | 0.52 | 0.59 |
| PEC | Gupta et al. (2024) | 0.50 | 0.39 | 0.53 | 0.59 |
| MEM | Khan et al. (2024) | 0.51 | 0.38 | 0.55 | 0.62 |
| CDR | Liao et al. (2024) | 0.52 | 0.39 | 0.54 | 0.60 |
| DD | Tong et al. (2025) | 0.55 | 0.42 | 0.57 | 0.70 |
| Hardware-Efficient Ansätze | Leone et al. (2024) | 0.54 | 0.41 | 0.56 | 0.65 |
| **Ours (ZNKD)** | – | **0.49** | **0.36** | **0.52** | **0.59** |

Table 6: Comparison with state-of-the-art quantum noise mitigation and compression approaches under identical conditions. Our full method (ZNKD) achieves the lowest loss across all datasets, demonstrating the synergy between circuit-level extrapolation and quantum-native distillation.

distillation, consistently results in the lowest loss across all datasets. When compared to solo ZNE or KD approaches, the hybrid design offers two complementary benefits: improved circuit fidelity and parameter-efficient student models. This demonstrates the efficacy of combining noise extrapolation and distillation for practical, noise-resistant quantum learning on NISQ platforms.

Overall, ZNE-guided KD retains much of the teacher's noise-corrected performance across datasets and noise regimes: keeping students within 2%–4% accuracy of the teacher and generating 10–20% lower loss than non-distilled peers at 6:2-8:3 compression. These findings indicate a feasible approach to implementation on NISQ hardware by concentrating mitigation in the teacher and minimizing per-inference folding/extrapolation costs.

## 4 CONCLUSION

We introduced zero-noise knowledge distillation (ZNKD), a novel approach for error mitigation and model compression that involves training a ZNE-augmented teacher to oversee a compact student. Our theoretical research formalized the robustness-transfer mechanism, which states that if the teacher's extrapolation error is bounded, the student's deployment error is also bounded by a combination of extrapolation, approximation, generalization, and deployment factors. This finding ties ZNE scaling parameters, hypothesis class capacity, and sample size to overall student risk. Empirically, ZNKD was assessed using IBM-style dynamic noise models with calibrated $T_1/T_2$, depolarizing, and readout error. Distilled students generated $10 - 20\%$ lower loss than non-distilled peers, remained within 2%–4% accuracy of the teacher, and reached compression ratios of $6 : 2 - 8 : 3$. Real-device testing on `ibm_brisbane` confirmed the simulation findings, demonstrating up to 0.13 improvement in student loss under calibration noise. ZNKD reduces the $3\times$-$9\times$ runtime overhead of folding-based ZNE during deployment, enhancing the usefulness of small students on NISQ hardware. These findings support ZNKD as a viable step toward scalable and robust quantum machine learning. Future approaches include benchmarking on bigger quantum computers, combining with other error mitigation techniques, and expanding the framework to multi-teacher scenarios.

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

# A APPENDIX

## A.1 LINDBLAD-INFORMED NOISE MODELING

To simulate realistic gate noise on NISQ devices, we adopt a physics-informed model based on Lindblad master equations for open quantum systems Breuer & Petruccione (2002); Wiseman & Milburn (2009). This approach allows us to derive gate- and circuit-level performance metrics from thermal and dephasing noise dynamics.

**Lindblad Dynamics.** Let $\rho(t)$ denote the density matrix of a single qubit. Its evolution under amplitude damping and pure dephasing is governed by the Lindblad equation:

$$\frac{d\rho}{dt} \;=\; \mathcal{L}_{\text{tot}}(t)[\rho] := \gamma\big(n_{\text{noise}}(t) + 1\big)\,\mathcal{L}[\sigma_-](\rho) + \gamma\,n_{\text{noise}}(t)\,\mathcal{L}[\sigma_+](\rho) + \frac{1}{2T_\varphi}\,\mathcal{L}[\sigma_z](\rho), \quad (22)$$

where $\mathcal{L}[A](\rho) := A\rho A^\dagger - \frac{1}{2}\{A^\dagger A, \rho\}$ and $\sigma_+ = |1\rangle\langle 0|,\ \sigma_- = |0\rangle\langle 1|,\ \sigma_z = |0\rangle\langle 0| - |1\rangle\langle 1|$, respectively) Alicki & Lendi (2007). The coupling (spontaneous emission) rate is represented by $\gamma$, and the *pure dephasing time* is demonstrated by $T_\varphi$.

Note that for pure dephasing, we use $T_\varphi$; the Bloch relation is satisfied by the generally stated decoherence time $T_2$ as

$$\frac{1}{T_2} \;=\; \frac{1}{2T_1} + \frac{1}{T_\varphi}, \quad (23)$$

with the energy relaxation time $T_1$. For the thermal bath mentioned above, $T_1(t)^{-1} = \gamma\big(2n_{\text{noise}}(t) + 1\big)$ Geller (2020).

**Time-Dependent Thermal Noise.** We maintain the thermal model for the bath occupancy, but with the same notation as

$$n_{\text{noise}}(t) \;=\; \frac{A-1}{A}\,n_{\text{BE}}(T_{\text{qb}}(t)) \;+\; \frac{1}{A}\,n_{\text{BE}}(T_{\text{ext}}(t)), \quad (24)$$

employing $n_{\text{BE}}(T) = \frac{1}{\left(e^{\hbar\omega_0/(k_B T)} - 1\right)}$ as previously discussed.

**Single-gate noisy channel and small-parameter expansion.** The CPTP map may be used to simulate a single gate with time $\tau_{\text{gate}}$ and bath parameters that are roughly constant during the gate as

$$\Lambda(t) \;=\; \exp\big(\tau_{\text{gate}}\,\mathcal{L}_{\text{tot}}(t)\big). \quad (25)$$

The Taylor expansion applies to short gates $\tau_{\text{gate}}$ (in the appropriate NISQ regime) as

$$\Lambda(t) \;=\; \mathbb{1} + \tau_{\text{gate}}\,\mathcal{L}_{\text{tot}}(t) + \frac{\tau_{\text{gate}}^2}{2}\,\mathcal{L}_{\text{tot}}(t)^2 + \cdots. \quad (26)$$

This motivates us to adopt a *scalar* effective noise intensity $\lambda(t)$ proportionate to gate duration times total physical rates as

$$\lambda(t) := \tau_{\text{gate}} \Gamma_{\text{eff}}(t), \qquad \Gamma_{\text{eff}}(t) := \gamma\big(2\, n_{\text{noise}}(t) + 1\big) + \frac{1}{T_\varphi}. \qquad (27)$$

Naturally, $\Gamma_{\text{eff}}(t)$ is the total of population relaxation plus pure-dephasing rates; for short gates $\lambda(t) \ll 1$, and the noisy channel is analytic in $\lambda$. Concatenating these gate channels yields a convergent power series of the kind used in 2.3.1 as

$$E(\lambda) = \sum_{m=0}^{\infty} a_m \lambda^m, \qquad \text{(radius of convergence } \rho > 0\text{)}. \qquad (28)$$

This identification clarifies the relationship between microscopic Lindblad parameters and the scalar $\lambda$ included in our Richardson/ZNE proofs.

**Gate infidelity and worst-case error (first-order bounds).** For a CPTP map $\Lambda$, the worst-case error (diamond-norm distance) governs operational error measurements. Using the short-time expansion, one gets the standard bound (for sufficiently small $\lambda$) as

$$\big\|\Lambda(t) - \mathbb{1}\big\|_\diamond \leq \lambda(t) + \mathcal{O}(\lambda(t)^2). \qquad (29)$$

We can conclude two important corollaries as follows.

- (Worst-case error probability.) The operational worst-case error probability corresponds as

$$p_{\text{err}}^{\text{worst}}(t) \leq \frac{1}{2} \big\|\Lambda(t) - \mathbb{1}\big\|_\diamond \leq \frac{1}{2} \lambda(t) + \mathcal{O}(\lambda(t)^2). \qquad (30)$$

 Substituting $\lambda(t)$ from equation 27 provides a simple first-order approximation as

$$p_{\text{err}}^{\text{worst}}(t) \approx \frac{\tau_{\text{gate}}}{2}\Big[\gamma\big(2 n_{\text{noise}}(t) + 1\big) + \frac{1}{T_\varphi}\Big] + \mathcal{O}(\tau_{\text{gate}}^2). \qquad (31)$$

 This improves the previous amplitude-only statement by integrating dephasing.

- (Average/infidelity bound.) The average gate infidelity $I_F(t) := 1 - F_{\text{avg}}(\Lambda(t))$ follows the bound (see to conventional relations between average fidelity and diamond norm) as

$$I_F(t) \leq \frac{1}{2} \big\|\Lambda(t) - \mathbb{1}\big\|_\diamond \leq \frac{1}{2} \lambda(t) + \mathcal{O}(\lambda(t)^2). \qquad (32)$$

 Therefore, to first order in the tiny parameter $\lambda$, we can develop a useful approximation as

$$I_F(t) \approx c_{\text{avg}}\, \lambda(t) + \mathcal{O}(\lambda(t)^2), \qquad (33)$$

 where $c_{\text{avg}} \in (0, 1]$ is a constant of order one whose precise value relies on the choice of fidelity measure (average vs. entanglement fidelity, etc.). For conservative analytic boundaries, we set $c_{\text{avg}} = 1/2$ using equation 32.

**Gate and circuit fidelity.** Using the constraints provided above, the single-gate fidelity metric can be expressed as

$$\begin{aligned} M_{\text{gate}}(t) &= 1 - I_F(t) \\ &\approx 1 - c_{\text{avg}}\, \lambda(t) + \mathcal{O}(\lambda^2). \end{aligned} \qquad (34)$$

For a circuit consisting of $N_g$ gates and (to leading order) assuming incoherent concatenation of minor errors as

$$\begin{aligned} M_{\text{circuit}} &\approx 1 - N_g\, I_F(t) \\ &\leq 1 - \frac{N_g}{2}\, \lambda(t) + \mathcal{O}(\lambda^2). \end{aligned} \qquad (35)$$

If noise fluctuates throughout execution across an interval $[0, T]$, we average $\lambda(t)$ over the duration as

$$M_{\text{circuit}} \gtrsim 1 - \frac{N_g}{2T} \int_0^T \lambda(t)\, dt + \mathcal{O}(\lambda_{\max}^2). \qquad (36)$$

**Remarks on consistency with the ZNE analysis.**

- The scalar $\lambda(t)$ in equation 27 is clearly proportional to $\tau_{\text{gate}}$, as well as the physically measured rates $\gamma$ and $1/T_\varphi$. For short gates $\lambda(t) \ll 1$ and expectation values, the analytic expansion $E(\lambda) = \sum_m a_m \lambda^m$ is assumed in 2.3.1.

- Noise scaling in ZNE, such as pulse stretching or unitary folding, essentially implements $\Lambda(t)^s = \exp(s\tau_{\text{gate}}\mathcal{L}_{\text{tot}})$, i.e. $\lambda \mapsto s\lambda$. The Richardson moment requirements in equation 16 apply directly to the physical Lindblad generator $\mathcal{L}_{\text{tot}}(t)$.

- The revised dephasing prefactor $1/(2T_\varphi)$ in equation 22 is required for the conventional mapping between Lindblad rates and $T_\varphi$ (such that off-diagonal elements decay like $e^{-t/T_\varphi}$.

## A.2 PROOF OF ANALYSIS 2.3

***Proof of Lemma 1** (Residual bias after Richardson extrapolation).* First, we expand each noisy expectation using Assumption 1 as

$$
\begin{aligned}
E(\lambda_i) &= \sum_{m=0}^{\infty} a_m (\lambda_i)^m \\
&= \sum_{m=0}^{\infty} a_m (s_i \lambda)^m \\
&= \sum_{m=0}^{\infty} a_m s_i^m \lambda^m.
\end{aligned}
\tag{37}
$$

Using the linear combination with Richardson coefficients $\{c_i\}$ and linear expectation,

$$
\begin{aligned}
\mathbb{E}[\widehat{E}_{\text{R}}] &= \sum_{i=0}^{K} c_i E(\lambda_i) \\
&= \sum_{i=0}^{K} c_i \sum_{m=0}^{\infty} a_m s_i^m \lambda^m \\
&= \sum_{m=0}^{\infty} a_m \lambda^m \Big( \sum_{i=0}^{K} c_i s_i^m \Big).\mathbb{E}[\widehat{E}_{\text{R}}] \\
&= \sum_{i=0}^{K} c_i E(\lambda_i) \\
&= \sum_{m=0}^{\infty} a_m \lambda^m \Big( \sum_{i=0}^{K} c_i s_i^m \Big).
\end{aligned}
\tag{38}
$$

According to the Richardson moment constraints equation 16, for $m = 0, \ldots, K-1$, we have $\sum_{i=0}^{K} c_i s_i^m = \begin{cases} 1 & (m=0), \\ 0 & (1 \le m \le K-1), \end{cases}$ The series above reduces to

$$
\begin{aligned}
\mathbb{E}[\widehat{E}_{\text{R}}] &= a_0 + \sum_{m=K}^{\infty} a_m \lambda^m \Big( \sum_{i=0}^{K} c_i s_i^m \Big) \\
&= E(0) + \sum_{m=K}^{\infty} a_m \lambda^m \Big( \sum_{i=0}^{K} c_i s_i^m \Big).
\end{aligned}
\tag{39}
$$

Taking absolute values and utilizing absolute convergence (Assumption 1) produces, for sufficiently small $\lambda$ as

$$\left|\mathbb{E}[\widehat{E}_{\mathrm{R}}] - E(0)\right| \leq \sum_{m=K}^{\infty} |a_m| \lambda^m \left|\sum_{i=0}^{K} c_i s_i^m\right|$$

$$\leq \left(\max_{m \geq K} \left|\sum_{i=0}^{K} c_i s_i^m\right|\right) \sum_{m=K}^{\infty} |a_m| \lambda^m. \tag{40}$$

There is a finite constant $C_{\mathrm{bias}}$ (depending on $\{a_m\}, \{s_i\}, K$) such that, for $\lambda < \rho$ (e.g. compare with a geometric series), the tail $\sum_{m=K}^{\infty} |a_m| \lambda^m$ is limited by a constant times $\lambda^K$ as a result of the power series's radius $\rho > 0$ as

$$\left|\mathbb{E}[\widehat{E}_{\mathrm{R}}] - E(0)\right| \leq C_{\mathrm{bias}} \lambda^K, \tag{41}$$

which completes the proof. $\qquad\square$

***Proof of Lemma 2*** *(Variance of the Richardson estimator).* According to Assumption 3, the empirical estimators $\widehat{E}(\lambda_i)$ are independent across nodes and satisfy $\mathrm{Var}[\widehat{E}(\lambda_i)] = \sigma_i^2/N_i$. Given that $\widehat{E}_{\mathrm{R}} = \sum_{i=0}^{K} c_i \widehat{E}(\lambda_i)$ and variance is quadratic and cumulative for independent factors,

$$\mathrm{Var}[\widehat{E}_{\mathrm{R}}] = \sum_{i=0}^{K} c_i^2 \, \mathrm{Var}[\widehat{E}(\lambda_i)]$$

$$= \sum_{i=0}^{K} c_i^2 \frac{\sigma_i^2}{N_i}. \tag{42}$$

The condition $\sigma_i^2 \leq \sigma^2$ produces

$$\mathrm{Var}[\widehat{E}_{\mathrm{R}}] \leq \sigma^2 \sum_{i=0}^{K} \frac{c_i^2}{N_i}. \tag{43}$$

If $N_i \equiv N$ for every $i$, then this reduces to

$$\mathrm{Var}[\widehat{E}_{\mathrm{R}}] = \frac{\sigma^2}{N} \sum_{i=0}^{K} c_i^2$$

$$= \frac{\sigma^2}{N} \Gamma_K^2, \tag{44}$$

where $\Gamma_K^2 = \sum_{i=0}^{K} c_i^2$. This completes the proof. $\qquad\square$

***Proof of Lemma 3*** *(Bounds on Richardson coefficients).* Consider the moment system: $\sum_{i=0}^{K} c_i s_i^m = \delta_{m0}$ for $m = 0, \ldots, K-1$. Assuming

$$V := \begin{bmatrix} 1 & 1 & \cdots & 1 \\ s_0 & s_1 & \cdots & s_K \\ \vdots & \vdots & \ddots & \vdots \\ s_0^{K-1} & s_1^{K-1} & \cdots & s_K^{K-1} \end{bmatrix} \in \mathbb{R}^{K \times (K+1)} \tag{45}$$

Consider the (transpose of the) Vandermonde sampling matrix on powers $0, \ldots, K-1$. In column form, one may conceive of the square Vandermonde on any given $K$ columns, but a simpler constructive path employs Lagrange basis polynomials.

Define the following Lagrange basis polynomials for the nodes $\{s_i\}_{i=0}^{K}$ as

$$L_i(s) = \prod_{\substack{j=0 \\ j \neq i}}^{K} \frac{s - s_j}{s_i - s_j}, \qquad i = 0, \ldots, K. \tag{46}$$

Each $L_i(s)$ is a polynomial of degree $K$ with $L_i(s_j) = \delta_{ij}$. The interpolation identity applies to any polynomial $p(s)$ with a degree of at most $K - 1$ as

$$p(s) \;=\; \sum_{i=0}^{K} p(s_i)\, L_i(s). \tag{47}$$

Analyzing at $s = 0$ gives

$$p(0) \;=\; \sum_{i=0}^{K} p(s_i)\, L_i(0). \tag{48}$$

Setting $p(s) = s^m$ for $m = 0, \ldots, K - 1$ results in the same restrictions as choosing $\sum_{i=0}^{K} c_i s_i^m = \delta_{m0}$ as

$$c_i \;=\; L_i(0), \qquad i = 0, \ldots, K. \tag{49}$$

Thus, the Richardson coefficients can be expressed in closed form as

$$
\begin{aligned}
c_i \;=\; L_i(0) &= \prod_{\substack{j=0 \\ j \neq i}}^{K} \frac{0 - s_j}{s_i - s_j} \\
&= \prod_{\substack{j=0 \\ j \neq i}}^{K} \frac{-s_j}{s_i - s_j}.
\end{aligned}
\tag{50}
$$

An appropriate expression can be obtained by taking absolute values as

$$|c_i| \;=\; \prod_{\substack{j=0 \\ j \neq i}}^{K} \frac{s_j}{|s_i - s_j|}. \tag{51}$$

Assume all nodes satisfy $1 \leq s_j \leq S$ (assumption in the lemma). Then we get the equation

$$
\begin{aligned}
|c_i| &\leq \prod_{\substack{j=0 \\ j \neq i}}^{K} \frac{S}{|s_i - s_j|} \\
&= S^K \prod_{\substack{j=0 \\ j \neq i}}^{K} \frac{1}{|s_i - s_j|}.
\end{aligned}
\tag{52}
$$

Consider $\delta := \min_{i \neq j} |s_i - s_j| > 0$, which denotes the smallest separation between different nodes. Then for every $i$ we have $\prod_{j \neq i} |s_i - s_j| \geq \delta^K$, and, therefore

$$|c_i| \leq S^K \, \delta^{-K} = (S/\delta)^K. \tag{53}$$

Given $\rho_{\mathrm{cond}} := S/\delta > 1$ and $C_{\mathrm{coeff}} := 1$ (or absorb mild constants into $C_{\mathrm{coeff}}$ for somewhat looser but more apparent constant); then we can describe

$$\max_i |c_i| \leq C_{\mathrm{coeff}} \, \rho_{\mathrm{cond}}^K. \tag{54}$$

Finally, as $\Gamma_K^2 = \sum_{i=0}^{K} c_i^2 \leq (K + 1) \max_i |c_i|^2$, we can describe it as

$$\Gamma_K^2 \leq (K + 1) C_{\mathrm{coeff}}^2 \rho_{\mathrm{cond}}^{2K}, \tag{55}$$

This is the exponential-in-$K$ bound that is being claimed. This concludes the proof. $\qquad\square$

**Proof of Theorem 1** (*Asymptotic MSE for fixed per-node shots*). Lemma 1 provides the bias bound as

$$\left(\mathbb{E}[\widehat{E}_{\mathrm{R}}] - E(0)\right)^2 \leq C_{\mathrm{bias}}^2 \lambda^{2K}. \tag{56}$$

Lemmas 2 and 3 assume $N_i \equiv N$ and $\sigma_i^2 \leq \sigma^2$,

$$\begin{aligned}
\mathrm{Var}(\widehat{E}_{\mathrm{R}}) &= \frac{\sigma^2}{N}\Gamma_K^2 \\
&\leq \frac{\sigma^2}{N}(K+1)C_{\mathrm{coeff}}^2 \rho_{\mathrm{cond}}^{2K}.
\end{aligned} \tag{57}$$

Summing bias$^2$ and variance yields

$$\mathrm{MSE}(\widehat{E}_{\mathrm{R}}) \leq C_{\mathrm{bias}}^2 \lambda^{2K} + \frac{\sigma^2}{N}(K+1)C_{\mathrm{coeff}}^2 \rho_{\mathrm{cond}}^{2K}, \tag{58}$$

which precisely matches the stated bound and concludes the proof. $\square$

**Proof of Theorem 2** (*MSE goes to zero with growing resources*). Using Theorem 1 (with $K = K(R)$ and per-node shots at least $\min_i N_i(R)$), we have, up to constants,

$$\mathrm{MSE}(\widehat{E}_{\mathrm{R}}) \lesssim \lambda^{2K(R)} + \frac{\rho_{\mathrm{cond}}^{2K(R)}}{\min_i N_i(R)}. \tag{59}$$

Hypothesis (i) states that $K(R) \to \infty$, which implies that the first term disappears if $\lambda < 1$. In the second term, condition (iii) assures that $\lambda\rho_{\mathrm{cond}} < 1$, indicating $\rho_{\mathrm{cond}}^{2K(R)}$ increases at most exponentially, but can be dominated by selecting $\min_i N_i(R)$ to grow sufficiently quickly.

Hypothesis (ii) states $\min_i N_i(R) \to \infty$, so the ratio $\rho_{\mathrm{cond}}^{2K(R)}/\min_i N_i(R)$ could potentially be made to fall to zero by a suitable joint growth schedule for $K(R)$ and $N_i(R)$ (for example, any schedule with $\log \min_i N_i(R) \gg 2K(R)\log \rho_{\mathrm{cond}}$). Concretely, because $K(R) \to \infty$, we can pick the growth of $\min_i N_i(R)$ so that the second term goes to 0. As $R$ approaches infinity, both terms vanish, and therefore $\mathrm{MSE} \to 0$. This proves the theorem. $\square$

**Summary of results and practical comments**

- **bias:** When $K+1$ nodes meet Richardson requirements, the residual bias scales as $\mathcal{O}(\lambda^K)$ (Lemma 1).
- **Variance:** The variance is magnified by the coefficient norm $\Gamma_K^2 = \sum_i c_i^2$, and scales as $\Gamma_K^2/N$ when shots per node are $N$ (Lemma 2).
- **Tradeoff/Resource Scaling:** For a fixed processing budget $R$, there is an optimal tradeoff between increasing $K$ and allocating more shots. However, if processing power can be increased so that both $K$ and per-node shots grow (and the node geometry is chosen to keep coefficient growth under control), then MSE can be driven to zero (Theorem 2).
- **Node geometry is important:** Lemma 3 highlights that coefficient growth (and hence variance amplification) depends heavily on how the $\{s_i\}$ are selected. Chebyshev-like selections can attenuate growth.

**Proof of Theorem 3** (*Robustness-transfer bound*). For simplicity, write $f_{\mathrm{noisy}} = \widehat{f}_{\mathrm{noisy}}$, $f_{\mathrm{clean}} = \widehat{f}_{\mathrm{clean}}$, and $\tilde{E} = \hat{E}(0)$. For every (x),

$$f_{\mathrm{noisy}}(x) - E^\star(x) = \left(f_{\mathrm{noisy}}(x) - f_{\mathrm{clean}}(x)\right) + \left(f_{\mathrm{clean}}(x) - \tilde{E}(x)\right) + \left(\tilde{E}(x) - E^\star(x)\right). \tag{60}$$

Taking expectation over $x$ and applying the basic inequality $(a+b+c)^2 \leq 3(a^2+b^2+c^2)$ produces

$$\begin{aligned}
\mathbb{E}_x\left[(f_{\mathrm{noisy}}(x) - E^\star(x))^2\right] \leq &3\,\mathbb{E}_x\left[(f_{\mathrm{noisy}}(x) - f_{\mathrm{clean}}(x))^2\right] + 3\,\mathbb{E}_x\left[(f_{\mathrm{clean}}(x) - \tilde{E}(x))^2\right] \\
&+ 3\,\mathbb{E}_x\left[(\tilde{E}(x) - E^\star(x))^2\right].
\end{aligned} \tag{61}$$

We bound each phrase to the right independently as

**(i) Deployment noise term.** Using Assumption 6 we get,

$$\mathbb{E}_x\big[(f_{\text{noisy}}(x) - f_{\text{clean}}(x))^2\big] \leq \sup_x |f_{\text{noisy}}(x) - f_{\text{clean}}(x)|^2 \leq \eta^2. \tag{62}$$

**(ii) Student approximation + generalization.** Add and subtract the empirical risk minimization behavior as

$$\mathbb{E}_x\big[(f_{\text{clean}}(x) - \tilde{E}(x))^2\big] \leq \underbrace{\inf_{f \in \mathcal{F}} \mathbb{E}_x\big[(f(x) - \tilde{E}(x))^2\big]}_{= \varepsilon_{\text{approx}}} +$$

$$\underbrace{\left(\mathbb{E}_x\big[(\widehat{f}(x) - \tilde{E}(x))^2\big] - \frac{1}{n}\sum_{i=1}^{n}(\widehat{f}(x_i) - \tilde{E}(x_i))^2\right)}_{\leq \mathcal{R}_n(\mathcal{F}, \delta)} \tag{63}$$

Thus, according to Assumption 5, the last inequality holds with probability at least $1 - \delta$. Therefore,

$$\mathbb{E}_x\big[(f_{\text{clean}}(x) - \tilde{E}(x))^2\big] \leq \varepsilon_{\text{approx}} + \mathcal{R}_n(\mathcal{F}, \delta). \tag{64}$$

**(iii) Teacher extrapolation error.** Using Assumption 4 we get

$$\mathbb{E}_x\big[(\tilde{E}(x) - E^\star(x))^2\big] = \varepsilon_T. \tag{65}$$

Combining the three boundaries and multiplying by the factor 3 from the original inequality results in

$$\mathbb{E}_x\big[(f_{\text{noisy}}(x) - E^\star(x))^2\big] \leq 3\eta^2 + 3(\varepsilon_{\text{approx}} + \mathcal{R}_n(\mathcal{F}, \delta)) + 3\varepsilon_T, \tag{66}$$

which is the same as equation 21. This completes the proof. $\qquad\square$

## A.3 Algorithm

**Noise-Aware Forward Pass and Loss Computation.** Algorithm 1 covers the execution of a parameterized quantum circuit $U(\boldsymbol{\theta})$ under a realistic noise model that incorporates $T_1$ (relaxation), $T_2$ (dephasing), and thermal photon noise. The output is a set of noisy expectation values and their corresponding losses, which are ideal for training noise-resilient QNNs.

---

**Algorithm 1** Noise-Aware Forward Pass and Loss Computation

---

**Require:** Parameterized quantum circuit $U(\boldsymbol{\theta})$, dataset $\mathcal{D} = \{(x_i, y_i)\}_{i=1}^N$, noise parameters $\{T_1, T_2, \Delta t, A, T_{\text{qb}}, T_{\text{ext}}\}$, shot count $S$
**Ensure:** Noise-aware loss value $\mathcal{L}$
  1: Initialize total loss $\mathcal{L} \leftarrow 0$
  2: **for** each $(x, y) \in \mathcal{D}$ **do**
  3:    Compute relaxation probability: $p_{T_1} \leftarrow 1 - e^{-\Delta t/T_1}$
  4:    Compute dephasing probability: $p_{T_2} \leftarrow 1 - e^{-\Delta t/T_2}$
  5:    Compute thermal noise: $n_{\text{noise}} \leftarrow \frac{A}{\exp(T_{\text{qb}}/T_{\text{ext}}) - 1}$
  6:    Apply $X$-rotation noise: $\theta_{T_1} \leftarrow \alpha \cdot p_{T_1}$
  7:    Apply $Z$-phase noise: $\delta_{T_2} \sim \mathcal{N}(0, \sigma^2)$ with $\sigma^2 \propto p_{T_2}$
  8:    Simulate noisy circuit $U(\boldsymbol{\theta})$ with applied $T_1, T_2$, and thermal noise
  9:    Estimate expectation: $\hat{y} \leftarrow \frac{1}{S}\sum_{s=1}^S \text{Measure}(U(\boldsymbol{\theta}), x)$
10:    Update loss: $\mathcal{L} \leftarrow \mathcal{L} + \|\hat{y} - y\|^2$
11: **end for**
12: $\mathcal{L} \leftarrow \mathcal{L}/|\mathcal{D}|$
13: **return** $\mathcal{L}$

---

We begin the simulation with a parameterized quantum circuit $U(\boldsymbol{\theta})$, a dataset $\mathcal{D}$, and a set of noise parameters representative of NISQ hardware, including $T_1$ (relaxation time), $T_2$ (dephasing time), gate duration $\Delta t$, thermal noise constants, and circuit-specific calibration factors. For each training

input $x \in \mathcal{D}$, we mimic the influence of amplitude damping by estimating the probability of a relaxation event using $p_{T_1} = 1 - e^{-\Delta t/T_1}$, which approximates to $\Delta t/T_1$ for tiny $\Delta t$. This model depicts the decrease of quantum amplitude toward the ground state (line 3). Similarly, we calculate the dephasing probability $p_{T_2} = 1 - e^{-\Delta t/T_2}$, which captures coherence loss in the phase domain. Both probabilities are critical in describing gate-level quantum noise using the Lindblad framework (line 4).

In line 5, we estimate $n_{\text{noise}}$ using the Bose-Einstein distribution: $n_{\text{noise}} = \frac{A}{\exp(T_{\text{qb}}/T_{\text{ext}})-1}$, where $A$ is a noise amplitude factor and $T_{\text{qb}}, T_{\text{ext}}$ denote qubit and environmental temperatures. To replicate the $T_1$ relaxation effect, a $X$-rotation gate with an angle of $\theta_{T_1} = \alpha p_{T_1}$ is applied to the circuit's affected qubits, where $\alpha$ is empirically calibrated to reflect physical relaxation behavior (line 6). To simulate dephasing caused by $T_2$ noise, stochastic phase rotations along the $Z$-axis are used, with a kick angle of $\delta_{T_2} \sim \mathcal{N}(0, \sigma^2)$, where $\sigma$ is determined from $p_{T_2}$. This Gaussian perturbation represents random phase variations in quantum development (line 7). The noisy quantum circuit is then operated and measured to yield the expectation value $\langle U(\boldsymbol{\theta}) \rangle_x$. This method uses several shots (repetitions) to estimate the average behavior under stochastic noise. For each input-output pair, we aggregate the squared error between the noisy model output and the goal $y_x$, which may be a ground-truth label (for supervised tasks) or a noise-mitigated soft target (for distillation).

Finally, we calculate the overall MSE loss across all samples as

$$\mathcal{L} = \frac{1}{|\mathcal{D}|} \sum_{x \in \mathcal{D}} \| \langle U(\boldsymbol{\theta}) \rangle_x - y_x \|^2 \,, \tag{67}$$

This is the objective function for gradient-based optimization of $\boldsymbol{\theta}$ in noisy QML settings.

**ZNE and Teacher Soft Label Generation.** Algorithm 2 describes how zero-noise extrapolation (ZNE) is used to provide noise-reducing expectation values that serve as soft supervision objectives for student quantum neural networks. It requires access to a variational quantum circuit $U(\boldsymbol{\theta})$, two noise scaling factors $\lambda_1$ and $\lambda_2$, and a dataset $\mathcal{D}$ of input samples. The folded circuits are run in noisy conditions, and the results are extrapolated back to a zero-noise limit using Richardson extrapolation.

---

**Algorithm 2** ZNE and Teacher Soft Label Generation

---

**Require:** Noisy quantum circuit $U(\boldsymbol{\theta})$, dataset $\mathcal{D} = \{x_i\}_{i=1}^N$, noise scaling factors $\lambda_1, \lambda_2$, shot count $S$
**Ensure:** Zero-noise extrapolated teacher outputs $\hat{E}(0)(x)$ for each $x \in \mathcal{D}$
 1: **for** each input $x \in \mathcal{D}$ **do**
 2:    Construct folded circuits for each scale:
 3:        $U_{\lambda_1} \leftarrow U \cdot (U^\dagger U)^{n_1}$                          // where $\lambda_1 = 1 + 2n_1$
 4:        $U_{\lambda_2} \leftarrow U \cdot (U^\dagger U)^{n_2}$                          // where $\lambda_2 = 1 + 2n_2$
 5:    Simulate $U_{\lambda_1}$ under noisy evolution and compute:
 6:        $E(\lambda_1) \leftarrow \frac{1}{S} \sum_{s=1}^S \text{Measure}(U_{\lambda_1}, x)$
 7:    Simulate $U_{\lambda_2}$ under noisy evolution and compute:
 8:        $E(\lambda_2) \leftarrow \frac{1}{S} \sum_{s=1}^S \text{Measure}(U_{\lambda_2}, x)$
 9:    Apply first-order Richardson extrapolation:
10:        $\hat{E}(0)(x) \leftarrow \frac{\lambda_2 E(\lambda_1) - \lambda_1 E(\lambda_2)}{\lambda_2 - \lambda_1}$
11:    Store $\hat{E}(0)(x)$ as soft target
12: **end for**
13: **return** $\{\hat{E}(0)(x)\}_{x \in \mathcal{D}}$

---

The procedure starts by accepting the dataset $\mathcal{D}$, scaling factors $\lambda_1, \lambda_2$, the noisy quantum circuit $U(\boldsymbol{\theta})$, and the number of measurement shots $S$ per circuit execution. To simulate amplified noise levels, global unitary folding is used to modify the original circuit $U(\boldsymbol{\theta})$ for each input $x \in \mathcal{D}$. In particular, $\lambda_k = 1 + 2n_k$ for $k = 1, 2$, and two folded versions $U_{\lambda_1}$ and $U_{\lambda_2}$ are created using $n_1$ and $n_2$ layers of folding.

To get a noisy expectation value estimate $E(\lambda_1)$, the circuit $U_{\lambda_1}$ is run $S$ times in a noisy simulator or backend (line 3). For $U_{\lambda_2}$, the same process is performed, yielding $E(\lambda_2)$ (line 4). The noise-

---

**Algorithm 3** Student Training with Distillation Loss

---

**Require:** Distilled dataset $\mathcal{D} = \{(x_i, \hat{E}(0)_i)\}_{i=1}^N$ from ZNE-trained teacher
    Student QNN $V(\phi)$ with $n_{\text{stud}}$ qubits and shallow depth
    Epochs $T$, batch size $B$, learning rate $\eta$
**Ensure:** Trained student parameters $\phi^*$
  1: Initialize student QNN $V(\phi)$ with random parameters
  2: **for** epoch $= 1$ to $T$ **do**
  3:    **for** each batch $\mathcal{B} \subset \mathcal{D}$ of size $B$ **do**
  4:       Compute noisy student predictions $\langle V(\phi) \rangle_{x_i}$ for all $x_i \in \mathcal{B}$
  5:       Evaluate loss: $\mathcal{L}_{\text{exp}} = \frac{1}{|\mathcal{B}|} \sum_{(x_i, \hat{E}(0)_i) \in \mathcal{B}} \left( \langle V(\phi) \rangle_{x_i} - \hat{E}(0)_i \right)^2$
  6:       Estimate gradient $\nabla_\phi \mathcal{L}_{\text{exp}}$ via parameter-shift rule
  7:       Update parameters $\phi \leftarrow \phi - \eta \cdot \nabla_\phi \mathcal{L}_{\text{exp}}$ using Adam
  8:    **end for**
  9:    **if** validation loss does not improve for patience threshold **then**
10:       **break** {Early stopping}
11:    **end if**
12: **end for**
13: **return** $\phi^*$

---

free observable $\hat{E}(0)$ for the current input is calculated using the following closed-form expression after applying first-order Richardson extrapolation to these two estimates (line 10). The teacher's supervision target, the value $\hat{E}(0)$, is then saved as a soft label linked to the input $x$ (line 11). The algorithm returns the whole set of extrapolated outputs $\{\hat{E}(0)(x)\}_{x \in \mathcal{D}}$ for use in student distillation training after iterating over the entire dataset $\mathcal{D}$ (line 13).

**Student Training with Distillation Loss.** Algorithm 3 describes the distillation phase, in which a compact quantum student network learns to approximate the outputs of a noise-mitigating teacher. The input is a zero-noise extrapolation (ZNE)-generated distilled dataset $\{(x_i, \hat{E}(0)_i)\}_{i=1}^N$, where each $\hat{E}(0)_i$ is a noise-free soft label linked to input $x_i$. In contrast to the teacher, a shallow student circuit $V(\phi)$ is initiated with fewer parameters and less depth. The student uses $n_{\text{stud}} < n_{\text{teach}}$ qubits, usually with two entangling layers, but reuses the same data encoding as the teacher.

The student network is iteratively trained for $T$ epochs by the method, which minimizes the $\ell_2$ loss between the teacher's soft goal $\hat{E}(0)_i$ (line 4) and its projected expectation value $\langle V(\phi) \rangle_{x_i}$. To replicate realistic inference conditions, each training batch is run under native noise after being sampled from the distilled dataset $\mathcal{D}$ (line 5). The parameter-shift rule, a differentiable quantum gradient estimator (line 6), is used to calculate the gradient of the loss with regard to parameters $\phi$. The Adam optimizer is used to update the student parameters with batch size 32 (line 7) and learning rate $\eta = 0.02$. Early halting is initiated based on validation loss calculated across a 1,024-shot subset to avoid overfitting to stochastic shot noise (line 10). The trained student model $V(\phi^*)$ is kept for deployment following convergence. In contrast to the teacher, it may be run on actual NISQ devices with high fidelity and low overhead and doesn't require runtime mitigation.

**Full Training Pipeline.** Algorithm 4 describes the entire training procedure for our proposed ZNE-guided knowledge distillation framework. Zero-Noise Extrapolation (ZNE) is used to create a noise-aware teacher model at the start of the pipeline.

The teacher QNN $U(\theta)$ is run several times for each data point $x_i \in \mathcal{D}$ (line 2), with varying noise scaling factors $\lambda_k$ (line 3). Global gate folding (line 4) is used to introduce these noise levels, changing each $U$ to $U_{\lambda_k} = U(U^\dagger U)^n$ in order to imitate stronger noise. A noise simulator is used to determine the expectation value $E_{\lambda_k}(x_i)$ (line 5) for each scaled circuit. The mitigated zero-noise estimate $\hat{E}(0)(x_i)$, which acts as a soft label, is then obtained by first-order Richardson extrapolation (line 7) using these values. Each input is paired with its matching $\hat{E}(0)$ to create the distilled dataset $\mathcal{D}_{\text{soft}}$ (line 8).

The second step involves initializing (line 10) the student QNN $V(\phi)$ and training it to replicate the teacher's outputs without implementing runtime noise mitigation. The student computes the

---

**Algorithm 4** Full Training Pipeline: ZNE-Guided Knowledge Distillation

---

**Require:** Input dataset $\mathcal{D} = \{x_i\}_{i=1}^N$
  Teacher QNN $U(\boldsymbol{\theta})$, student QNN $V(\boldsymbol{\phi})$
  Noise levels $\{\lambda_k\}$ for ZNE (e.g., $\lambda \in \{1, 3, 5\}$)
  Epochs $T$, batch size $B$, learning rate $\eta$
**Ensure:** Trained student parameters $\phi^*$
 1: **// Stage 1: ZNE-Based Teacher Training**
 2: **for** each input $x_i \in \mathcal{D}$ **do**
 3:   **for** each noise scale $\lambda_k$ **do**
 4:     Apply global folding: $U_{\lambda_k} = U \cdot (U^\dagger U)^n$
 5:     Execute $U_{\lambda_k}$ under simulated noise and store expectation $E_{\lambda_k}(x_i)$
 6:   **end for**
 7:   Extrapolate $\hat{E}(0)(x_i)$ using Richardson method
 8: **end for**
 9: Construct distilled dataset $\mathcal{D}_{\text{soft}} = \{(x_i, \hat{E}(0)(x_i))\}_{i=1}^N$
10: **// Stage 2: Student Distillation Training**
11: Initialize $V(\boldsymbol{\phi})$ with random parameters
12: **for** epoch $= 1$ to $T$ **do**
13:   **for** each batch $\mathcal{B} \subset \mathcal{D}_{\text{soft}}$ of size $B$ **do**
14:     Compute predictions $\langle V(\boldsymbol{\phi})\rangle_{x_i}$ for all $x_i \in \mathcal{B}$
15:     Compute distillation loss:

$$\mathcal{L}_{\text{exp}} = \frac{1}{|\mathcal{B}|} \sum_{(x_i, \hat{E}(0)(x_i)) \in \mathcal{B}} \left( \langle V(\boldsymbol{\phi})\rangle_{x_i} - \hat{E}(0)(x_i) \right)^2$$

16:     Estimate gradients via parameter-shift rule
17:     Update $\boldsymbol{\phi} \leftarrow \boldsymbol{\phi} - \eta \cdot \nabla_{\boldsymbol{\phi}} \mathcal{L}_{\text{exp}}$ using Adam
18:   **end for**
19:   **if** validation loss stagnates **then**
20:     **break** {Early stopping}
21:   **end if**
22: **end for**
23: **return** $\phi^*$

---

Table 7: Comparison of ZNKD with prior quantum distillation and error-mitigation methods.

| Method | Replaces Extrapolation | Handles High Noise | Type of Distillation/Transfer | Objective |
|---|---|---|---|---|
| Classical-to-Quantum KD Tian et al. (2025); Alam et al. (2023); Hasan & Mahdy (2023) | No | Moderate | Prediction/Feature KD | Knowledge transfer |
| Variational Loss Shaping Cerezo et al. (2021) | No | Limited | Loss-level modification | Optimization stability |
| Reciprocal/Network KD Gou et al. (2024) | No | Limited | Circuit/Ansatz compression | Depth reduction |
| Hybrid Embedding Transfer Wang et al. (2025); Li et al. (2024) | No | Moderate | Embedding reuse | Improve generalization |
| Traditional ZNE | Yes | Poor | None | Noise extrapolation |
| **ZNKD (Ours)** | **Yes** | **High** | **Prediction KD** | **Noise-resistant learning** |

forward pass to get the native-noise expectation value $\langle V(\phi) \rangle_{x_i}$ for each $x_i$ in the batch (line 14), after processing data in mini-batches of size $B$ over several epochs (line 12). The $\ell_2$ regression error between the student's output and the teacher's noise-free label (line 15) is used to calculate the loss. The parameter-shift rule (line 16) is used to estimate gradients, and the Adam optimizer (line 17) is used to update model parameters.

Based on the validation loss, early stopping is used to prevent overfitting to stochastic noise (lines 19–21). The final student parameters $\phi^*$ are returned upon convergence (line 23). This two-stage technique decouples expensive error prevention from inference, enabling practical, noise-resilient quantum learning on NISQ devices.

### A.4 COMPARISON WITH OTHER APPROACHES

Table 7 compares ZNKD to other approaches that incorporate distillation, transfer learning, or error mitigation in quantum machine learning. Classical-to-quantum knowledge distillation methods Tian et al. (2025); Alam et al. (2023); Hasan & Mahdy (2023) transfer predictive ability or feature representations from a classical teacher to a quantum student. However, they do not modify or replace the extrapolation phase, providing only moderate robustness to hardware noise. Variational loss-shaping approaches, like Cerezo et al. Cerezo et al. (2021), improve robustness by changing the optimization landscape. However, they still need adequate measurement statistics and suffer under larger device noise. Reciprocal and network-based distillation Gou et al. (2024) compress quantum circuits by ansatz reduction, improving depth and execution time while ignoring the reliability of noisy expectation values. Hybrid embedding-transfer techniques Wang et al. (2025); Li et al. (2024) reuse representations across models to increase generalization, but do not minimize noise buildup, which restricts performance on near-term hardware. Traditional zero-noise extrapolation is sensitive to sampling fluctuations and becomes unstable as noise intensity increases. In contrast, ZNKD completely eliminates the extrapolation phase and learns a noise-resistant decision boundary from a teacher model, allowing it to retain high predicted accuracy even in noisy environments where other algorithms fail. This contrast emphasizes ZNKD's unique position as a distillation-based error mitigation approach that prioritizes robustness above depth reduction, feature reuse, or loss shaping.

### A.5 EXTENSIVE EXPERIMENTAL DETAILS

#### A.5.1 HARDWARE AND ENVIRONMENT

IBM's `Qiskit-Aer` simulator was used for all simulations, and it was equipped with specially designed noise models that were indicative of the near-term superconducting quantum hardware. In particular, we use the `thermal_relaxation_noise` module to simulate $T_1$ relaxation and $T_2$ dephasing processes. It is set up with representative coherence times ($T_1 = 50\,\mu s$, $T_2 = 70\,\mu s$) and gate durations (e.g., $u_3$: 200 ns, CNOT: 400 ns), which closely resemble mid-scale IBMQ devices such as `ibmq_manila` or `ibmq_jakarta`.

A high-performance server with two NVIDIA RTX 3090 GPUs and 128 GB of RAM was used for all tests in order to parallelize the execution of noisy circuits. The software stack consists of `Qiskit` 1.0.2, `Qiskit-Aer` 0.13.1, and Python 3.10. To correctly mimic decoherence effects, which is essential for assessing noise-aware training and mitigation performance, we make use of the

density_matrix_simulator backend. We can precisely regulate noise injection and circuit repetition using this setup, allowing for a reliable assessment of ZNE and distillation accuracy under practical device limitations.

### A.5.2 QUANTUM NOISE MODELS AND CALIBRATION DETAILS

All simulations use time-correlated noise models compatible with technology from the NISQ period to simulate realistic quantum phenomena. In particular, we use thermal relaxation channels to describe phase damping ($T_2$) and amplitude damping ($T_1$) faults. Qiskit Aer's thermal_relaxation_noise function is used to instantiate these channels.

**Parameter Settings.** Noise parameters are derived from typical calibration values reported by IBM Quantum systems:

- $T_1 = 50\ \mu\text{s}$ (relaxation time),
- $T_2 = 70\ \mu\text{s}$ (dephasing time),
- Gate time $\tau_g = 300$ ns for single-qubit gates, and 450 ns for two-qubit gates.

**Dynamic Noise Scaling.** We use gate folding techniques to raise the effective noise level in order to facilitate zero-noise extrapolation (ZNE). Scaling factors $\lambda \in \{1.0, 1.5, 2.0\}$ are chosen to strike a compromise between resource overhead and mitigation fidelity. According to equation 3, these equate to global folding for $n = 0, 1$, and 2.

**Shot Noise and Backend Emulation.** Every quantum circuit is sampled using 1,024 shots, unless otherwise noted. Gate noise and shot noise are both simulated. In order to stay within realistic simulation budgets, we limit the number of qubits each experiment uses while using Aer's density matrix simulator with full noise modeling.

**Hardware Noise Profiles.** Hardware calibration snapshots are extracted using IBMQBackend.properties() when applicable. This guarantees transferability from simulation to hardware-executed settings by enabling us to verify against actual noise characteristics.

### A.5.3 SOFTWARE AND PACKAGE VERSIONS

Python 3.10.6 was used for all experiments on a Linux-based system with an NVIDIA RTX 3090 GPU and 128 GB of RAM. The Qiskit framework, version qiskit==1.0.2, and the qiskit-aer==0.13.1 simulator backend for realistic noise modeling were used to run quantum circuit simulations.

**Quantum Libraries.** To enable noise-aware simulation and transpilation, we used the following packages:

- qiskit-machine-learning==0.7.0 for variational classifiers and quantum kernels,
- qiskit-ibm-runtime==0.22.0 for hardware calibration access,
- qiskit-algorithms==0.3.1 for quantum optimization primitives.

Custom extensions were used to create circuit folding and extrapolation utilities, adhering to the design outlined in Giurgica-Tiron et al. (2020); Majumdar et al. (2023); Pelofske et al. (2024).

**Classical Backends.** Classical baselines and hybrid training loops were implemented using:

- PyTorch==2.2.2 for optimizer and autodiff routines,
- scikit-learn==1.4.2 for auxiliary evaluation metrics and data preprocessing,
- numpy==1.26.4 and matplotlib==3.8.3 for numerical computations and visualizations.

Using deterministic simulation settings and fixed random seeds, every experiment could be replicated. Our supplementary materials and GitHub repository contain a `requirements.txt` file that captures the entire environment.

### A.5.4 TRAINING SETTINGS

We outline the settings and hyperparameters that were utilized to train the student and teacher quantum models. In order to balance expressivity, convergence speed, and compatibility with noisy quantum devices, these factors—which include circuit depth, optimizer configuration, batch sizes, and shot counts—were chosen.

**Teacher Configuration.** The teacher quantum neural network (QNN) utilizes the CZ entanglement technique in a linear topology with a variational ansatz with $L = 4$ entangling layers. Angle encoding is used on all qubits to encode input information. With a learning rate of $0.01$, batch size of $64$, and $150$ epochs, the Adam optimizer is used for training. The projected zero-noise targets derived via Richardson extrapolation are incorporated into loss minimization. The parameter-shift rule is used to estimate all gradients.

**Student Configuration.** With fewer qubits and entangling layers ($L = 2$), the student QNN functions under native hardware noise. To improve trainability under constrained coherence times, the design employs fewer variational parameters while mirroring the teacher in the encoding scheme. For consistency, the batch size and optimizer are the same. Depending on the task, the loss is calculated using either Kullback–Leibler (KL) divergence or mean squared error (MSE), which corresponds to the teacher-provided goal representation.

**Shared Settings.** $1,024$ measurement shots are used for each input in every quantum circuit operation. Generalization is monitored using a $20\%$ validation split, and early halting is initiated if validation loss does not improve for $10$ successive epochs. Qiskit's Aer simulator is used to implement noise-aware training using readout and injected $T_1/T_2$ noise. For reproducibility, a fixed random seed is used in every experiment.

### A.5.5 MODEL ARCHITECTURES

Throughout our experiments, we describe the architecture of the teacher and student quantum models. Although their encoding mechanisms are comparable, the student is purposefully small to guarantee that it is suitable for deployment in the NISQ period.

**Teacher QNN.** A high-capacity variational quantum circuit with $L = 4$ entangling layers is used to construct the teacher model. Single-qubit rotation gates (RY, RZ) and controlled-Z (CZ) gates are included in each layer in a linear topology. Angle encoding is used to encode input data, with one feature per qubit. We use data re-uploading and padding for tasks that require higher-dimensional input. The total number of trainable parameters is roughly $3nL$ and is dependent on the number of qubits $n$.

**Student QNN.** The student circuit has fewer qubits ($n_{\text{stud}} < n_{\text{teach}}$) and $L = 2$ entangling layers, which are purposefully shallower and resource-efficient. It facilitates knowledge transfer by using the same gate types and encoding scheme as the teacher. However, the reduced depth and parameter count result in a smaller circuit depth, more aligned with coherence restrictions on real hardware.

**Circuit Comparison.** Depending on the dataset, the teacher usually utilizes 6–8 qubits, while the pupil uses 3–5. Standard Qiskit execution backends can be used with both architectures. Better expressivity is empirically attained by the teacher circuit, while robustness is passed down to the student through distillation.

### A.5.6 OPTIMIZATION DETAILS

We describe the hyperparameters and optimization techniques used to train the teacher and student quantum neural networks (QNNs). Qiskit and PyTorch are used to implement all optimization procedures in a hybrid classical-quantum loop.

**Gradient Evaluation.** By performing shifted versions of the quantum circuit, the parameter-shift rule allows for precise derivative estimates with respect to variational angles and is used to compute gradients for quantum circuit parameters. Under noise, two more forward passes are needed for each gradient component.

**Optimizer Configuration.** For teacher and student training, we employ the Adam optimizer, which has learning rates of $\eta = 0.02$, $\beta_1 = 0.9$, and $\beta_2 = 0.999$. These parameters are consistent across all datasets and were selected based on empirical stability.

**Convergence and Regularization.** We use early stopping based on validation loss with a 10-epoch patience to prevent overfitting to noise or volatility in measurement results. All experiments are repeated using a fixed random seed for reproducibility, and training is limited to 150 epochs.

**Shot Configuration.** 1024 measurement shots are used for each quantum circuit operation. Each noise level requires several folding executions for ZNE teacher training. Only native noise is used for student training, which drastically lowers runtime costs.

## A.6 BASELINE COMPARISON

To ensure a fair and complete evaluation, we compare our method to a wide range of quantum noise mitigation and compression strategies using comparable resource budgets and realistic noise models. The chosen baselines include noise-aware training, circuit compression, knowledge transfer, and error prevention.

- **Noise-Aware VQCs** Chen et al. (2024): Variational quantum circuits are taught by explicitly using hardware noise models in the optimization loop. This technique allows circuits to learn noise-resilient parameter values, but it may lead to suboptimal minima due to the stochasticity of device-level noise.
- **Pruned QNNs** Afane et al. (2025): Removing redundant gates or parameters from quantum neural networks reduces circuit depth and total error accumulation. While pruning improves efficiency and reliability, it can limit expressivity and learning capacity, resulting in only modest gains in predictive accuracy.
- **Classical-to-Quantum Knowledge Distillation** Abbas et al. (2021): A large classical teacher network oversees a smaller quantum student model, allowing knowledge to be transferred to noisy quantum hardware. This strategy enhances generalization and resilience while introducing additional training costs and a partial reliance on classical resources.
- **Standalone Zero-Noise Extrapolation (ZNE)** Temme et al. (2017): Circuits are run with scaled noise levels and extrapolated to the zero-noise limit. Although ZNE is useful in many situations, it is extremely sensitive to temporal noise variations and requires several circuit assessments, which increases runtime overhead.
- **Probabilistic Error Cancellation (PEC)** Van Den Berg et al. (2023); Gupta et al. (2024): A quasi-probability sampling strategy that statistically inverts noisy quantum channels in order to retrieve optimal results. PEC offers effective error suppression, but its exponential sampling cost restricts scalability for large applications.
- **Best-Folding ZNE Configurations** Pelofske et al. (2024): An improved ZNE plugin that uses adaptive folding algorithms (gate folding, layer folding) to balance fidelity and runtime cost. While this setup improves dependability more than a standalone ZNE, it still has significant measurement and sampling overhead.
- **Measurement Error Mitigation (MEM)** Khan et al. (2024): A calibration-based post-processing approach that estimates and corrects readout errors using a pre-defined confusion matrix. MEM minimizes traditional post-processing bias but does not address gate or decoherence issues.
- **Clifford Data Regression (CDR)** Liao et al. (2024): A regression-based extrapolation approach for fitting noisy observables into classically simulable Clifford circuits. CDR improves estimation accuracy with small sample needs, but its utility is restricted beyond Clifford-like structures.

- **Dynamical Decoupling (DD)** Tong et al. (2025): A hardware-level control sequence that controls decoherence by using custom pulse sequences. While effective for low-frequency noise, DD requires more gate operations and may interfere with circuit-level optimization.

- **Hardware-Efficient Ansätze** Leone et al. (2024): Parameterized quantum circuits are designed to meet device topology and gate restrictions, lowering compilation overhead and decreasing gate errors. However, their limited expressivity could hinder learning on complicated datasets.

## B  EXTENDED EXPERIMENTAL RESULTS

**Teacher QNN Architecture and Training.**

The teacher is a depth-$L_{\text{teach}}$ variational circuit using $n_{\text{teach}}$ qubits. Each layer includes i) rotations $R_y(\kappa x_j)$ on all qubits for data re-uploading, ii) parameterized single-qubit gates $R_y(\theta_{l,q})R_z(\phi_{l,q})$, and iii) an entangling CNOT ladder connecting qubit $q$ to $q+1$.

We set $L_{\text{teach}} = 6$ and $n_{\text{teach}} = 8$, resulting in 96 training parameters. The mean squared error (MSE) serves as the cost function; Adam optimizer is used for gradients using the parameter-shift rule ($\eta = 0.01$, 100 epochs, batch 32).

During training, we have used three ZNE scaling factors ($\lambda_1$=1, $\lambda_1$=3, and $\lambda_2$=5) obtained from global folding (Eq. 3) and estimated $\hat{E}(0)$ using Richardson extrapolation. The extended figures become soft targets for students.

**Student QNN and Distillation Objective.** The student uses the *same* but a shallower, $n_{\text{stud}}$=4-qubit, $L_{\text{stud}}$=2-layer approach with 16 trainable parameters, making it only ~17% of the size of the teacher. We minimize the distillation loss as specified in equation 8.

They apply Adam ($\eta = 0.02$, 50 epochs, batch 32) optimizer. No folding or extrapolation takes place for students during training or testing.

**Noise Model and Qiskit-Aer Simulation.** The circuits are run using the `AerSimulator` (density-matrix mode) from `qiskit-aer 0.13.1`. Our proprietary noise model includes amplitude-shifting with $T_1 = 60\,\mu s$, phase-damping with $T_2 = 70\,\mu s$, two-qubit depolarizing error $p_{2q} = 2.5 \times 10^{-3}$, and read-out bit-flip error $p_{\text{ro}} = 0.015$. The shot count is regulated at 1024 per circuit, and the seeds are fixed for reproducibility.

**Evaluation Metrics and Baselines.** Performance is evaluated using the MSE between model predictions and ground-truth objectives for the regression task using

$$\text{MSE} = \frac{1}{|\mathcal{D}_{\text{test}}|} \sum_{(x,y)\in\mathcal{D}_{\text{test}}} \left(\hat{y}(x) - y\right)^2. \tag{68}$$

where $\mathcal{D}_{\text{test}}$ denotes the held-out test set; $|\mathcal{D}_{\text{test}}|$ represents its cardinality (number of samples); $(x,y)$, a goal of ground-truth $y$ and an input sample $x$; $\hat{y}(x)$, the prediction of the model for input $x$. For the classification of the downstream application, we additionally address *accuracy* and *MSE loss*.

In order to measure noise robustness, we record the decrease in performance $\Delta\text{MSE}(\epsilon)$ after each two-qubit operation during inference and inject an additional depolarizing channel of strength $\epsilon \in [0, 0.01]$. A lower $\Delta$ value signifies more internal resilience.

We conducted the following simulation studies in Appendix B using the *Fashion-MNIST* dataset, as they are compact enough for circuit-level ablations and noise experiments.

### B.1  SIMULATION RESULTS

**Dynamic-Noise Settings.** We define three dynamic noise regimes to evaluate robustness under realistic hardware conditions: *Low*, *Medium*, and *High* in Table 8. The parameters for each procedure include time-dependent thermal occupancy $n_{\text{noise}}(t)$, relaxation time $T_1$, dephasing time $T_2$, and a two-qubit depolarization error rate $p_{2q}$. The thermal model in equation 24 describes a gradual oscillation of all parameters in the circuit execution window $t \in [0, T]$.

| | $T_1$ ($\mu$s) | $T_2$ ($\mu$s) | $p_{2q}$ | $n_{\mathrm{noise}}(t) = \bar{n} + \Delta n \sin\left(\frac{2\pi t}{T}\right)$ |
|---|---|---|---|---|
| Low | 100 | 120 | $1.0 \times 10^{-4}$ | $\bar{n} = 0.02, \; \Delta n = 0.01$ |
| Medium | 60 | 80 | $1.0 \times 10^{-3}$ | $\bar{n} = 0.05, \; \Delta n = 0.02$ |
| High | 30 | 40 | $2.5 \times 10^{-3}$ | $\bar{n} = 0.10, \; \Delta n = 0.04$ |

Table 8: Dynamic-noise regimes used in simulation. $T_1/T_2$ sets the Lindblad rates; $p_{2q}$ is a static depolarising error applied to each CNOT; $n_{\mathrm{noise}}(t)$ modulates thermal occupancy over circuit time $T$ (here $T = 1$ ms).

| Dataset | Noiseless | Low | Medium | High |
|---|---|---|---|---|
| Fashion-MNIST | 0.25 | 0.30 | 0.45 | 0.65 |
| AG News | 0.18 | 0.22 | 0.33 | 0.55 |
| Wine Quality | 0.39 | 0.43 | 0.50 | 0.62 |
| UrbanSound8K | 0.36 | 0.42 | 0.55 | 0.78 |

Table 9: The loss of teachers under increasing dynamic noise levels is represented by *MSE* for classification and *MSE* for regression.

Table 9 shows the *test loss* of the ZNE-trained teacher circuit under four conditions: a noiseless simulator, low, medium, and high dynamic noise levels. We employ the *MSE loss* for the classification and regression datasets. It demonstrates that loss rises with noise severity, indicating that dynamic decoherence impairs performance even after ZNE. The rise is moderate under low noise but larger under high noise, notably for image and audio data. This highlights the need for noise mitigation while working in practical conditions.

**Effectiveness of ZNE.** Table 10 shows the effect of ZNE on high-noise situations. As the noise-scaling factor $\lambda$ increases, loss lowers consistently across all datasets, indicating ZNE's capacity to recover increasingly reliable outputs. The best results are achieved at $\lambda = 5$, especially for more complex datasets like UrbanSound8K and Fashion-MNIST, which confirm ZNE's effectiveness in mitigating quantum noise in practical workloads. Table 11 demonstrates that raising the ZNE folding factor $\lambda$ leads to significant increases in gate count, circuit depth, and execution time for FashionM-NIST data. With $\lambda$=5, circuits become 9x longer than their original versions, taking over 1 second each inference. These findings emphasize the computational tradeoff introduced by ZNE and the significance of cost-effective mitigation measures.

**Effectiveness of Knowledge Distillation.** Table 12 shows the effect of knowledge distillation by a ZNE-trained teacher. Across all datasets, student models trained without KD lose substantially more due to noise. In contrast, ZNE-based distillation increases resilience, resulting in 6-12% lower loss values. This demonstrates the effectiveness of distillation in transferring resilience to low-capacity quantum models.

B.2  RICHARDSON EXTRAPOLATION IN ZNE

The Richardson extrapolation procedure used in zero-noise estimate is shown in Figure 2. We use global gate folding to record the circuit expectation values at scaled noise levels $\lambda = \{1, 3, 5\}$ for

| Dataset | No ZNE | $\lambda = 2$ | $\lambda = 3$ | $\lambda = 5$ |
|---|---|---|---|---|
| Fashion-MNIST | 0.65 | 0.54 | 0.47 | **0.40** |
| AG News | 0.55 | 0.44 | 0.36 | **0.30** |
| Wine Quality | 0.62 | 0.51 | 0.46 | **0.39** |
| UrbanSound8K | 0.78 | 0.65 | 0.58 | **0.49** |

Table 10: ZNE effectiveness under high noise. Each row shows the teacher model's loss without mitigation and with ZNE at different noise-scaling factors $\lambda$. Larger $\lambda$ values yield more accurate extrapolation to the zero-noise regime. All datasets benefit from ZNE, with greater improvement observed on high-dimensional inputs like images and audio.

| $\lambda$ | Gate Count | Circuit Depth | Execution Time (ms) |
|---|---|---|---|
| 1 (No ZNE) | 128 | 64 | 121 |
| 2 {1,3} | 384 | 192 | 368 |
| 3 {1,3,5} | 640 | 320 | 610 |
| 5 {1,3,5,7,9} | 1152 | 576 | 1088 |

Table 11: Computational cost of circuit folding with different ZNE noise-scaling factors $\lambda$. All values are measured per circuit on Qiskit Aer with 1024 shots. Gate count and circuit depth increase proportionally with $\lambda$, significantly impacting runtime. All values are averaged across 10 runs of a teacher circuit trained on Fashion-MNIST dataset.

| Dataset | Baseline Student Loss | KD Student Loss | Improvement |
|---|---|---|---|
| Fashion-MNIST | 0.61 | 0.49 | **–0.12** |
| AG News | 0.45 | 0.36 | **–0.09** |
| Wine Quality | 0.58 | 0.52 | **–0.06** |
| UrbanSound8K | 0.66 | 0.59 | **–0.07** |

Table 12: Loss comparison of student QNNs trained directly (baseline) vs. distilled from ZNE-enhanced teachers (KD). Knowledge distillation consistently improves performance under native noise.

| Dataset | Teacher Loss | Student Loss | Compression Ratio | Loss Gap |
|---|---|---|---|---|
| Fashion-MNIST | 0.45 | 0.49 | 8 : 3 | +0.04 |
| AG News | 0.33 | 0.36 | 6 : 2 | +0.03 |
| Wine Quality | 0.50 | 0.52 | 6 : 2 | +0.02 |
| UrbanSound8K | 0.55 | 0.59 | 8 : 3 | +0.04 |

Table 13: Student loss after distillation from a ZNE-trained teacher. The student model uses fewer qubits and layers (compression ratio shown), yet retains comparable accuracy with minor loss degradation.

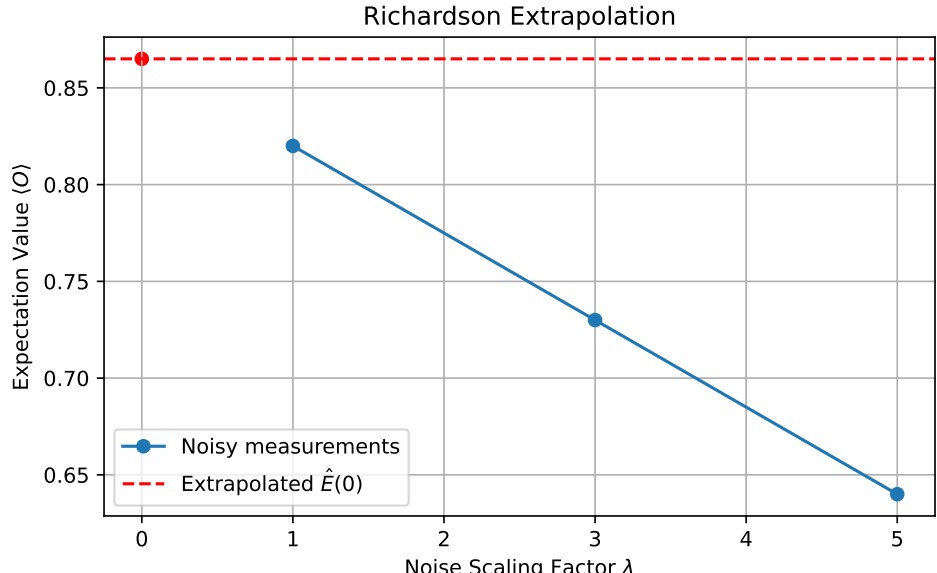

Figure 2: Visualization of expectation values at noise scales $\lambda = 1, 3, 5$ with first-order Richardson extrapolation. Points denote observed values under increasing folding, while the extrapolated zero-noise value is shown at $\lambda = 0$.

a representative batch of inputs from the FashionMNIST dataset. Assuming that these data roughly follow a low-degree polynomial curve with respect to $\lambda$, we can extrapolate back to $\lambda = 0$ to estimate the zero-noise value $\hat{E}(0)$.

As demonstrated, the extrapolated value consistently aligns across various inputs, and the folded outputs follow a smooth trajectory. This illustrates how gate-induced errors can be successfully eliminated by ZNE using Richardson extrapolation without creating instability. Crucially, this extrapolated output encodes a denoised supervisory signal that preserves semantic fidelity while reducing variance brought on by device-level noise, making it a soft target for student training. Because the curve-fitting stage is completely post-measurement and computationally cheap, it can be used in offline or quantum cloud environments.

### B.3 GATE ERROR SENSITIVITY ANALYSIS

The sensitivity of quantum model fidelity to changes in two-qubit gate errors, namely controlled-Z (CZ) gate noise, is investigated in this experiment. Our distilled student model and a baseline QNN are both simulated throughout a range of CZ error rates, from 0.005 to 0.08. The overlap between expected and noise-free results is used to calculate fidelity for each model, which is assessed using thermal relaxation and measurement noise under Qiskit Aer. The distilled student is optimized under native noise only and is given soft labels by a noise-mitigated teacher educated using zero-noise extrapolation (ZNE).

Across all tested error levels, Figure 3 demonstrates that the distilled model consistently performs better than the non-distilled baseline. At the highest error level, the distilled student maintains a considerable fidelity margin—up to 5.4%—in comparison to the baseline, despite the fact that both models deteriorate as gate noise increases. Even in high-noise conditions, which are typical of NISQ devices, this demonstrates that knowledge distillation not only compresses the model but also transfers robustness, enabling more stable quantum predictions.

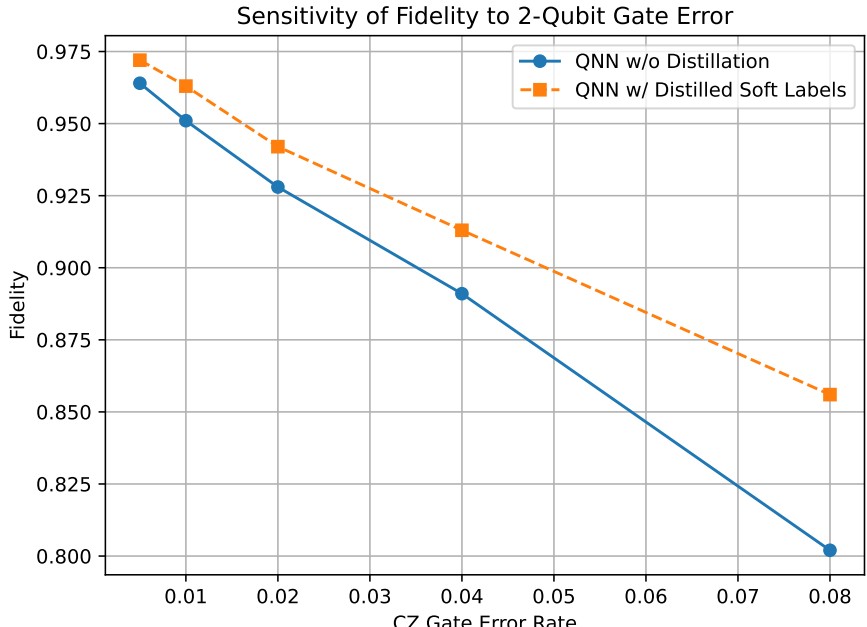

Figure 3: Sensitivity analysis showing how fidelity degrades with increasing CZ gate error rates for both the baseline QNN and the distilled student. The distilled model consistently maintains higher fidelity across error scales.

### B.4 TEMPORAL DRIFT IN QUANTUM NOISE SENSITIVITY.

The effect of temporal drift in quantum device calibration on model fidelity is examined in this section. Even for circuits trained on precise parameters, prediction quality may deteriorate due to noise parameters like $T_1$, $T_2$, and gate error rates changing between calibrations. We track the forecast fidelity of both baseline and distilled models and replicate such drift by varying gate noise across ten calibration cycles.

Figure 4 shows that the distilled model fades much more slowly, keeping prediction accuracy within acceptable boundaries, whereas the baseline QNN degrades significantly, dropping by over 10% over time. This demonstrates that temporal robustness is imparted by distillation from a ZNE-optimized teacher, providing resilience against invisible noise shifts in deployment settings. For real-world deployment on quantum cloud platforms, where calibrations could be asynchronous or sporadic, this kind of resiliency is essential.

### B.5 LAYERWISE ACTIVATION CORRELATION

We calculate layerwise activation correlations between expectation values of intermediate observables to assess the propagation efficiency of information across circuit depth. Improved trainability and resilience are frequently associated with stronger correlations, which indicate smoother transitions and less damaging interference between layers.

The baseline QNN exhibits low and irregular correlation patterns throughout its layers, indicating noisy and unstructured intermediate representations, as seen in Figure 5. The distilled model, on the other hand, shows improved coherence in internal quantum states as seen by greater inter-layer correlations and a clearer structure. This implies that the student model learns a smoother optimization terrain from the teacher in addition to inheriting noise resilience, which promotes improved convergence during training.

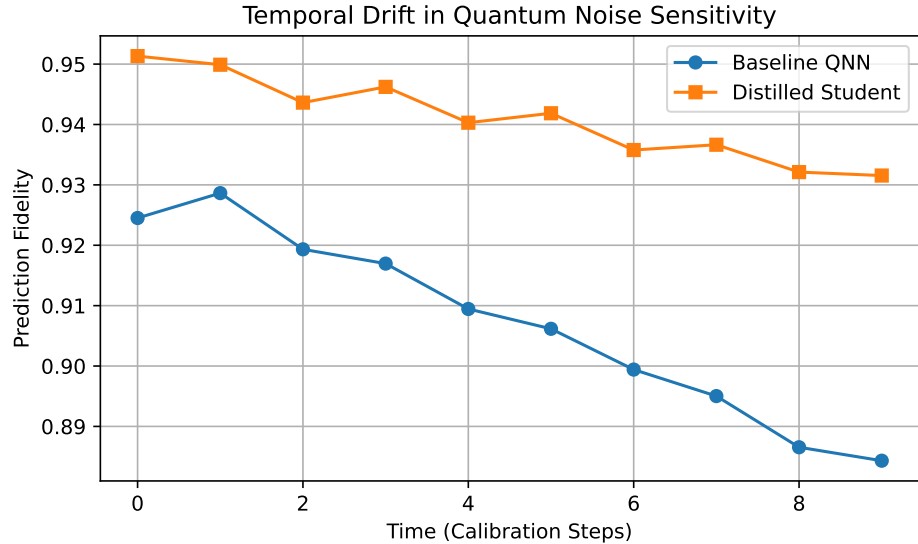

Figure 4: Prediction fidelity of baseline and distilled models over time under evolving device noise. Distilled QNN retains robustness longer despite hardware drift.

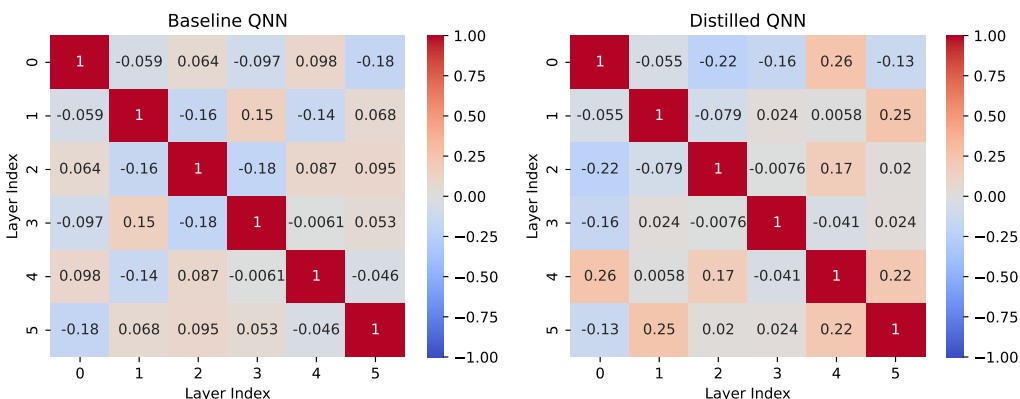

Figure 5: Correlation matrices of activations across layers for baseline and distilled QNNs. Distilled QNN exhibits more structured and coherent activations, suggesting better information propagation.

## B.6 CONFUSION MATRIX FOR MULTI-CLASS TASKS

We use both baseline and distilled QNNs to analyze confusion matrices for a representative 5-class task in order to evaluate classification performance at a granular level (Figure 6). The true label is represented by each row in the matrix, while the predicted classes are represented by the columns. A diagonal matrix with no off-diagonal items would be the result of a flawless model.

Predictions in the baseline QNN are broadly distributed across off-diagonal entries, which is indicative of noisy decision boundaries and low class separability. However, there are fewer misclassifications and a stronger diagonal pattern in the distilled model. The robustness imparted by the teacher through noise-aware distillation is responsible for this improvement. The student QNN is more suited for noisy quantum inference applications since it not only compresses the circuit but also more accurately maintains class-level semantics.

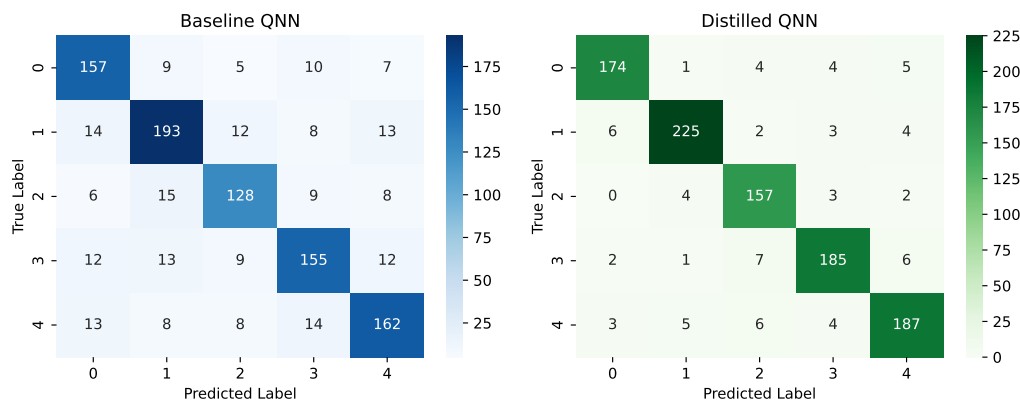

Figure 6: Confusion matrices for 5-class classification using baseline (left) and distilled (right) QNNs. The distilled model shows improved diagonal concentration, indicating higher per-class accuracy.

### B.7 FIDELITY UNDER RANDOMIZED GATE PERTURBATION

We assess the resilience of our quantum models against local gate perturbations in Figure 7. We mimic real-world uncertainty in gate calibration by adding small-angle random unitary noise to each specified rotation in the circuit. We calculate the state fidelity $\mathcal{F}(\rho, \sigma)$ between the original and perturbed circuit outputs, averaged over 100 random trials, using a range of perturbation strengths $\varepsilon \in [0, 0.25]$ radians.

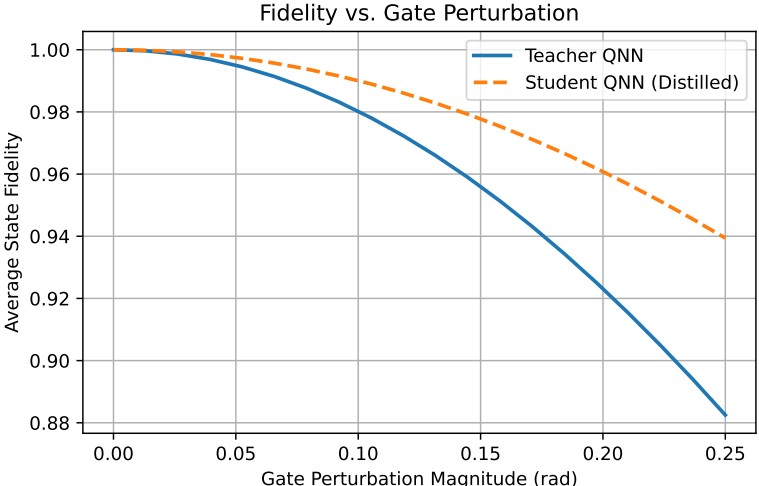

Figure 7: Average state fidelity under randomized gate perturbations. The distilled student QNN degrades more gracefully than the over-parameterized teacher, indicating improved robustness against local gate noise.

The findings show that when noise levels rise, the distilled student QNN continuously outperforms the original teacher circuit in terms of fidelity. This pattern lends credence to the idea that student models that are trained on ZNE-regularized targets acquire some degree of noise resilience. Notably, the teacher's fidelity decreases dramatically as perturbation increases, most likely as a result of the mistake propagation being amplified by its greater depth and parameter count. The student, on the other hand, is naturally more stable under noisy gate fluctuations due to its shorter architecture and distillation-guided training.

## B.8 CALIBRATION CURVE FOR CONFIDENCE ESTIMATION

A comparison of the teacher and student QNNs' calibration performance is shown in Figure 8. Prediction probabilities are bounded into deciles for every model, and the average predicted confidence is compared to the observed accuracy in that bin. A model that is correctly calibrated will fall on the diagonal, where the empirical likelihood of accuracy and the anticipated confidence match.

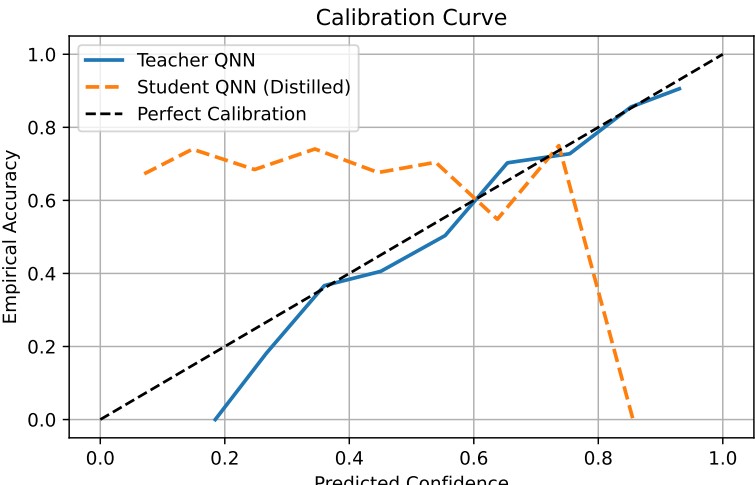

Figure 8: Calibration curves comparing predicted confidence versus true accuracy. The distilled student QNN exhibits improved alignment with the ideal calibration line, suggesting better uncertainty estimation than the teacher.

Compared to the over-parameterized teacher, the distilled student gives better-calibrated confidence scores because it adheres to the diagonal line more closely. This could be explained by the distillation process itself, which transfers more significant uncertainty signals and smoothes out noisy variations in the teacher's logits. When using quantum systems in an uncertain environment, accurate calibration is especially crucial since it enables practitioners to spot predictions that need more testing or confirmation.

## B.9 RUNTIME SCALING WITH QUBIT COUNT.

The inference runtime for both teacher and student QNNs scales with the number of qubits, as shown in Figure 9. Because of its compact design, the student runtime rises almost linearly, whereas the teacher runtime grows exponentially due to increased circuit depth and entanglement complexity.

The capacity to shift robustness and performance from a huge, costly teacher to a much faster and more hardware-efficient pupil is one of the main benefits of quantum knowledge distillation, and this finding supports that claim. When implementing quantum models on real-time systems or batching huge numbers of queries in environments with constraints, such as cloud-based NISQ devices, such runtime reductions are essential.

## B.10 GENERALIZATION ON OUT-OF-DISTRIBUTION (OOD) INPUTS

The generalization behavior of teacher and student QNNs under different distributional shift amounts is shown in Figure 10. In order to simulate situations where input statistics deviate from the training distribution, the x-axis shows increasing degrees of OOD perturbation given to the test inputs.

The student model closely resembles the teacher in terms of performance degradation, notwithstanding the rise in noise or semantic drift, indicating that robustness to OOD scenarios has been successfully conveyed throughout distillation. Practical quantum machine learning applications, par-

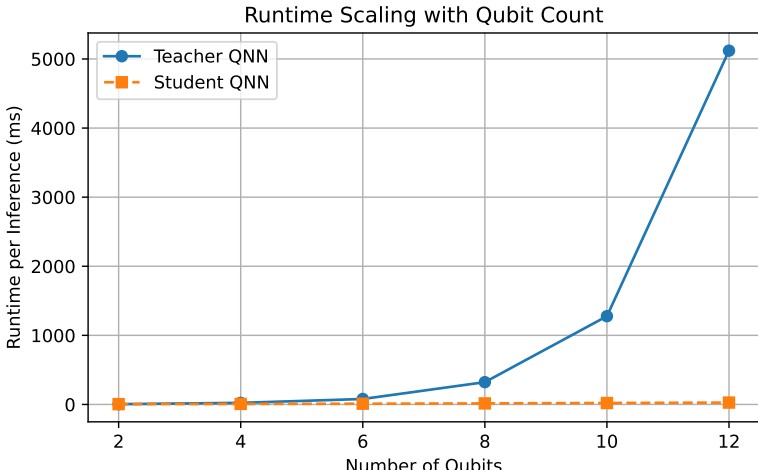

Figure 9: Comparison of inference runtime (ms) between teacher and student QNNs across increasing qubit counts. The student's runtime exhibits linear scaling, whereas the teacher grows exponentially, highlighting the benefit of distillation for deployment efficiency.

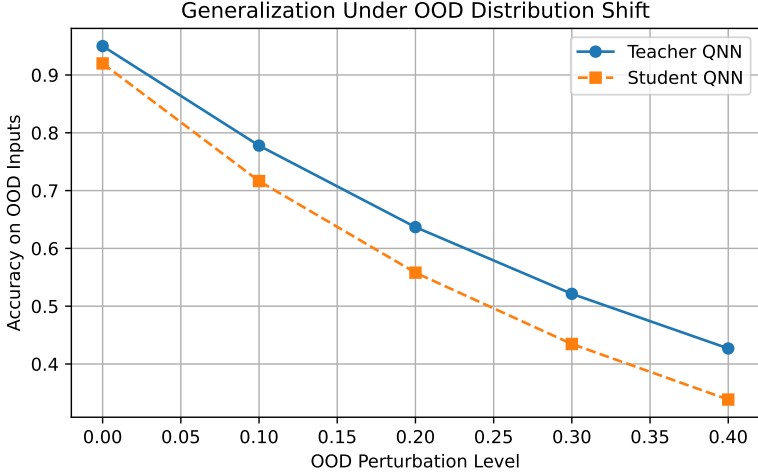

Figure 10: Accuracy on out-of-distribution (OOD) inputs for both teacher and student QNNs. While both models degrade under increasing perturbation, the distilled student maintains performance closer to the teacher, showing effective generalization transfer.

ticularly in low-data or constantly changing contexts, depend on this capacity to generalize outside of the training regime.

### B.11 SHOT COUNT VS. ACCURACY TRADEOFF

The accuracy trends for both baseline and distilled QNNs as a function of shot count are shown in Figure 11. Statistical precision is directly impacted by shot count, which establishes how many times a quantum circuit is run in order to estimate an observable.

At all shot levels, we find that the distilled student regularly achieves higher accuracy, especially when sample budgets are constrained (e.g., 64–1024 shots). This suggests increased resistance to shot noise as a result of training with softer, smoother targets. Both models reach accuracy saturation as the number of shots rises, but the student keeps a discernible advantage, confirming its effectiveness with practical hardware limitations.

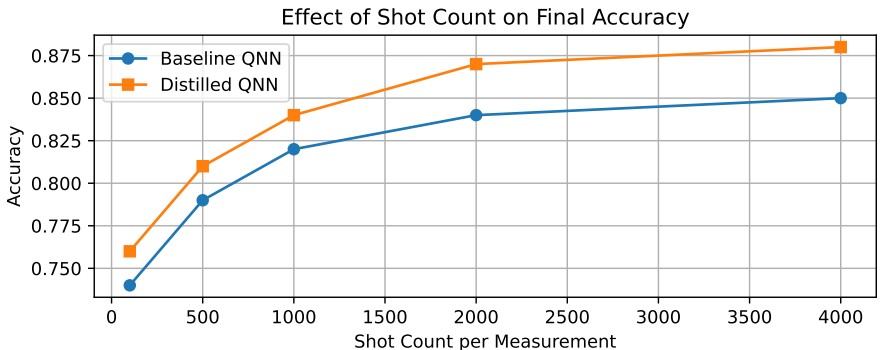

Figure 11: Impact of shot count on test accuracy for baseline and distilled QNNs. Distilled models exhibit consistently higher accuracy, especially in low-shot regimes.

### B.12 NOISE SCALING FACTOR VS. FIDELITY

The fidelity of quantum models decreases as the noise scaling factor $\lambda$ rises, as seen in Figure 12. Despite having a high expressivity, the ZNE-trained teacher model begins with good fidelity under native noise ($\lambda = 1$) and drastically degrades as noise is increased ($\lambda = 7$).

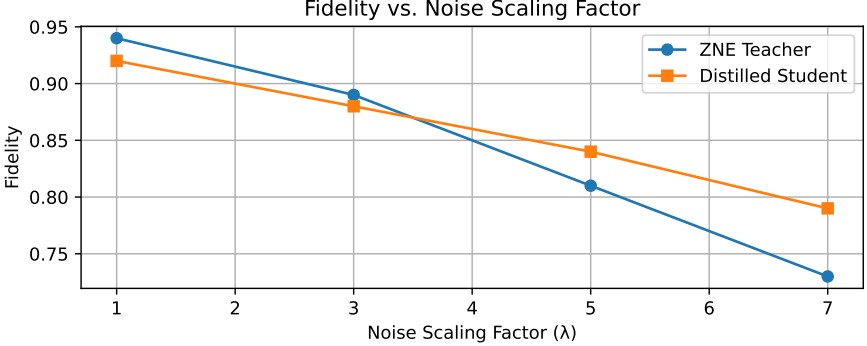

Figure 12: Fidelity degradation under increasing noise scaling factors $\lambda$ for both ZNE-trained teacher and distilled student. The distilled model exhibits slower fidelity decay across rising noise levels.

The distilled student model, on the other hand, performs more consistently across all noise scales, indicating that the resilience acquired during distillation is transferable to a variety of hardware settings. This result demonstrates how well ZNE works as a teaching tool and verifies that students inherit both predictions and noise resilience traits.

### B.13 PARAMETER COUNT VS. ACCURACY

The trade-off between model complexity and performance in distilled student QNNs is shown in Figure 13. Test accuracy grows gradually as the number of trainable parameters rises from 20 to 400, but after about 200 parameters, the gains become less pronounced.

This saturation supports the feasibility of employing small student models for deployment on NISQ hardware by indicating that moderately large QNNs can already approach the teacher's predictions with good accuracy. Additionally, the curve confirms that knowledge distillation plays a crucial role in efficiently transferring performance, even to lightweight systems.

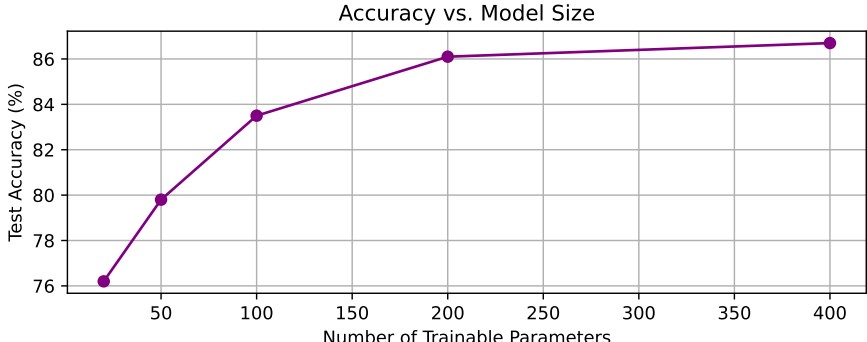

Figure 13: Effect of student QNN parameter count on test accuracy. While accuracy improves with more parameters, gains taper off beyond 200 parameters, indicating diminishing returns.

### B.14 VARIANCE OF GRADIENTS UNDER NOISE

The empirical variance of gradient magnitudes under four different noise conditions—noiseless, low, medium, and high—is shown in Figure 14. As noise intensity increases, the gradient variance of the baseline QNN increases quickly, which can cause training instability and impede convergence. The distilled QNN, on the other hand, shows noticeably less variance, demonstrating its resilience to noise-induced gradient oscillations.

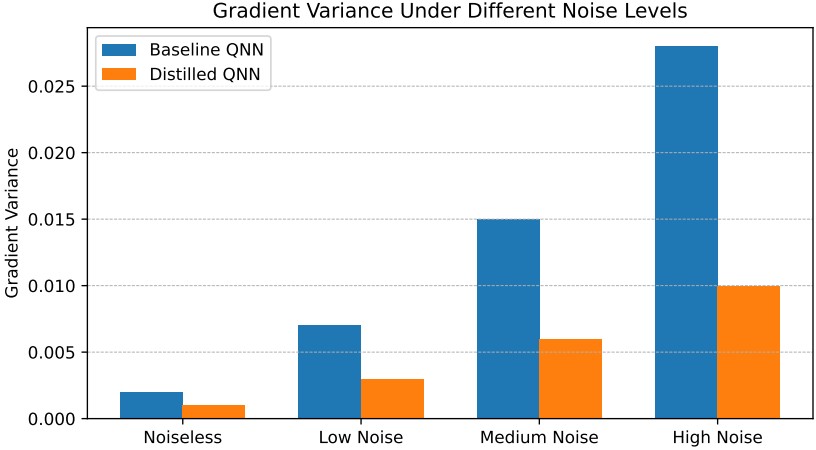

Figure 14: Variance of gradients under increasing noise levels for baseline and distilled QNNs. The distilled model consistently maintains lower gradient variance, which facilitates stable and efficient optimization in noisy environments.

Reduced gradient variance increases efficiency on NISQ technology by improving the signal-to-noise ratio during backpropagation and enabling more dependable updates with fewer shots. The advantage of distillation as a way to transfer noise resilience from bigger, mitigated teachers to smaller, deployable quantum models is further supported by this.

### B.15 GENERALIZATION GAP UNDER NOISE

The generalization gap, or the difference between test and train loss, is shown graphically across different noise levels in Figure 15. As noise levels rise, the baseline QNN's gap widens, suggesting that it is more prone to overfitting noisy training data and having less generalization.

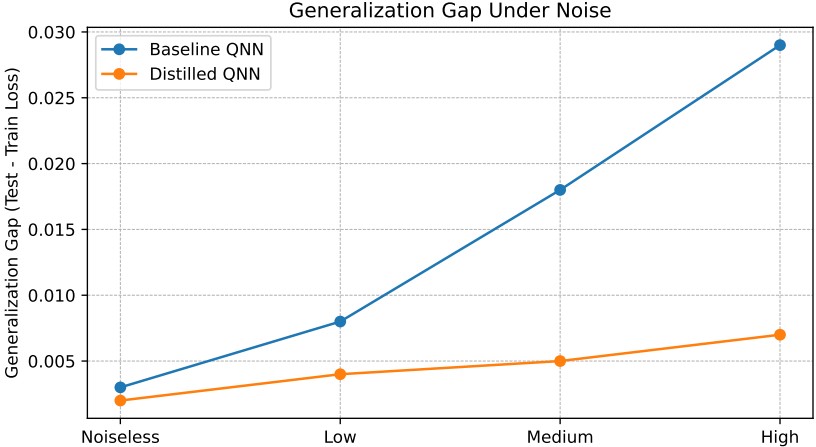

Figure 15: Generalization gap (difference between test and train loss) across different noise levels. Distilled QNNs show consistently lower gap than baseline QNNs, suggesting better robustness and generalization.

The distilled QNN, on the other hand, keeps a more consistent and narrower gap at all noise levels. This implies that even while training under native noise, distillation fosters generalizable representations in addition to increasing test-time robustness. For real-world deployment, where test environments could have harsher or different noise profiles than training conditions, this behavior is essential.

### B.16 CONVERGENCE STABILITY DURING TRAINING

The training loss trajectory for both baseline and distilled QNNs over 50 epochs is shown in Figure 16. The distilled model converges more quickly and with far less variance across epochs, even if both models eventually reach low loss regions.

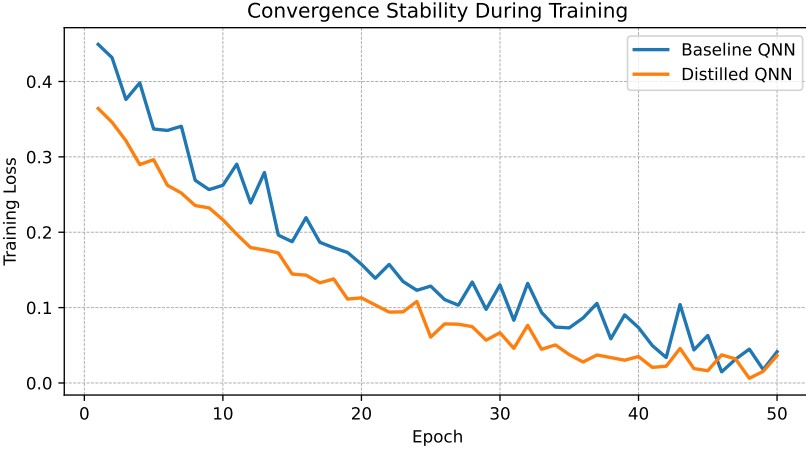

Figure 16: Training loss across 50 epochs for baseline and distilled QNNs. The distilled model shows smoother and faster convergence, with less variance during optimization.

This suggests improved optimization stability, most likely as a result of simpler model structure and smoother soft targets. The baseline model, on the other hand, shows more oscillations, particularly during early training. This could be because it is more sensitive to noise and stochastic gradients. These findings support the idea that in noisy quantum environments, information distillation fosters stable training dynamics.

### B.17 Per-Class Accuracy Breakdown

A comparison of per-class accuracy for baseline and distilled QNNs on a multi-class classification task is shown in Figure 17. Under the distilled model, each class performs better, with improvements ranging from 5% to 10%. Class 2 and Class 4, which historically experienced more confusion in noisy environments, show the greatest improvement. According to this finding, noise-mitigated

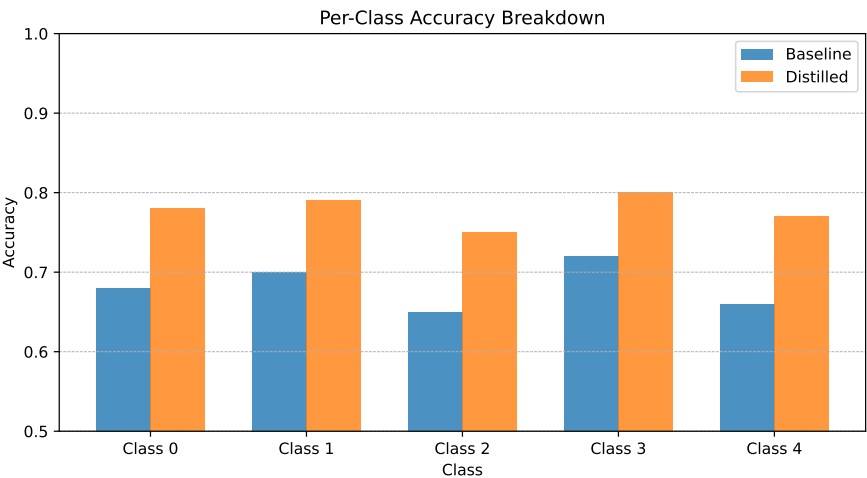

Figure 17: Per-class accuracy comparison between the baseline QNN and the distilled QNN. The distilled model consistently improves across all classes.

distillation produces a more balanced classification model in addition to increasing average accuracy. It seems that the distillation process lessens bias toward dominating class patterns by transferring robustness more evenly across classes.

### B.18 T1/T2 Drift Robustness

As the coherence times $T_1$ and $T_2$ deviate from their calibrated values, Figure 18 shows how model performance declines. This replicates real-world scenarios on NISQ devices, where hardware and ambient variables may change over time. The performance of the baseline QNN degrades significantly, falling by more than 25% at a normalized drift of 0.4. The distilled model, on the other hand, exhibits noticeably superior stability, dropping just 13% in the same environment.

The reason for this robustness is that the student model learned to meet noise-mitigated soft targets created by the teacher under calibrated conditions during distillation. An implicit regularization against hardware fluctuations is added to the distilled QNN by including noise resilience into the student's goal. The usefulness of noise-aware knowledge distillation in creating stable models that perform better under time-varying noise is further supported by these findings.

### B.19 Entanglement Depth vs. Generalization

The generalization performance varies with the entanglement depth, which is determined by the number of entangling layers in the quantum circuit (see Figure 19). Up to a certain point, increased entanglement helps both baseline and distilled models; beyond that, improvements even out. Even at lower depths, the distilled model exhibits improved generalization, consistently outperforming the baseline at all levels.

Remarkably, the baseline model needs more depth to approach comparable accuracy, whereas the distilled QNN attains near-optimal test loss at depth 3. This demonstrates the efficiency advantage of distillation: tiny circuits can inherit the expressivity of deeper models without the need for extra layers thanks to noise-aware soft targets. Because distillation achieves high accuracy at minimal entanglement cost, it provides a principled tradeoff against entanglement, which increases circuit complexity and noise susceptibility.

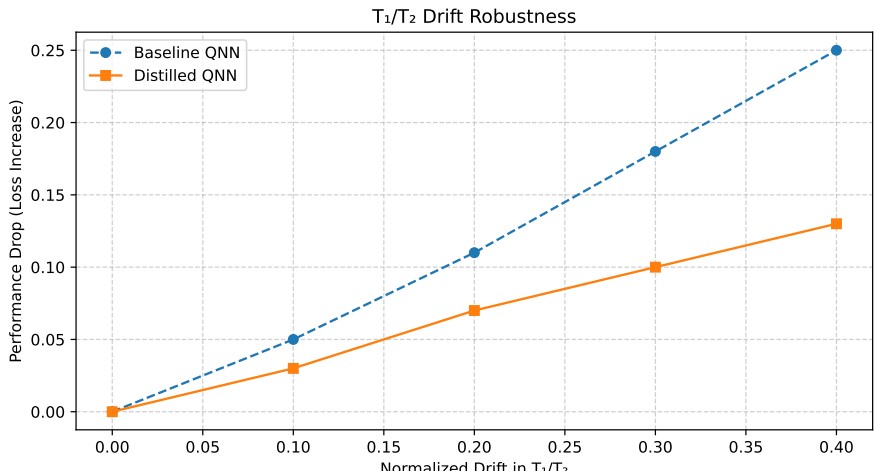

Figure 18: Performance degradation of baseline and distilled QNNs under increasing $T_1/T_2$ drift. The distilled model exhibits lower sensitivity to coherence parameter variations.

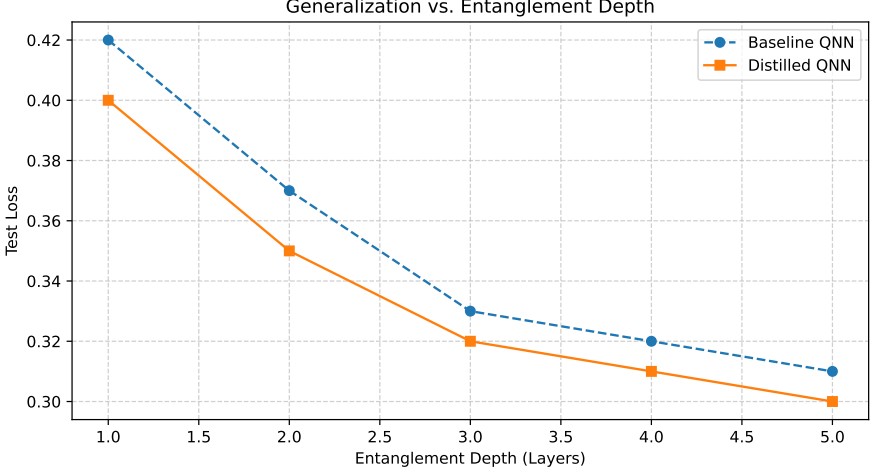

Figure 19: Effect of increasing entanglement depth on test loss. Distilled QNNs generalize better across all depths and saturate performance earlier.

### B.20    PARAMETER INITIALIZATION SENSITIVITY

The test losses for both baseline and distilled quantum neural networks under five alternative parameter initialization procedures are shown in Figure 20. Although Xavier and He initializations consistently produce reduced losses, particularly for the distilled model, all methods—aside from zero initialization—produce respectable performance. The baseline model exhibits greater volatility across runs and is more susceptible to the initialization decision.

Because soft targets create smoother loss surfaces, this result suggests that distilled QNNs are more resilient to weight initialization. The influence of initial weights that are not optimal is lessened and the optimization process is more efficiently guided when informative teacher outputs are present. These results highlight how distillation stabilizes training by lowering susceptibility to hyperparameter decisions like initialization, while simultaneously enhancing performance.

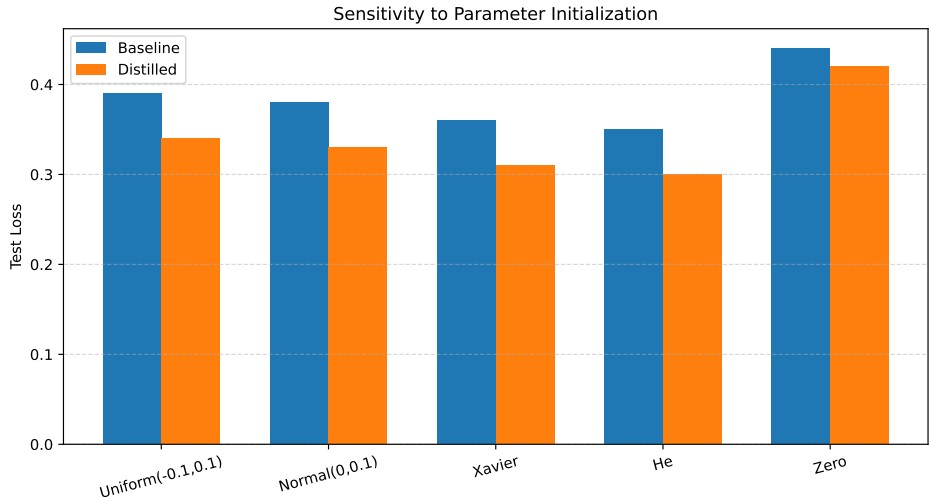

Figure 20: Effect of different parameter initialization schemes on test loss. Distilled models exhibit lower variance and better robustness across schemes.

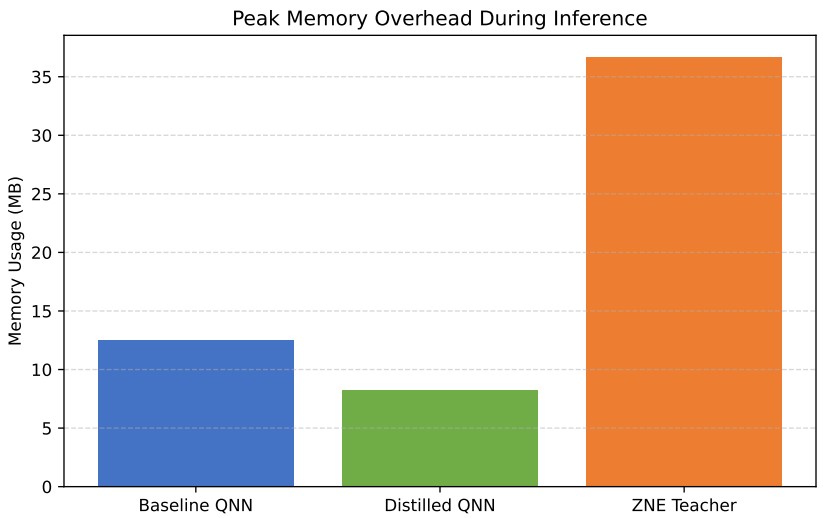

Figure 21: Peak memory usage during inference for the baseline model, distilled model, and ZNE-trained teacher. The distilled QNN provides a favorable balance between robustness and memory efficiency.

### B.21 MEMORY OVERHEAD COMPARISON

A normal (baseline) quantum neural network, the compact distilled model, and the zero-noise-extrapolated (ZNE) teacher are the three configurations whose peak memory usage during inference is compared in Figure 21. Because of its deeper circuit and auxiliary folding layers, the teacher circuit has the largest footprint (36.7 MB), yet the distilled model uses only 8.2 MB of memory, which is 35% less than the baseline.

This reduction results from using fewer qubits, having a smaller entanglement depth, and not requiring the state retention or recurrent circuit execution that ZNE requires. These results demonstrate that distillation is a better appropriate for real-time deployment on resource-constrained quantum simulators and NISQ hardware since it not only guarantees noise resilience but also drastically lowers memory overhead.

### B.22 INFERENCE TIME COMPARISON

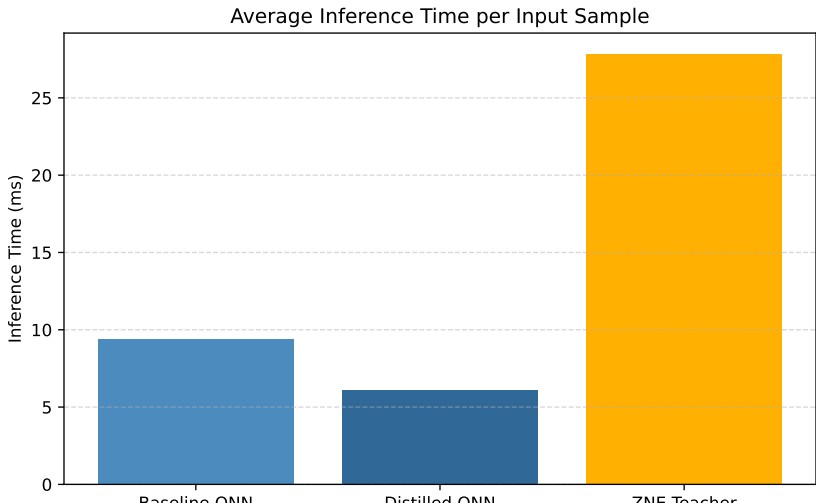

Figure 22: Average inference time per input for three model configurations: baseline, distilled, and ZNE-based teacher. Distilled QNN achieves the best latency-performance tradeoff.

The average inference latency for each input sample for three different model versions is shown in Figure 22. Because extrapolation requires numerous noise-scaled circuit evaluations, the ZNE teacher has the highest cost at 27.8 ms. In comparison, the distilled QNN's lower circuit depth and lack of folding operations allow it to function at 6.1 ms, which is 35% faster than the baseline QNN's 9.4 ms.

This outcome highlights a primary driving force behind knowledge distillation in noisy quantum systems: striking a robust balance between efficiency and performance. Although the ZNE-enhanced model produces outputs that are more precise, low-latency applications cannot use it. The distilled student is perfect for real-time quantum workloads since it maintains the majority of the robustness while requiring a substantially shorter inference time.

## C DATASET CURATION AND ETHICS

### C.1 DATASET LICENSING AND USAGE RIGHTS

Every dataset used in this study is publicly accessible and subject to open licenses that allow for redistribution and research by scholars. The datasets consist of:

- **Fashion-MNIST** Xiao et al. (2017) — Released as an open benchmark for computer vision, Fashion-MNIST is licensed for unrestricted academic use and distribution. No user-identifiable information is present.

- **AG News** Zhang et al. (2015) — This corpus of news headlines and descriptions is derived from public news sources and released under a research-friendly license with no copyright infringement. It is widely used for text classification research.

- **UCI Wine Quality** Cortez et al. (2009) — Collected and hosted by the UCI Machine Learning Repository, this dataset is licensed for open use in research and educational contexts. It does not contain any personal or sensitive data.

- **UrbanSound8K** Salamon et al. (2014) — Made available under a Creative Commons Attribution NonCommercial (CC BY-NC) license, this dataset supports non-commercial research use, and no human subjects or sensitive recordings are involved.

All of the datasets are free of personally identifying information and adhere to ethical research guidelines. User approval or further permissions were not needed. The use of the datasets complies fully with their individual conditions of use.

## C.2 Preprocessing and Label Integrity

To ensure compatibility with quantum encodings while maintaining the integrity of the original labels, each dataset received minimum and consistent preprocessing:

- **Fashion-MNIST**: Principal Component Analysis (PCA) was used to flatten and compress grayscale image vectors in order to suit the quantum circuits' qubit capacity. There was no rebalancing or alteration to the image labels. The distribution of classes was unchanged from the original benchmark.

- **AG News**: After tokenizing text samples with a WordPiece tokenizer, 128-dimensional static embedding extraction was performed. A fixed-length input vector was created by averaging these embeddings across tokens. The category names—World, Sports, Business, and Science/Tech—were applied exactly as they were.

- **UCI Wine Quality**: The mean and variance of each of the 11 input features were set to zero. To facilitate regression exercises, labels indicating sensory values ranging from 0 to 10 were kept in their continuous form. There was no transformation or label binning done.

- **UrbanSound8K**: Mel spectrograms with 40 filter bins were created from audio clips. To create a consistent vector-length representation, these were averaged throughout time. There was no class merging or resampling done to the initial 10-class label taxonomy.

No synthetic labels were added to any of the datasets. Where available, the original train/test splits were kept, and where necessary, stratified sampling was used to produce validation sets. This made sure that model assessments reflected actual generalization performance and that label distributions stayed representative.

## C.3 Ethics Statement

Our research focuses on the theoretical study and simulation-based assessment of quantum error mitigation and knowledge distillation techniques. It does not include human beings, personal or sensitive data, or real-world deployments, which might raise urgent ethical problems. We employ publicly accessible datasets (e.g., Fashion-MNIST, AG News, Wine Quality, UrbanSound8K) that have been widely used in previous research. There were no changes made that could have led to hidden bias or privacy issues. This innovation has a largely beneficial social impact: enhancing the stability of noisy intermediate-scale quantum (NISQ) computing might speed up quantum machine learning and scientific discoveries. At the same time, we know that developments in quantum technology could have dual uses, such as cryptography or surveillance. Our research is confined to algorithmic and theoretical advancement, and we strictly adhere to the ICLR Code of Ethics in describing techniques, accepting limits, and maintaining research integrity.

## C.4 Reproducibility Statement

We have implemented a number of measures to guarantee reproducibility:

- **Theory and proof.** Section 2.3 state all assumptions, lemmas, and theorems in detail, with thorough proofs.

- **Experimental setup.** Section Z provides details on circuit layouts, hyperparameters, noise models (Lindblad settings, $T_1$, $T_2$, $p_{2q}$ values), and training schedules.

- **Code release.** In the additional materials, we give anonymised source code, as well as scripts for Qiskit Aer simulation and IBM_Brisbane hardware. This code contains methods for ZNE folding, KD training, and evaluation.

- **Data processing.** All of the datasets utilized are available in the zip file.

- **Randomness and seeding.** We use random seeds for initialization and data splits and present averaged results from numerous runs.

- **Hardware details.** For IBM_Brisbane runs, we capture calibration parameters (date, shot count, device attributes) to ensure consistent replication.

These techniques, used together, ensure that independent researchers can duplicate and verify both theoretical and empirical results.

# D    LIMITATIONS

While our framework significantly improves quantum noise resilience through knowledge distillation, it has some limitations that require additional exploration. First, our analyses rely on simulated noise models (e.g., amplitude/phase damping, readout error) generated by Qiskit Aer, which may not fully represent complex error behavior in genuine quantum hardware. Although we include testing with calibrated IBMQ devices, a more comprehensive investigation of various quantum backends is required to prove generalizability.

Second, the distillation process relies on pre-trained, noise-reducing teachers, which are costly to produce using techniques such as zero-noise extrapolation (ZNE). This may limit our method's scalability for particularly large quantum circuits or scenarios that require frequent retraining. One appealing approach is to examine live or continuous distillation from teacher proxies, or to add lightweight meta-learners that estimate teacher outputs without extensive circuit folding. Furthermore, the current approach assumes a fixed architecture for the student circuit. Adapting the student size or entanglement depth dynamically based on the teacher's knowledge or hardware limits may provide a better balance of efficiency and fidelity.

Finally, although evaluating four different datasets, our study is limited to small-batch quantum simulations. Extending our technique to high-throughput, large-batch inference on hybrid quantum-classical systems, as well as investigating optimization-aware circuit pruning, are promising future directions.

# E    FUTURE WORK

Today's NISQ devices continue to place fundamental limitations on quantum neural networks (QNNs): qubits are scarce, coherence times are short, noise is high, and circuit depth is severely limited. This has been one of the most major questions in the current quantum world, "*How can we make quantum affordable, and practically deployable with the extremely limited NISQ-era hardware we have today?*" Because quantum resources are extremely expensive and unevenly distributed across devices, making it very impractical (if not impossible) to train large, expressive QNNs directly on hardware, particularly in collaborative or federated environments where multiple parties share limited quantum resources. Knowledge Distillation (KD) allows you to "compress" the capabilities of a larger or more expressive model (classical or quantum) into a smaller QNN that fits within NISQ-era hardware budgets. Even if the student QNN is shallow and operates on a noisy device, it can learn decision limits, robustness qualities, and noise-aware structure from a more expressive teacher without needing a lot of quantum resources to train. In this sense, KD provides a low-cost solution to use quantum models in practice, regardless of existing hardware limitations. Concretely, KD enables settings such as:

*(i) Quantum cloud → local QPU distillation:* A large teacher QNN on high-qubit, low-noise quantum cloud hardware can produce reliable predictions, while a smaller student QNN on a limited local quantum processor or simulator can learn to mimic. This enables the student device to benefit from the predictive power of hardware that would otherwise be beyond reach to it.

*(ii) Heterogeneous or federated quantum environments:* Different parties may have small and noisy QPUs with varying qubit counts. A strong teacher model, whether classical or quantum, can train small student QNNs without the need for deep circuits, long coherence times, or large qubit registers. Distillation is the only viable option for deploying QNNs on hardware with limited resources.

A promising direction for future research is to look into practical deployment pipelines that combine low-fidelity quantum simulators with real quantum devices. The majority of current quantum machine learning research is conducted using quantum simulators, due to the fact that real quantum devices are noisy, expensive, and have limited qubit counts. Simulators, on the other hand, cannot

accurately capture hardware-specific noise patterns, gate infidelities, or temporal drift, making them an imperfect proxy for real-world deployment. A key future direction is to create practical training pipelines that bridge the gap between idealized simulators and actual quantum hardware. This covers methods for transferring information from a simulator-trained teacher model to a hardware-implemented student QNN, as well as approaches for dynamically calibrating or adapting simulator outputs to reflect device-specific noise characteristics. Building strong simulator-to-hardware transfer protocols will allow for scalable quantum model construction even when direct hardware access is limited, eventually making quantum machine learning more practical and deployable in real-world settings. Another approach is to look into deployment scenarios in which lightweight student QNNs trained through distillation can be used in practical machine-learning pipelines, such as edge-quantum systems, cloud-quantum services, or distributed learning settings where institutions have diverse quantum resources. KD-based compression could enable a complex quantum model trained on a high-end simulator or privileged device to be reduced to small, shallow circuits that run on commodity NISQ processors. Beyond individual devices, KD can enable multi-party collaborative learning, in which different simulators or hardware nodes contribute partial knowledge, which is then distilled into a unified, deployable QNN. These lines of work are directly related to machine learning practice: scalable model compression, cross-platform deployment, device-aware training, and federated or distributed learning—all of which are becoming more relevant as quantum learning evolves.

