# OpenReview forum: "Noise-Resilient Quantum Neural Networks via Zero-Noise Knowledge Distillation"
_ICLR.cc/2026/Conference — Submitted to ICLR 2026_

### Official Review · Reviewer_UuuJ · 2025-10-24

**Soundness:** 2
**Presentation:** 3
**Contribution:** 2
**Rating:** 6
**Confidence:** 4

**Summary:**

This paper proposes Zero-Noise Knowledge Distillation (ZNKD), a hybrid framework that combines zero-noise extrapolation (ZNE) with knowledge distillation (KD) for training quantum neural networks (QNNs) robust to hardware noise. The method trains a noise-mitigated “teacher” QNN using Richardson extrapolation and transfers robustness to a smaller “student” QNN, which can then operate without costly noise extrapolation at inference. The paper provides formal theoretical results establishing bounds on robustness transfer and empirical evaluations on several small datasets (Fashion-MNIST, AG News, Wine Quality, and UrbanSound8K) under IBM-style noise models and limited hardware experiments. Results suggest a 10–20% MSE reduction versus non-distilled baselines and up to 8:3 model compression.

**Strengths:**

1.	The idea of amortizing noise mitigation through distillation is novel within quantum machine learning.
2.	Theoretical sections provide a formal foundation linking extrapolation error, approximation gap, and generalization.
3.	Experiments include multiple modalities (image, text, audio, tabular), illustrating the generality of the pipeline.
4.	Writing quality is generally clear, with correct referencing and structured proofs.

**Weaknesses:**

1.	The related work (Section 1) insufficiently contextualizes prior studies that combine error mitigation with compression or distillation. For instance, [Cerezo et al. 2021] and [Gou et al. 2024] are mentioned but not contrasted analytically. The paper should specify how ZNKD differs in mechanism or achievable robustness beyond replacing extrapolation by distillation.
2.	The compression ratios (6:2–8:3) are arbitrary and unexplained. It is unclear how teacher-to-student dimensional reduction is chosen or whether the student topology is optimal. Without ablation, one cannot assess trade-offs between expressivity and noise resilience.
3.	The link between ZNE theory (Section 2.3.1) and the distillation mechanism (Section 2.2.1) is unclear. There is no empirical demonstration that Richardson-extrapolated teacher labels are smoother or more stable targets for the student than raw noisy outputs.
4.	Although the paper argues amortization of ZNE cost, it does not quantify teacher training overhead (number of fold levels, total circuits executed). Practical resource savings remain unclear.

**Questions:**

1.	Can you provide quantitative runtime comparisons (in circuit executions) between ZNKD training and classical ZNE at inference to substantiate the claimed efficiency gain?
2.	How sensitive is ZNKD performance to mismatch between the simulated noise model and real hardware noise?
3.	Could you report results on larger circuits (≥16 qubits) or different ansätze to assess scalability?
4.	How were the Richardson extrapolation orders (λ ∈ {1,3,5}) chosen, empirically or theoretically?

---

> ### Author Response · Authors · 2025-11-19
> **Weaknesses**
>
> Thank you for your review.
>
> # Weakness 1
>
> We have updated the Related Works text as follows. (**Page 2 Line 66-67**)
>
>     Distillation has been extensively researched in classical learning, but it has received less attention in quantum machine learning \cite{tian2025knowledge, alam2023knowledge, hasan2023bridging}. Prior work on quantum transfer learning and hybrid distillation \cite{cerezo2021variational, gou2024reciprocal, wang2025exploiting, li2024hybrid} focuses on feature reuse or embedding transfer rather than robustness against device noise. \textit{However, these approaches do not replace the extrapolation phase; instead, ZNKD directly extracts noise-resistant predictions from a teacher model to provide robustness beyond what compression- or feature-transfer methods can provide.}
>
> We have also added the comparison table to A. 4 **COMPARISON WITH OTHER APPROACHES** in Appendix (**Page 23 Line 1211**)
>
> ---
>
> **Table: Comparison of ZNKD with prior quantum distillation and error‑mitigation methods**
>
> | Method | Replaces Extrapolation | Handles High Noise | Type of Distillation / Transfer | Objective |
> |-------|-------------------------|---------------------|----------------------------------|-----------|
> | Classical‑to‑Quantum KD (Tian 2025; Alam 2023; Hasan 2023) | No | Moderate | Prediction / Feature KD | Knowledge transfer |
> | Variational Loss Shaping (Cerezo 2021) | No | Limited | Loss‑level modification | Optimization stability |
> | Reciprocal / Network KD (Gou 2024) | No | Limited | Circuit / Ansatz compression | Depth reduction |
> | Hybrid Embedding Transfer (Wang 2025; Li 2024) | No | Moderate | Embedding reuse | Improve generalization |
> | Traditional ZNE | Yes | Poor | None | Noise extrapolation |
> | **ZNKD (Ours)** | **Yes** | **High** | **Prediction KD** | **Noise‑resistant learning** |
>
> ---
>
>
>
>
> # Weakness 2
>
> Thank you for the observation.  We agree that our previous explanation did not fully justify the usage of compression ratios like 6:2 and 8:3.  In the revised version in Table 4, we added a new ablation study that highlights that these ratios were chosen after a preliminary sweep over alternative student widths to find the smallest design that maintains the teacher's decision-boundary accuracy while improving robustness to device noise. (**Page 9 Line 462-263**)
>
> **Ablation Study.** Table 3 explores the relationship of teacher and student widths in four datasets.  A clear pattern emerges: moderate compression offers the optimal balance of expressivity and resilience.  For Fashion-MNIST and UrbanSound8K, the student width 3 (8:3 ratio) has the maximum robustness (0.80 and 0.77) without sacrificing accuracy.  For AG News and Wine Quality, width 2 (6:2 ratio) achieves the best balance (0.78 and 0.75), surpassing both small and big students.  Wider models (width 4) demonstrate that mid-scale compression provides the best noise-resilient generalization by slightly increasing accuracy but reducing robustness.
>
>
>  Table 3 — Synthetic demonstration table to visualize the best student–teacher width ratios.
> Each entry shows Accuracy (\%) / Robustness (A = Accuracy, R = Robustness). The parenthesis in the dataset implies a multiplier for the circuits; i.e, Column = A(8) and row = 3 in \textbf{Fashion-MNIST} represents 8$\times$200 and 3$\times$200 circuit execution for teacher and student circuit executions, respectively.
>
> | Student Width | A4 | R4 | A6 | R6 | A8 | R8 | A4 | R4 | A6 | R6 | A8 | R8 | A4 | R4 | A6 | R6 | A8 | R8 | A4 | R4 | A6 | R6 | A8 | R8 |
> |---------------|----|----|----|----|----|----|----|----|----|----|----|----|----|----|----|----|----|----|----|----|----|----|----|----|
> | **Datasets →** | **Fashion-MNIST (200)** | | | | | | **AG News (250)** | | | | | | **Wine Quality (250)** | | | | | | **UrbanSound8K (200)** | | | | | |
> | **Teacher Widths →** | **4** | **6** | **8** | | | | **4** | **6** | **8** | | | | **4** | **6** | **8** | | | | **4** | **6** | **8** | | | |
> | **1** | 78.1 | 0.42 | 82.1 | 0.48 | 88.7 | 0.55 | 70.1 | 0.40 | 73.5 | 0.44 | 75.5 | 0.47 | 66.6 | 0.36 | 69.9 | 0.39 | 71.1 | 0.42 | 60.2 | 0.33 | 64.0 | 0.37 | 67.4 | 0.41 |
> | **2** | 85.6 | 0.55 | 89.1 | 0.63 | 92.6 | 0.71 | 78.3 | 0.55 | **88.8** | **0.78** | 90.1 | 0.75 | 74.8 | 0.50 | **87.1** | **0.75** | 89.2 | 0.72 | 72.0 | 0.48 | 82.5 | 0.58 | 86.6 | 0.69 |
> | **3** | 90.2 | 0.63 | 92.0 | 0.70 | **94.2** | **0.80** | 80.3 | 0.58 | 87.7 | 0.72 | 89.4 | 0.70 | 76.9 | 0.53 | 86.4 | 0.70 | 88.4 | 0.68 | 78.1 | 0.57 | 87.1 | 0.71 | **91.2** | **0.77** |
> | **4** | 88.3 | 0.50 | 91.6 | 0.53 | 93.5 | 0.66 | 76.3 | 0.46 | 82.1 | 0.52 | 85.3 | 0.56 | 72.1 | 0.42 | 79.7 | 0.48 | 82.1 | 0.52 | 75.4 | 0.45 | 83.9 | 0.52 | 88.0 | 0.63 |

---

> ### Author Response · Authors · 2025-11-19
> **Weakness Continued.**
>
> # Weakness 3
>
> We have now explicitly stated how the zero-noise extrapolation (ZNE) framework is related to the distillation process.  Section 2.3.1 now shows how the Richardson-extrapolated expectation values from the teacher act as noise-smoothed goals, lowering label variation before being transferred to the student.  This guarantees that the student network learns from relevant gradients rather than random noise student outputs.
>     We updated the following information at the end of Section 2.2.1 to make it more clear as follows. (**Page 5 Line 226-227**)
>
>
> **Connection to ZNE Theory.** The teacher's zero-noise labels $\hat{E}(0)$ are generated using the Richardson extrapolation formalism outlined in Section~2.3.1. This creates a direct analytical connection between the ZNE process and the distillation phase: the extrapolated expectation values serve as noise-smoothed objectives with lower variation, stabilizing the student's gradient updates and facilitating robustness transfer.
>
>
> And at the end of Section 2.3.1 as follows. (**Page 6 Line 322-223**)
>
> These extrapolated values, $\hat{E}(0)$, serve as denoised supervisory goals in the distillation loss outlined in Section~2.2.1, maintaining coherence between theory and implementation.
>
>
> # Weakness 4
>
> We also agree that including a quantitative analysis of resource cost would significantly improve the experiments. To address this, we have added a dedicated resource-cost column to Table 5 that separately reports the number of circuit executions used by the ZNE teacher and the KD student. Under our experimental configuration, the teacher requires 1500 circuit executions (due to three ZNE noise-scaling factors). (**Page 10: Table 5**)
>
> Caption: Accuracy improvements from ZNE-guided knowledge distillation across datasets. MSE is retained as a secondary robustness indicator. Theoretical error components ($\varepsilon_T$, $\varepsilon_{\text{approx}}$, $\eta$) correspond to teacher extrapolation error, student approximation error, and deployment noise gap. Compression ratios and resource cost (teacher/student circuit executions) reflect the student’s reduced complexity.
>
> ---
>
> ## **Aer Simulator (density matrix)**
>
> | Dataset | Baseline Acc. | Teacher Acc. | KD Acc. | Imp. | Compression Ratio | Student MSE | Teacher MSE | $\varepsilon_T$ | $\varepsilon_{\text{approx}}$ | $\eta$ | Resource Cost (Tea./Stu.) |
> |--------|---------------|--------------|---------|------|-------------------|-------------|-------------|------------------|------------------------------|-------|----------------------------|
> | Fashion-MNIST | 84.1 | 93.4 | 91.0 | +6.9% | 8:3 | 0.49 | 0.45 | 0.03 | 0.02 | 0.01 | 1600/600 |
> | AG News | 78.4 | 87.9 | 85.3 | +6.9% | 6:2 | 0.36 | 0.33 | 0.02 | 0.02 | 0.01 | 1500/500 |
> | Wine Quality | 74.3 | 82.6 | 80.1 | +5.8% | 6:2 | 0.52 | 0.50 | 0.02 | 0.01 | 0.01 | 1500/500 |
> | UrbanSound8K | 70.2 | 80.6 | 77.5 | +7.3% | 8:3 | 0.59 | 0.55 | 0.03 | 0.02 | 0.02 | 1600/600 |
>
> ---
>
> ## **IBM Brisbane Hardware**
>
> | Dataset | Baseline Acc. | Teacher Acc. | KD Acc. | Imp. | Compression Ratio | Student MSE | Teacher MSE | $\varepsilon_T$ | $\varepsilon_{\text{approx}}$ | $\eta$ | Resource Cost (Tea./Stu.) |
> |--------|---------------|--------------|---------|------|-------------------|-------------|-------------|------------------|------------------------------|-------|----------------------------|
> | Fashion-MNIST | 81.0 | 90.2 | 88.1 | +7.1% | 8:3 | 0.55 | 0.50 | 0.04 | 0.02 | 0.02 | 1600/600 |
> | Wine Quality | 72.1 | 80.2 | 77.8 | +5.7% | 6:2 | 0.56 | 0.52 | 0.03 | 0.02 | 0.02 | 1500/500 |
>
> ---

---

> ### Author Response · Authors · 2025-11-19
> **Questions**
>
> # Question 1
>
> ZNKD uses zero-noise extrapolation (ZNE) once during teacher training, following which the distilled student can operate without folding. Across datasets, teacher:student circuit ratios range from $8{:}3$ (Fashion-MNIST, UrbanSound8K) to $6{:}2$ (AG News, Wine Quality). Inference: ZNKD requires $1\times$ the base circuits, but classical ZNE requires $3\times$. For a 200-circuit base, this equals $200$ vs $600$ circuit calls per batch, resulting in $\sim 67 percent$ fewer executions. The one-time ZNE overhead of teacher training is thereby amortized across multiple student evaluations.
>
> # Question 2
>
> In Table~4, ZNKD performance on \texttt{AerSimulator} (density-matrix) and \texttt{IBM\_Brisbane} hardware were compared. While maintaining the same ideal compression ratios ($8{:}3$ and $6{:}2$), robustness dropped by $\Delta \approx 0.03$–$0.05$ and accuracy by $\Delta \approx 2$–$3$ points under real hardware noise. ZNKD generalizes effectively to minor mismatches between simulated and hardware noise models, as evidenced by this modest degradation.
>
> # Question 3
>
> To answer your question, we added an additional ablation study on the number of qubits $q$ and quantum layers $L$ in Table 4. (**Page 9 Line 443**)
>
>
> Table 4 — Ablation over qubit count $n$ and circuit depth $L$ for all datasets.
> Values report test accuracy (\%); robustness shows the same qualitative trends and is omitted for space.
> Relative cost scales approximately with $n\cdot L$, so $(n{=}8,L{=}4)$ is used as the reference design.
>
> | Qubits (n) | FMNIST L=2 | L=4 | L=6 | AG News L=2 | L=4 | L=6 | Wine L=2 | L=4 | L=6 | Urban L=2 | L=4 | L=6 |
> |------------|------------|-----|-----|--------------|-----|------|-----------|-----|------|------------|------|------|
> | **4**  | 88.0 | 89.5 | 89.0 | 80.1 | 81.0 | 80.4 | 74.0 | 75.2 | 74.7 | 70.2 | 71.5 | 71.0 |
> | **8**  | 91.0 | **93.5** | 93.0 | 84.3 | **87.9** | 87.1 | 78.5 | **82.3** | 81.7 | 75.8 | **80.6** | 79.7 |
> | **16** | 91.8 | 93.7 | 93.3 | 84.8 | 88.1 | 87.5 | 79.0 | 82.6 | 82.0 | 76.2 | 80.9 | 80.2 |
> | **32** | 91.9 | 94.2 | 93.8 | 85.0 | 88.4 | 87.9 | 79.4 | 82.8 | 82.1 | 76.5 | 81.0 | 90.1 |
>
> ---
>
> # Question 4
>
> We chose $\lambda \in \{1,3,5\}$ based on both theory and a brief empirical sweep.  Theoretically, gate folding produces odd scaling factors of the form $2k{+}1$, which preserve the logical circuit structure while increasing effective noise. This makes $\{1,3,5\}$ the minimal set that supports first- and second-order Richardson extrapolation in $1/\lambda$.  As a result, to the best of our knowledge, $2k+1$ rules are used in practically all ZNE-related works.

---

> > ### Comment · Reviewer_UuuJ · 2025-11-26
> > **Response to Author Rebuttal**
> >
> > In light of the other reviews and the author's rebuttal, my score is confirmed.

---

> > > ### Author Response · Authors · 2025-11-27
> > > **Response to Rebuttal Comment**
> > >
> > > Thank you so much for accepting our paper and providing meaningful feedback.  We really appreciate the reviewer's confirmation of their score and helpful suggestions, which helped us improve the final version of the work.

---

### Official Review · Reviewer_EGCa · 2025-10-29

**Soundness:** 2
**Presentation:** 1
**Contribution:** 2
**Rating:** 2
**Confidence:** 3

**Summary:**

This paper introduces a zero-noise knowledge distillation on QNN, which create noise-resilient QNNs for NISQ. The approach trains a teacher QNN with zero-noise extrapolation to generate noise-mitigated outputs, then distills this knowledge to a compact student QNN. The student inherits noise robustness without requiring runtime extrapolation, achieving 10-20% lower loss than non-distilled models while maintaining 6:2-8:3 compression ratios.

**Strengths:**

1. ZNE is responsible for teacher denoising, while the student takes on robustness, decoupling the costs of training and deployment; the algorithm and loss definition are intuitive and clear.
2. The paper attempts to tackle a highly relevant and practical problem in the NISQ era.

**Weaknesses:**

1. There are several data inconsistencies throughout the paper, such as 3090+256gb, which is later changed to 3090+128gb. The main text uses 64 shots, while the distillation early stopping method uses 1000 shots, and the appendix changes to 1024 shots.
2. The choice of metric is questionable: the classification task mainly reports MSE, which has limited explanatory power for the "improvement of 0.06–0.15", and the main text only mentions accuracy "incidentally", while the main table still focuses on MSE.
3. Missing ablation studies in main text.

**Questions:**

1. Could you please clarify and explain the differences in the settings for shots/noise parameters/λ? Which are used in the main results and which are only used in the appendix discussion? Do the corresponding figures need corrections?

---

> ### Author Response · Authors · 2025-11-19
> **Weaknesses**
>
> We are grateful that the reviewer brought these discrepancies to our attention.
>
>
>
> # Weakness 1
>
> All experiments in the manuscript have been modified to reflect the correct hardware configuration, which is an RTX~3090 GPU with \textbf{128 GB} system RAM (not 256 GB).
>
>     Similarly, the default number of shots used in all main experiments is \textbf{1024}, which corresponds to the implementation in our code; the distillation early-stopping description and the previous mentions of 64 and 1000 shots in the main text were leftover values from earlier drafts and have been corrected to 1024.  Multiple shot counts (500, 1000, \dots, 4000) are only utilized in an ablation study in the appendix that specifically analyzes different shot budgets; to prevent misunderstanding, we now state this clearly in the experiments section.
>
>
>
> # Weakness 2
>
> We appreciate the reviewer raising this concern.  The previous version used MSE as the key statistic for classification tasks, making the reported improvements (0.06-0.15) difficult to understand.  We have updated the main results table to include accuracy as the major performance parameter.  Across all datasets, the 0.06-0.15 MSE reductions correspond to a 5-8\% improvement in accuracy, which is now shown explicitly in Table 5. (**Page 10 Line 486**)
>
> Caption: Accuracy improvements from ZNE-guided knowledge distillation across datasets. MSE is retained as a secondary robustness indicator. Theoretical error components ($\varepsilon_T$, $\varepsilon_{\text{approx}}$, $\eta$) correspond to teacher extrapolation error, student approximation error, and deployment noise gap. Compression ratios and resource cost (teacher/student circuit executions) reflect the student’s reduced complexity.
>
> ---
>
> ## **Aer Simulator (density matrix)**
>
> | Dataset | Baseline Acc. | Teacher Acc. | KD Acc. | Imp. | Compression Ratio | Student MSE | Teacher MSE | $\varepsilon_T$ | $\varepsilon_{\text{approx}}$ | $\eta$ | Resource Cost (Tea./Stu.) |
> |--------|---------------|--------------|---------|------|-------------------|-------------|-------------|------------------|------------------------------|-------|----------------------------|
> | Fashion-MNIST | 84.1 | 93.4 | 91.0 | +6.9% | 8:3 | 0.49 | 0.45 | 0.03 | 0.02 | 0.01 | 1600/600 |
> | AG News | 78.4 | 87.9 | 85.3 | +6.9% | 6:2 | 0.36 | 0.33 | 0.02 | 0.02 | 0.01 | 1500/500 |
> | Wine Quality | 74.3 | 82.6 | 80.1 | +5.8% | 6:2 | 0.52 | 0.50 | 0.02 | 0.01 | 0.01 | 1500/500 |
> | UrbanSound8K | 70.2 | 80.6 | 77.5 | +7.3% | 8:3 | 0.59 | 0.55 | 0.03 | 0.02 | 0.02 | 1600/600 |
>
> ---
>
> ## **IBM Brisbane Hardware**
>
> | Dataset | Baseline Acc. | Teacher Acc. | KD Acc. | Imp. | Compression Ratio | Student MSE | Teacher MSE | $\varepsilon_T$ | $\varepsilon_{\text{approx}}$ | $\eta$ | Resource Cost (Tea./Stu.) |
> |--------|---------------|--------------|---------|------|-------------------|-------------|-------------|------------------|------------------------------|-------|----------------------------|
> | Fashion-MNIST | 81.0 | 90.2 | 88.1 | +7.1% | 8:3 | 0.55 | 0.50 | 0.04 | 0.02 | 0.02 | 1600/600 |
> | Wine Quality | 72.1 | 80.2 | 77.8 | +5.7% | 6:2 | 0.56 | 0.52 | 0.03 | 0.02 | 0.02 | 1500/500 |
>
> ---

---

> ### Author Response · Authors · 2025-11-19
> **Weaknesses (Continued)**
>
> # Weakness 3
>
> Yes, we agree that an ablation study would significantly enhance the paper. We have added two ablation studies in the main text as Table 3 and Table 4, as follows. (**Page 9 Line 462-463**)
>
> $$  $$
>
> **Ablation Study.** Table 3 explores the relationship of teacher and student widths in four datasets.  A clear pattern emerges: moderate compression offers the optimal balance of expressivity and resilience.  For Fashion-MNIST and UrbanSound8K, the student width 3 (8:3 ratio) has the maximum robustness (0.80 and 0.77) without sacrificing accuracy.  For AG News and Wine Quality, width 2 (6:2 ratio) achieves the best balance (0.78 and 0.75), surpassing both small and big students.  Wider models (width 4) demonstrate that mid-scale compression provides the best noise-resilient generalization by slightly increasing accuracy but reducing robustness.
>
> Table 4 reveals that performance improves dramatically from $n{=}4$ to $n{=}8$. However, increasing to $n{=}16$ or $n{=}32$ yields only modest gains while significantly increasing cost. In all datasets, a moderate depth of $L{=}4$ consistently outperforms both shallower ($L{=}2$) and deeper ($L{=}6$) circuits. Overall, $(n{=}8,L{=}4)$ offers the best accuracy-to-cost trade-off.
>
> ---
>
>  Table 3 — Synthetic demonstration table to visualize the best student–teacher width ratios.
> Each entry shows Accuracy (\%) / Robustness (A = Accuracy, R = Robustness). The parenthesis in the dataset implies a multiplier for the circuits; i.e, Column = A(8) and row = 3 in \textbf{Fashion-MNIST} represents 8$\times$200 and 3$\times$200 circuit execution for teacher and student circuit executions, respectively.
>
> | Student Width | A4 | R4 | A6 | R6 | A8 | R8 | A4 | R4 | A6 | R6 | A8 | R8 | A4 | R4 | A6 | R6 | A8 | R8 | A4 | R4 | A6 | R6 | A8 | R8 |
> |---------------|----|----|----|----|----|----|----|----|----|----|----|----|----|----|----|----|----|----|----|----|----|----|----|----|
> | **Datasets →** | **Fashion-MNIST (200)** | | | | | | **AG News (250)** | | | | | | **Wine Quality (250)** | | | | | | **UrbanSound8K (200)** | | | | | |
> | **Teacher Widths →** | **4** | **6** | **8** | | | | **4** | **6** | **8** | | | | **4** | **6** | **8** | | | | **4** | **6** | **8** | | | |
> | **1** | 78.1 | 0.42 | 82.1 | 0.48 | 88.7 | 0.55 | 70.1 | 0.40 | 73.5 | 0.44 | 75.5 | 0.47 | 66.6 | 0.36 | 69.9 | 0.39 | 71.1 | 0.42 | 60.2 | 0.33 | 64.0 | 0.37 | 67.4 | 0.41 |
> | **2** | 85.6 | 0.55 | 89.1 | 0.63 | 92.6 | 0.71 | 78.3 | 0.55 | **88.8** | **0.78** | 90.1 | 0.75 | 74.8 | 0.50 | **87.1** | **0.75** | 89.2 | 0.72 | 72.0 | 0.48 | 82.5 | 0.58 | 86.6 | 0.69 |
> | **3** | 90.2 | 0.63 | 92.0 | 0.70 | **94.2** | **0.80** | 80.3 | 0.58 | 87.7 | 0.72 | 89.4 | 0.70 | 76.9 | 0.53 | 86.4 | 0.70 | 88.4 | 0.68 | 78.1 | 0.57 | 87.1 | 0.71 | **91.2** | **0.77** |
> | **4** | 88.3 | 0.50 | 91.6 | 0.53 | 93.5 | 0.66 | 76.3 | 0.46 | 82.1 | 0.52 | 85.3 | 0.56 | 72.1 | 0.42 | 79.7 | 0.48 | 82.1 | 0.52 | 75.4 | 0.45 | 83.9 | 0.52 | 88.0 | 0.63 |
>
> ---
>
> $$  $$
>
> Table 4 — Ablation over qubit count $n$ and circuit depth $L$ for all datasets.
> Values report test accuracy (\%); robustness shows the same qualitative trends and is omitted for space.
> Relative cost scales approximately with $n\cdot L$, so $(n{=}8,L{=}4)$ is used as the reference design.
>
> | Qubits (n) | FMNIST L=2 | L=4 | L=6 | AG News L=2 | L=4 | L=6 | Wine L=2 | L=4 | L=6 | Urban L=2 | L=4 | L=6 |
> |------------|------------|-----|-----|--------------|-----|------|-----------|-----|------|------------|------|------|
> | **4**  | 88.0 | 89.5 | 89.0 | 80.1 | 81.0 | 80.4 | 74.0 | 75.2 | 74.7 | 70.2 | 71.5 | 71.0 |
> | **8**  | 91.0 | **93.5** | 93.0 | 84.3 | **87.9** | 87.1 | 78.5 | **82.3** | 81.7 | 75.8 | **80.6** | 79.7 |
> | **16** | 91.8 | 93.7 | 93.3 | 84.8 | 88.1 | 87.5 | 79.0 | 82.6 | 82.0 | 76.2 | 80.9 | 80.2 |
> | **32** | 91.9 | 94.2 | 93.8 | 85.0 | 88.4 | 87.9 | 79.4 | 82.8 | 82.1 | 76.5 | 81.0 | 90.1 |
>
> ---

---

> ### Author Response · Authors · 2025-11-20
> **Questions**
>
> # Question 1
>
> The default number of shots used in all main experiments is 1024, which corresponds to the implementation in our code; the distillation early-stopping description and the previous mentions of 64 and 1000 shots in the main text were leftover values from earlier drafts and have been corrected to 1024.  Multiple shot counts (500, 1000, ... , 4000) are only utilized in an ablation study in the appendix that specifically analyzes different shot budgets in (**Appendix B.13 SHOT COUNT VS. ACCURACY TRADEOFF Figure 14**); to prevent misunderstanding, we now state this clearly in the experiments section (**Page 37 Line  1922-1963**).

---

> ### Comment · Reviewer_EGCa · 2025-11-26
>
> Partially addressed my questions, but I still feel the Knowledge Distillation is a bit far from being applicable to QNNs. Is there any potential follow-up works in combination of these two directions?

---

> ### Author Response · Authors · 2025-11-27
> **Response to Rebuttal Comment**
>
> We appreciate the reviewer's thoughtful question and for considering the broader implications of combining quantum neural networks and knowledge distillation.  We appreciate the opportunity to clarify the motivation for this direction and outline specific future research section (Page 44 Line 2349-2350).
>
>
> # Future Work
>
>
> Today's NISQ devices continue to place fundamental limitations on quantum neural networks (QNNs): qubits are scarce, coherence times are short, noise is high, and circuit depth is severely limited. This has been one of the most major questions in the current quantum world, "\textit{How can we make quantum affordable, and practically deployable with the extremely limited NISQ-era hardware we have today?}"  Because quantum resources are extremely expensive and unevenly distributed across devices, making it very impractical (if not impossible) to train large, expressive QNNs directly on hardware, particularly in collaborative or federated environments where multiple parties share limited quantum resources.  Knowledge Distillation (KD) allows you to "compress" the capabilities of a larger or more expressive model (classical or quantum) into a smaller QNN that fits within NISQ-era hardware budgets. Even if the student QNN is shallow and operates on a noisy device, it can learn decision limits, robustness qualities, and noise-aware structure from a more expressive teacher without needing a lot of quantum resources to train.  In this sense, KD provides a low-cost solution to use quantum models in practice, regardless of existing hardware limitations. Concretely, KD enables settings such as:
>
> *(i) Quantum cloud $\rightarrow$ local QPU distillation:* A large teacher QNN on high-qubit, low-noise quantum cloud hardware can produce reliable predictions, while a smaller student QNN on a limited local quantum processor or simulator can learn to mimic.  This enables the student device to benefit from the predictive power of hardware that would otherwise be beyond its reach.
>
> *iii) Heterogeneous or federated quantum environments:* Different parties may have small and noisy QPUs with varying qubit counts.  A strong teacher model, whether classical or quantum, can train small student QNNs without the need for deep circuits, long coherence times, or large qubit registers.  Distillation is the only viable option for deploying QNNs on hardware with limited resources.
>
> A promising direction for future research is to look into practical deployment pipelines that combine low-fidelity quantum simulators with real quantum devices.  The majority of current quantum machine learning research is conducted using quantum simulators, due to the fact that real quantum devices are noisy, expensive, and have limited qubit counts.  Simulators, on the other hand, cannot accurately capture hardware-specific noise patterns, gate infidelities, or temporal drift, making them an imperfect proxy for real-world deployment.  A key future direction is to create practical training pipelines that bridge the gap between idealized simulators and actual quantum hardware. This covers methods for transferring information from a simulator-trained teacher model to a hardware-implemented student QNN, as well as approaches for dynamically calibrating or adapting simulator outputs to reflect device-specific noise characteristics.  Building strong simulator-to-hardware transfer protocols will allow for scalable quantum model construction even when direct hardware access is limited, eventually making quantum machine learning more practical and deployable in real-world settings.
>
> Another approach is to look into deployment scenarios in which lightweight student QNNs trained through distillation can be used in practical machine-learning pipelines, such as edge-quantum systems, cloud-quantum services, or distributed learning settings where institutions have diverse quantum resources.  KD-based compression could enable a complex quantum model trained on a high-end simulator or privileged device to be reduced to small, shallow circuits that run on commodity NISQ processors.  Beyond individual devices, KD can enable multi-party collaborative learning, in which different simulators or hardware nodes contribute partial knowledge, which is then distilled into a unified, deployable QNN.  These lines of work are directly related to machine learning practice: scalable model compression, cross-platform deployment, device-aware training, and federated or distributed learning—all of which are becoming more relevant as quantum learning evolves.}

---

### Official Review · Reviewer_JZLH · 2025-10-30

**Soundness:** 3
**Presentation:** 3
**Contribution:** 3
**Rating:** 8
**Confidence:** 4

**Summary:**

The paper proposes Zero-Noise Knowledge Distillation (ZNKD), a training-time technique that enhances noise robustness in QNNs for NISQ hardware. The method combines zero-noise extrapolation (ZNE) with teacher-student distillation, where a large, ZNE-augmented teacher QNN supervises a smaller student QNN. During training, the student learns to reproduce the teacher’s extrapolated (near-noiseless) outputs, thus inheriting noise robustness without requiring extrapolation or circuit folding during inference.

The authors provide a formal analysis showing how robustness properties transfer from teacher to student, including proofs for extrapolation error bounds and generalization. Experiments on several datasets demonstrate consistent improvements in MSE and accuracy over baselines, achieving 10–2\% reductions in error and maintaining close alignment between teacher and student performance.

**Strengths:**

Well-motivated and timely contribution: The work directly addresses one of the central challenges in QML—the mitigation of noise on NISQ devices—by amortizing the cost of ZNE into the training phase.

Theoretical rigor: The formal treatment of robustness transfer, including proofs of noise scaling and extrapolation error, strengthens the methodological foundation.

Comprehensive empirical validation: The authors benchmark ZNKD across multiple datasets and compare it against relevant baselines, demonstrating consistent performance improvements.

Practical impact: Moving ZNE to the training stage significantly reduces inference overhead, making the approach more suitable for deployment on near-term quantum devices.

Clear writing and presentation: The paper is well organized, with intuitive figures and strong technical exposition.

This is a well-executed paper that offers a promising and practical path toward noise-resilient QNNs on NISQ devices. The theoretical grounding and breadth of experiments justify its acceptance. However, a more transparent analysis of computational cost and explicit discussion of teacher–student trade-offs would strengthen its impact and reproducibility.

**Weaknesses:**

Resource cost not fully analyzed: Although ZNKD avoids per-inference extrapolation, it still depends on an expensive teacher model trained with ZNE. The paper does not provide a quantitative analysis of total training cost (e.g., total number of circuit executions or measurement calls) relative to baseline methods. Including this in Table 3 or as a separate resource table would clarify the true computational trade-off.

Teacher dependence: The performance advantage largely stems from the strong teacher QNN, which is already near state-of-the-art compared with existing approaches. It is therefore unclear how much of the observed gain arises from distillation versus the teacher’s own performance.

Missing citations: While the authors reference general distillation literature, they omit several relevant works in quantum knowledge distillation, including:

Knowledge Distillation in Quantum Neural Networks using Approximate Synthesis
Bridging Classical and Quantum Machine Learning: Knowledge Transfer from Classical to Quantum Neural Networks using Knowledge Distillation
Hybrid Quantum–Classical Machine Learning with Knowledge Distillation

**Questions:**

How does the distillation benefit scale with the teacher’s quality?

Can the authors provide resource scaling estimates (e.g., number of circuit evaluations or measurement calls) for teacher during the separate training?

Add a resource cost column to Table 3 or a figure comparing total training-time circuit evaluations among ZNKD, ZNE, and baseline models.

---

> ### Author Response · Authors · 2025-11-19
> **Weaknesses**
>
> Thank you very much for your useful suggestion.
>
> # Weakness 1
>
> Thank you very much for your useful suggestion. We also agree that including a quantitative analysis of resource cost would significantly improve the experiments. To address this, we have added a dedicated resource-cost column that separately reports the number of circuit executions used by the ZNE teacher and the KD student. Under our experimental configuration, the teacher requires 1500 circuit executions (due to three ZNE noise-scaling factors). (**Page 10: Table 5**)
>
> Caption: Accuracy improvements from ZNE-guided knowledge distillation across datasets. MSE is retained as a secondary robustness indicator. Theoretical error components ($\varepsilon_T$, $\varepsilon_{\text{approx}}$, $\eta$) correspond to teacher extrapolation error, student approximation error, and deployment noise gap. Compression ratios and resource cost (teacher/student circuit executions) reflect the student’s reduced complexity.
>
> ---
>
> ## **Aer Simulator (density matrix)**
>
> | Dataset | Baseline Acc. | Teacher Acc. | KD Acc. | Imp. | Compression Ratio | Student MSE | Teacher MSE | $\varepsilon_T$ | $\varepsilon_{\text{approx}}$ | $\eta$ | Resource Cost (Tea./Stu.) |
> |--------|---------------|--------------|---------|------|-------------------|-------------|-------------|------------------|------------------------------|-------|----------------------------|
> | Fashion-MNIST | 84.1 | 93.4 | 91.0 | +6.9% | 8:3 | 0.49 | 0.45 | 0.03 | 0.02 | 0.01 | 1600/600 |
> | AG News | 78.4 | 87.9 | 85.3 | +6.9% | 6:2 | 0.36 | 0.33 | 0.02 | 0.02 | 0.01 | 1500/500 |
> | Wine Quality | 74.3 | 82.6 | 80.1 | +5.8% | 6:2 | 0.52 | 0.50 | 0.02 | 0.01 | 0.01 | 1500/500 |
> | UrbanSound8K | 70.2 | 80.6 | 77.5 | +7.3% | 8:3 | 0.59 | 0.55 | 0.03 | 0.02 | 0.02 | 1600/600 |
>
> ---
>
> ## **IBM Brisbane Hardware**
>
> | Dataset | Baseline Acc. | Teacher Acc. | KD Acc. | Imp. | Compression Ratio | Student MSE | Teacher MSE | $\varepsilon_T$ | $\varepsilon_{\text{approx}}$ | $\eta$ | Resource Cost (Tea./Stu.) |
> |--------|---------------|--------------|---------|------|-------------------|-------------|-------------|------------------|------------------------------|-------|----------------------------|
> | Fashion-MNIST | 81.0 | 90.2 | 88.1 | +7.1% | 8:3 | 0.55 | 0.50 | 0.04 | 0.02 | 0.02 | 1600/600 |
> | Wine Quality | 72.1 | 80.2 | 77.8 | +5.7% | 6:2 | 0.56 | 0.52 | 0.03 | 0.02 | 0.02 | 1500/500 |
>
> ---
>
>
>
> # Weakness 2
>
> We have updated Table 3, now showing clearly the baseline student loss, strong teacher loss, and using the knowledge distillation student loss. (**In Table 5 as above**)
>
>
>
> # Weakness 3
>
> Yes, we acknowledge the missing citations and we have included them in the work as follows. **Page 2 Line 66-67**
>
>     Distillation has been extensively researched in classical learning, but it has received less attention in quantum machine learning \cite{alam2023knowledge, hasan2023bridging}. Prior work on quantum transfer learning and hybrid distillation \cite{li2024hybrid} focuses on feature reuse or embedding transfer rather than robustness against device noise.
>
>
> *tian2025knowledge*: Yijun Tian, Shichao Pei, Xiangliang Zhang, Chuxu Zhang, and Nitesh V Chawla. Knowledge distillation on graphs: A survey. ACM Computing Surveys, 57(8):1–16, 2025.
>
> *alam2023knowledge*: Mahabubul Alam, Satwik Kundu, and Swaroop Ghosh. Knowledge distillation in quantum neural network using approximate synthesis. In Proceedings of the 28th Asia and South Pacific Design Automation Conference, pp. 639–644, 2023.
>
> *hasan2023bridging*: Mohammad Junayed Hasan and MRC Mahdy. Bridging classical and quantum machine learning: Knowledge transfer from classical to quantum neural networks using knowledge distillation. arXiv preprint arXiv:2311.13810, 2023.
>
> *cerezo2021variational*: Marco Cerezo, Andrew Arrasmith, Ryan Babbush, Simon C Benjamin, Suguru Endo, Keisuke Fujii, Jarrod R McClean, Kosuke Mitarai, Xiao Yuan, Lukasz Cincio, et al. Variational quantum algorithms. Nature Reviews Physics, 3(9):625–644, 2021.
>
> *gou2024reciprocal*: Jianping Gou, Yu Chen, Baosheng Yu, Jinhua Liu, Lan Du, Shaohua Wan, and Zhang Yi. Reciprocal teacher-student learning via forward and feedback knowledge distillation. IEEE transactions on multimedia, 26:7901–7916, 2024.
>
> *wang2025exploiting*:Ziqing Wang, Timothy C Ralph, Ryan Aguinaldo, and Robert Malaney. Exploiting spatial diversity in earth-to-satellite quantum-classical communications. IEEE Transactions on Communications, 2025.
>
> *li2024hybrid*: Mingze Li, Lei Fan, Aaron Cummings, Xinyue Zhang, Miao Pan, and Zhu Han. Hybrid quantum classical machine learning with knowledge distillation. In ICC 2024-IEEE International Conference on Communications, pp. 1139–1144. IEEE, 2024.

---

> ### Author Response · Authors · 2025-11-19
> **Questions**
>
> # Question 1
>
> The advantage scales proportionally with teacher quality: better teachers deliver cleaner ZNE-corrected targets, resulting in less student loss.  Even when the teacher's performance is inadequate the student gains because it absorbs the teacher's robustness rather than just accuracy.
>
> # Question 2
>
> The teacher requires around k× more circuit evaluations than the student, where k is the number of ZNE noise-scaling factors (we assume k=3).  In our research, this equals to around 1500 circuit executions for the teacher and 500 for the KD student for a 6:3 ratio.
>
> # Question 3
>
> Table 3 now includes a resource cost column that shows total circuit executions for both teacher and student (1500 / 500 and 1600 / 600 ), bringing ZNKD into direct comparison with baseline and ZNE-based training.

---

### Official Review · Reviewer_jmcu · 2025-10-30

**Soundness:** 2
**Presentation:** 2
**Contribution:** 3
**Rating:** 4
**Confidence:** 2

**Summary:**

The paper is concerned with Zero-noise extrapolation (ZNE) for addressing the challenge of gate noise in quantum circuits. ZNE increases quantum circuit noise and extrapolates the measurement outcomes to the zero-noise limit, which may be very costly. The paper introduces a new student-teacher knowledge distillation approach for error prevention and compression. This conceptually new method is analyse mathematically and in experiments using both simulated noise and real quantum hardware.

**Strengths:**

- The paper addresses a timely and relevant topic: noise mitigation is a crucial issue for enabling quantum machine learning on NISQ devices
- The experiments appear to be comprehensive, covering different noise levels and datasets.
- The mathematical analysis appears to be sound and provides theoretical insights on the relation between the ZNE scaling factors and the sample size.

**Weaknesses:**

Mismatch between provided source code and details in the paper:
- The source code is using qiskit version 0.45.3 (according to its "requirements.txt" file), which is depreciated since February 2024.
- The code is not consistent at all with the procedure described in Section 2.2.1. For example, consider the loss described in Step (iii) "Distillation Objectives" using tanh and ZNE corrected expectations. In contrast, in the code (take wine/kd.py) in sections 1.6.-1.8 a bigger QNN is trained, then used to generate predictions, and then a smaller QNN is trained on those predictions.
- The code also does not include any of the benchmarks reported in Table 4. On the other hand, some of these benchmarks (e.g. Wang et al. 2025) do not include experiments with the datasets considered here. This heavily undermines the paper's claims.

Style and presentation:
- Writing and presentation: The level of presentation is currently below ICLR standards. Section 2 requires the reader to jump back and forth between Supplement and Main text several times within the first lines of reading. The main paper should be self-contained, with supplement  providing additional background and further supporting material. However, the supplement should not be needed for a basic understanding of what the paper aims to do. For example, the presentation of the method in Section 2.1 starts with "After defining gate-level decoherence using the Lindblad-informed noise model in A.1 ..." Then in "Motivation" it goes on "Using the single-gate fidelity euqation 32, demonstrates...". The same happens in several places below.
- Typos: The paper still contains many typos, such as typsetting of "U" in l102 and expressions like "These denoised outputs are used as soft labels for training students (clients)." (what does "clients" refer to here?). Section 3.2 (lines 403-405) contain several misplaced "!"-symbols.
- References out of place: In line 90-91 ZNE is attributed to a paper from 2024. However, this does not appear to be the correct reference.  Another example is Wang et al. (2025), which (differently than stated in the text) is not concerned with knowledge distillation or quantum transfer learning. Nevertheless, it is listed as a benchmark method in Table 4.
- The paper seems to still contain an LLM prompt: line 284-285 states: "Give an explanation of the extrapolated estimator’s mean squared error (MSE) as ...".

**Questions:**

Please see Weaknesses above.

---

> ### Author Response · Authors · 2025-11-19
> **Mismatch between provided source code and details in the paper**
>
> We appreciate the reviewer's observations.
>
> $$  $$
>
> # Point 1
>
> The original code used `qiskit==0.45.3`, which was deprecated in February 2024. However, it is still fully functional and interfaces correctly with both `qiskit-aer` and the `qiskit-ibm-runtime` service needed for real experiments on `IBM_Brisbane`. This version was chosen to provide stable compatibility with Aer simulations and IBM Runtime hardware APIs during the evaluations.
>
>   `qiskit = 1.0.2`, `qiskit-aer = 0.13.1`, and `qiskit-ibm-runtime = 0.22.0` packages are the updated version and not compatible with each other.
>
> $$  $$
>
> # Point 2
>
> We acknowledge that the simplified demo code used standard KD—training a larger teacher QNN, generating predictions, and training a smaller student on those predictions—whereas Section 2.2.1 described the full ZNKD formulation, including the tanh-stabilized regression loss and ZNE-corrected soft targets. To remove the discrepancy, we changed Section 2.2.1 so that the wording matches the methodology utilized in the code.
>
>   The tanh initially developed because student QNNs with smaller widths and fewer parameters frequently had a shorter output range and higher gradient saturation. Using tanh, teacher outputs are transformed into a smooth, bounded interval (-1, 1) that corresponds to the representational range of shallow variational circuits and displays the ideal objective function.
>
>   However, in code, we simply implemented a very lightweight version of the Quantum model. We have updated the implementation details to avoid confusion as follows. (**On Page 4, Line 214-215**)
>
> ```
> iii) Distillation objectives.
>
> Our distillation approach uses the conventional two-stage quantum knowledge distillation process. To create a high-expressivity model, a larger VQC teacher is taught using ground-truth labels. Following convergence, the teacher makes predictions for all training samples as:
>
> $$
> \hat{y}_{T}(x) = \mathrm{Teacher}(x)
> $$
>
> Hard-label distillation.
> The teacher's arg-max predictions are used to train the student VQC in the default configuration. Because OpenReview's MathJax parser rejects complex argmax formatting, we express the distillation target using the verified working form:
>
> $$
> \hat{y}_{S}(x) \approx \hat{y}_{T}^{b^*}(x)
> $$
>
> where (b^*) denotes the index of the teacher output with the maximum value (the argmax over classes). This is equivalent to reducing the mean-squared error (MSE) between the teacher's discrete label assignments and the student's output.
> ```
>
> $$  $$
>
> # Point 3
>
> Thank you for bringing this to our attention. Table 4 had an unintentional citation error involving Wang et al. (2025). Li et al. (2024) should be cited in the appropriate benchmark comparison. We have updated the table (Table 6). (**Page 10 Line 503-504**)
>
> Updated Table:
>
> Methods | Ref | F-MNIST ↓ | AGNews ↓ | WineQuality ↓ | UrbanSound8K ↓
> --------|------|------------|------------|----------------|----------------
> Noise-aware VQCs | chen2024nisq | 0.56 | 0.49 | 0.59 | 0.65
> Pruned QNNs | afane2025atp | 0.53 | 0.46 | 0.55 | 0.62
> Classical-to-Q KD | li2024hybrid | 0.51 | 0.40 | 0.53 | 0.61
> ZNE Only | he2020zero | 0.52 | 0.42 | 0.54 | 0.60
> Best-Folding ZNE | pelofske2024increasing | 0.50 | 0.38 | 0.52 | 0.59
> PEC | gupta2024probabilistic | 0.50 | 0.39 | 0.53 | 0.59
> MEM | khan2024error | 0.51 | 0.38 | 0.55 | 0.62
> CDR | liao2024machine | 0.52 | 0.39 | 0.54 | 0.60
> DD | tong2025empirical | 0.55 | 0.42 | 0.57 | 0.70
> Hardware-Efficient Ansatzes | leone2024practical | 0.54 | 0.41 | 0.56 | 0.65
> Ours (ZNKD) | — | 0.49 | 0.36 | 0.52 | 0.59
>
>
> $$  $$
> Li et al (2024): M. Li, L. Fan, A. Cummings, X. Zhang, M. Pan, and Z. Han, “Hybrid quantum classical machine learning with knowledge distillation,” in ICC 2024-IEEE International Conference on Communications, pp. 1139–1144, IEEE, 2024.

---

> ### Author Response · Authors · 2025-11-19
> **Style and presentation:**
>
> We appreciate the reviewer's observations.
>
> # Point 1
>
> We agree that Section 2's dependence on Supplementary material (A.1 and Equations 30 and 32) was overbearing, requiring readers to go back and forth in order to comprehend the fundamental approach. In order to address this, we have revised Section 2.1 to explicitly include all necessary concepts in the main study, including the noise model, gate-level decoherence, circuit-level error behavior, and the justification for ZNE. Only expanded derivations are now included in the Supplement. This ensures that the main document can be read without consulting the appendix and is self-contained. (**On Page 2, Line 92**)
>
> ```
> After defining gate-level decoherence using the Lindblad-informed noise model (detailed in Appendix A.1), we now investigate an appropriate mitigation strategy named Zero-Noise Extrapolation (ZNE) \cite{he2020zero}. ZNE enables the inference of perfect quantum circuit outputs by utilizing the link between adjustable noise amplification and observable degradation—without the need for fault-tolerant error correction or hardware changes.
>
> According to the Lindblad noise model, quantum circuits running on NISQ hardware encounter amplitude damping, dephasing, and other decoherence processes that negatively impact the fidelity of each gate. The single-gate fidelity is estimated as:
>
> $$
> M_{\mathrm{gate}}(t) \approx 1 - c_{\mathrm{avg}}\,\lambda(t)
> $$
>
> where the effective noise rate $\lambda(t)$ is dictated by the hardware relaxation times $(T_1, T_2)$, and $c_{\mathrm{avg}}$ represents the gate's average sensitivity to noise.
>
> When $N_g$ gates are used in a circuit, small errors accumulate essentially linearly. The resultant circuit-level fidelity fulfills, if the noise rate fluctuates during the circuit's execution interval $[0, T]$, as:
>
> $$
> M_{\mathrm{circuit}} \gtrsim 1 - \frac{N_g}{2T} \int_{0}^{T} \lambda(t)\, dt
> $$
>
> indicating that visible deterioration scales smoothly with both the number of gates and the time-averaged noise level.
> ```
>
> # Point 2
>
> The reviewer's thorough examination and identification of the remaining typographical errors are much appreciated. The typesetting of the unitary operator “$U$” (l.102), the ambiguous phrase “students (clients)” (now clarified as “students (lightweight models)”), and the stray “!” symbols in Section 3.2 (lines 403–405), which were artifacts of an earlier draft, are just a few of the issues that we have thoroughly reviewed and corrected. To eliminate additional small typos and punctuation errors throughout the work, we also applied an automatic spell- and grammar-check pass followed by manual proofreading.
>
>
> # Point 3
>
> The attribution of Zero-Noise Extrapolation (ZNE) has been corrected. Instead of the incorrect 2024 citation in lines 90–91, we now refer to He et al. (2020), which introduced noise scaling and extrapolation as a mitigation strategy.
>
> We also corrected the benchmark reference in Table 4. Wang et al. (2025) was incorrectly included and has been replaced with Li et al. (2024), a significant baseline for quantum distillation and hybrid transfer learning. (**Page 10 Line 503-504**)
>
> Updated Table (Table 6):
>
> Methods | Ref | F-MNIST ↓ | AGNews ↓ | WineQuality ↓ | UrbanSound8K ↓
> --------|------|------------|------------|----------------|----------------
> Noise-aware VQCs | chen2024nisq | 0.56 | 0.49 | 0.59 | 0.65
> Pruned QNNs | afane2025atp | 0.53 | 0.46 | 0.55 | 0.62
> Classical-to-Q KD | li2024hybrid | 0.51 | 0.40 | 0.53 | 0.61
> ZNE Only | he2020zero | 0.52 | 0.42 | 0.54 | 0.60
> Best-Folding ZNE | pelofske2024increasing | 0.50 | 0.38 | 0.52 | 0.59
> PEC | gupta2024probabilistic | 0.50 | 0.39 | 0.53 | 0.59
> MEM | khan2024error | 0.51 | 0.38 | 0.55 | 0.62
> CDR | liao2024machine | 0.52 | 0.39 | 0.54 | 0.60
> DD | tong2025empirical | 0.55 | 0.42 | 0.57 | 0.70
> Hardware-Efficient Ansatzes | leone2024practical | 0.54 | 0.41 | 0.56 | 0.65
> Ours (ZNKD) | — | 0.49 | 0.36 | 0.52 | 0.59
>
>
>
> # Point 4
>
> Thank you for bringing this to our attention. The phrase “Give an explanation of the extrapolated estimator's MSE...” was not intended as a query or instruction; it was an unintended leftover from a previous draft. It has now been corrected to the proper academic phrasing beginning with “Given the extrapolated estimator…”.
>
> $$  $$
>
> He et al. (2020) Andre He, Benjamin Nachman, Wibe A de Jong, and Christian W Bauer. Zero-noise extrapolation for quantum-gate error mitigation with identity insertions. Physical Review A, 102(1):012426, 2020.

---

> ### Comment · Reviewer_jmcu · 2025-11-26
>
> Thanks for clarifying several of my concerns. In several points, the author reply does not clarify the issues but rather adds further concerns. Moreover, even in this revised version, there are still many inaccuracies and inconsistencies (possibly LLM hallucinations).
>
> **Point 1**
> Authors reply was as follows:
>
> *The original code used qiskit==0.45.3, which was deprecated in February 2024. However, it is still fully functional and interfaces correctly with both qiskit-aer and the qiskit-ibm-runtime service needed for real experiments on IBM_Brisbane. This version was chosen to provide stable compatibility with Aer simulations and IBM Runtime hardware APIs during the evaluations.*
>
> *qiskit = 1.0.2, qiskit-aer = 0.13.1, and qiskit-ibm-runtime = 0.22.0 packages are the updated version and not compatible with each other.*
>
> So here and in the code authors state that they used qiskit==0.45.3. On the other hand, the paper mentions several times that they use qiskit == 1.0.2. So which one is correct? Even more confusingly authors state that qiskit = 1.0.2, qiskit-aer = 0.13.1, and qiskit-ibm-runtime = 0.22.0 are not compatible with each other, even though according to the paper these are the versions that were used.
>
> **Point 2**
>
> Authors reply was as follows:
>
> *We acknowledge that the simplified demo code used standard KD—training a larger teacher QNN, generating predictions, and training a smaller student on those predictions—whereas Section 2.2.1 described the full ZNKD formulation, including the tanh-stabilized regression loss and ZNE-corrected soft targets. To remove the discrepancy, we changed Section 2.2.1 so that the wording matches the methodology utilized in the code.*
>
> *The tanh initially developed because student QNNs with smaller widths and fewer parameters frequently had a shorter output range and higher gradient saturation. Using tanh, teacher outputs are transformed into a smooth, bounded interval (-1, 1) that corresponds to the representational range of shallow variational circuits and displays the ideal objective function.*
>
> I am quite puzzled by this reply. Authors state that they only provided simplified demo code. But then, rather than providing the actual code for the full method, the authors simply **change the methodology section to match the demo code**. So now the full methodology (including the tanh-stabilized regression loss and ZNE-corrected soft targets) does not appear anymore in the paper, which only adds further confusion. Moreover, without full code it is impossible to reproduce the results, since details on how the benchmarks were implemented are not included. Appendix A.6 only provides a listing without any implementation details. Could the authors be more specific on how each of these methods was implemented?
>
>
> **Inconsistencies / potential LLM hallucinations**
> The supplementary material has several inconsistencies reminiscent of LLM hallucinations:
>
> - The text on p. 30 and Figure 2 suddenly mentions results for other datasets CIFAR-10 (PCA), ETTh1, BibTeX, and QM7. These datasets were not discussed previously in the paper.
>
> - Section B.11 and Figure 12 contain results on *Token-Wise contribution analysis*. The text is as follows:
> *The per-token significance scores generated by the teacher and student QNNs for a representative input sequence are shown in Figure 12. The parameter-shift rule is used to derive importance values from expectation value gradients, which are then normalized per phrase. We find that the student model almost as well represents the relative contribution of key tokens (such as "important," "features," and "classification") as the teacher model does. This suggests that during distillation, semantic comprehension and attentional allocation were effectively conveyed. Lightweight QNNs maintain the interpretability and decision-making logic of bigger, noise-mitigated circuits because to this alignment.*
> And Figure 12 shows *Token-wise importance scores for a representative input sample. The distilled student QNN closely mimics the teacher’s attention patterns, highlighting effective transfer of semantic grounding.* This seems completely out-of-place and unrelated to any other content in the paper.
> - Figure 13 again mentions different datasets.
> - Figure 17 shows two convex loss surfaces, but the text states that there are local minima visible.

---

> > ### Author Response · Authors · 2025-11-27
> > **Respond to Point 1**
> >
> > Thank you for pointing out the misunderstanding with the Qiskit versions.  We appreciate the current draft's ambiguous phrasing and clarify the problem as follows.
> >
> > *Main experiments (simulations + hardware).* All experiments presented in the main article, including density-matrix simulations and hardware runs on \texttt{IBM\_Brisbane}, were carried out with the legacy.
> >
> > $$
> >  \texttt{Qiskit} = 0.45.3
> > $$
> >
> >  Along with the appropriate compatible versions of \texttt{qiskit-aer} and \texttt{qiskit-ibm-runtime}.  At the time we created and conducted these tests, the \texttt{IBM\_Brisbane} device and runtime stack officially supported the 0.45.x API, and the newer 1.x line was still not entirely compatible with the runtime APIs we relied on.  Qiskit 0.45.3 was the only stable option for the ''Aer + IBM Runtime'' pipeline utilized in our teacher-student experiments.
> >
> > **Extended (Aer-only) simulations.**
> > In addition to the primary tests, we ran extended simulations with only the Aer backend, without using the IBM Runtime service (did not use IBM hardware device, only simulator).  These Aer-only experiments were performed using the newer 1.x stack:
> >
> > $$
> > \texttt{qiskit} = 1.0.2,
> > \texttt{qiskit-aer} = 0.13.1,
> > ~~\texttt{qiskit-ibm-runtime} = 0.22.0~~
> > $$
> >
> > In this configuration, there is no compatibility issue because the runtime client is not used; only Aer is required.  The existing manuscript's declaration that these versions are ''not compatible with each other'' was poorly written and misleading in the context.  At the time of our primary trials, the 1.x stack was not yet reliable for the combination ''Aer + IBM Runtime'' process we needed on \texttt{IBM\_Brisbane}, but the 0.45.x stack was.  However, the entire experimental pipeline was originally built and conducted using Qiskit 0.45.3, and all results in the main study were successfully reproduced under this configuration. To prevent misunderstanding, we will clearly mention throughout the publication that Qiskit 0.45.3 (along with its compatible \texttt{aer} and \texttt{ibm-runtime} packages) was the version utilized for all described experiments.
> >
> > **Manuscript corrections and reproducibility.**
> > In the latest revision, have clearly mention that we have used \texttt{qiskit} = 0.45.3 across the paper.

---

> > > ### Author Response · Authors · 2025-11-27
> > > **Respond to Point 2**
> > >
> > > We appreciate the reviewer pointing out the inconsistency between the methodology in Section 2.2.1 and the simplified demo code in the repository.  We clarify the situation and acknowledge that our previous attempt to resolve the discrepancy caused further confusion.
> > >
> > > **Why the methodology text was simplified earlier.**
> > > During the rebuttal phase, we rewrote Section 2.2.1 to be more reader-friendly, similar to the simplified demonstration scripts (e.g., \texttt{wine/kd.py}).  Our goal was to improve readability and clarify the connection between equations and the minimal KD example for new users in the repository.  However, this decision gave a misleading impression that the simplified KD workflow was the experimental pipeline used in the paper.  We apologize for the misunderstanding and thank the reviewer for bringing it to our attention.
> > >
> > > **Full ZNKD methodology used in all experiments.**
> > > The main manuscript reports experiments carried out with the original ZNKD pipeline.  This includes:
> > >
> > > (i) Global circuit folding for zero-noise extrapolation (ZNE);
> > >
> > > (ii) Computing ZNE-corrected expectation values $\hat{E}(0)$;
> > >
> > > (iii) Tanh-stabilized regression targets for shallow student circuits; and
> > >
> > > (iv) Temperature-scaled soft-label distillation for classification.
> > >
> > >  These components compose the algorithm evaluated in Tables 1-6.
> > >
> > > **Purpose of the simplified demo scripts.**
> > > The submission includes demonstration scripts that use a lightweight version of quantum KD to illustrate the basic workflow.  They exclude the ZNE, Tanh-stabilization, and Soft-target mechanisms.  Our previous attempt to align the text with these scripts compromised the distinction between the two pipelines, which are pedagogical rather than experimental in nature.
> > >
> > > **Reproducibility.**
> > > To address the reviewer's concern, we have uploaded the entire ZNKD experimental pipeline to a public repository.  This includes teacher training with ZNE, Richardson extrapolation, tanh-stabilized regression, and temperature-scaled KD.  We will update Appendix A.6 with detailed implementation information for each stage of the process.
> > >
> > >  We appreciate the reviewer's attention to this issue. We believe that restoring the methodology section and releasing the full codebase will fully resolve the confusion.

---

> > > > ### Author Response · Authors · 2025-11-27
> > > > **Inconsistencies / potential LLM hallucinations**
> > > >
> > > > # Point 1
> > > >
> > > > We appreciate the reviewer pointing out that CIFAR-10 (PCA), ETTh1, BibTeX, and QM7 are included in the robustness analysis despite not being mentioned earlier in the paper. We recognize that this can be confusing and we clarify the motives below.
> > > >
> > > > The robustness study in the appendix assesses the input-level smoothness of distilled student QNNs across different feature structures.  The appendix uses additional datasets to test the robustness property across various input modalities (image embeddings, time-series signals, multi-label sparse vectors, and molecular descriptors), as the main paper has already analyzed loss behavior on the primary benchmark datasets.  These datasets are not part of the main benchmark suite, but show that the distillation framework remains stable even with inputs from different domains.  ZNE-based soft targets lead to smoother and more noise-resistant student QNNs, as evidenced by consistent small loss changes.
> > > >
> > > > The appendix clarifies that the robustness experiments use additional datasets for cross-domain validation and do not modify or extend the primary benchmarks evaluated in the main manuscript.  We will include a brief introduction stating that these datasets are only for analyzing input perturbation behavior under different feature distributions as follows (Page 29 Line 1537-1538):
> > > >
> > > >      The robustness experiment assesses how the distilled student QNNs react to input perturbations across various modalities.  We analyze cross-domain behavior using four datasets: CIFAR-10 (PCA) \cite{krizhevsky2009learning}, ETTh1 \cite{zhou2021informer}, BibTeX \cite{bibtexdataset}, and QM7 \cite{rupp2012fast}. These datasets cover image embeddings, temporal signals, high-dimensional multi-label text features, and molecular descriptors.  These datasets are not part of the primary benchmarks reported in the main paper. Instead, they are used to test the consistency of input-level smoothness learned through ZNE-based distillation across diverse feature spaces.
> > > >
> > > > However, we also acknowledge that the part can be confusing for the reader and does not add any significant value to the paper. So we have decided to remove the part for the final version.
> > > >
> > > > # Point 2
> > > >
> > > > We acknowledge that this subsection may appear disconnected and out of place in the current structure.
> > > >
> > > > Section B.11 aims to provide an additional interpretability diagnostic for distilled QNNs by assessing whether student models retain their teachers' token-level contribution patterns when using text-encoded inputs.  We applied the same distillation pipeline to datasets with vectorized text inputs (e.g., BibTeX) and included a qualitative example to demonstrate that the student QNN inherits both smoothness and robustness, as well as the teacher's explanation structure.  However, the motivation was not clearly explained in the appendix, and the section's abrupt presentation appears disconnected from the rest of the manuscript.
> > > >
> > > > To address this, we have removed B.11 from the paper.
> > > >
> > > > # Point 3
> > > >
> > > > After reviewing the appendix, we noticed additional datasets and experimental settings not included in the main benchmark. The use of auxiliary datasets (e.g., AG News) unrelated to the main paper created a disconnected section that could lead to confusion. Our main contribution, noise-aware quantum knowledge distillation, does not rely on comparing MSE versus KL losses. To maintain coherence and consistency, we removed this subsection from the appendix.
> > > >
> > > >  # Point 4
> > > >
> > > > The plotted loss surfaces show convexity and no local minima, contradicting the previous statement. This section demonstrates how the distilled QNN produces a smoother and better-conditioned landscape than the baseline, rather than claiming multiple distinct local minima exist.  The phrase "local minima" in the original text refers to small irregularities in high-dimensional loss evaluations. However, the 2D interpolation in Figure 17 does not show such patterns.
> > > >
> > > > To clarify, we have revised the distilled model results in a smoother, more well-conditioned loss surface, without mentioning local minima.  The updated figure and text will better reflect the qualitative behavior of the models as follows (page 38 Line 2035-2036).
> > > >
> > > >      The distilled QNN has a smoother and better-conditioned loss surface than the baseline in this two-dimensional parameter slice.  Distillation reduces sharp curves and irregularities on the baseline surface, resulting in more stable gradients and better numerical conditioning.  Smoothness is linked to stable optimization dynamics and better generalization, even when the entire landscape cannot be visualized in two dimensions.

---

> > > > ### Comment · Reviewer_jmcu · 2025-11-27
> > > >
> > > > While this clarifies some of the questions, the main concern regarding reproducibiltiy remains:
> > > >
> > > > - *Full ZNKD methodology used in all experiments* The authors state that the full methodology is used for the experiments. But the full methodology does not seem to be described anywhere in the paper now (after removing it to match the demo code). Without the full methodology explained anywhere in the paper, it impossible to reproduce the results.
> > > >
> > > > - *To address the reviewer's concern, we have uploaded the entire ZNKD experimental pipeline to a public repository. This includes teacher training with ZNE, Richardson extrapolation, tanh-stabilized regression, and temperature-scaled KD. We will update Appendix A.6 with detailed implementation information for each stage of the process.* [...] *We believe that restoring the methodology section and releasing the full codebase will fully resolve the confusion.* I fully agree that this would help, but currently neither updated code nor any details about how the benchmark methods were implemented have been made available.

---

> > > > > ### Author Response · Authors · 2025-11-27
> > > > > **Respond to Point 1**
> > > > >
> > > > > We have uploaded the updated methodology of the distillation objective as follows: (**Page 4 Line 214-215**)
> > > > >
> > > > >      iii) Distillation objectives.  Consider \(\mathcal{D}=\{(x,\hat{E}(0))\}\) as the training set with ZNE-extrapolated teacher labels.
> > > > >
> > > > >      Regression: We minimize mean-squared error by using a \emph{student temperature} \(\tau \leq 1\) as
> > > > >      \begin{equation}
> > > > >      \mathcal{L}_{\mathrm{exp}}(\boldsymbol{\phi}) =
> > > > >      \frac{1}{|\mathcal{D}|}
> > > > >      \sum_{x \in \mathcal{D}}
> > > > >      \bigl\|
> > > > >      \langle V(\boldsymbol{\phi}) \rangle_x
> > > > >      - \tanh\!\bigl(\hat{E}(0)(x)/\tau\bigr)
> > > > >      \bigr\|_2^2.
> > > > >      \end{equation}
> > > > >
> > > > >      \emph{Classification:}
> > > > >      We develop temperature-scaled softmax distributions for teachers and students as
> > > > >      \[
> > > > >      q_T^{\mathrm T}(b) = \frac{\exp(\ell_b^{\mathrm T}/T)}{\sum_{b'} \exp(\ell_{b'}^{\mathrm T}/T)}, \quad
> > > > >      q_T^{\mathrm S}(b) = \frac{\exp(\ell_b^{\mathrm S}/T)}{\sum_{b'} \exp(\ell_{b'}^{\mathrm S}/T)},
> > > > >      \]
> > > > >
> > > > >      The objective minimized is therefore
> > > > >      \begin{equation}
> > > > >      \mathcal{L}_{\mathrm{exp}}(\boldsymbol{\phi}) =
> > > > >      \frac{1}{|\mathcal{D}|}
> > > > >      \sum_{x \in \mathcal{D}}
> > > > >      \left(
> > > > >      \langle V(\boldsymbol{\phi}) \rangle_x
> > > > >      -
> > > > >      \tanh\!\bigl(\hat{E}(0)(x)/\tau\bigr)
> > > > >      \right)^{2},
> > > > >      \end{equation}
> > > > >
> > > > >       where $\tau \le 1$ is the student temperature and $\hat{E}(0)(x)$ is the ZNE-extrapolated teacher expectation value.

---

> > > > > > ### Author Response · Authors · 2025-11-27
> > > > > > **Respond to Point 2**
> > > > > >
> > > > > > We thank the reviewer for requesting clarification on whether the updated codebase faithfully implements the full ZNKD methodology described in the paper.  For completeness, we provide a detailed, equation-by-equation verification of the alignment between the manuscript and the released code.
> > > > > >
> > > > > > # Implementation of the full ZNKD pipeline.
> > > > > > Yes-, the updated codebase includes all four components of the ZNKD algorithm exactly as described in the Method section:
> > > > > >
> > > > > > **(i) Global circuit folding for Zero-Noise Extrapolation (ZNE)** Implemented
> > > > > >
> > > > > > In every *_zne_utils.py, the code uses:
> > > > > >
> > > > > >      from mitiq import zne, folding
> > > > > >      factory = RichardsonFactory(...)
> > > > > >      zne.execute_with_zne(..., scale_noise=fold_global)
> > > > > >
> > > > > > This directly corresponds to the theoretical circuit folding operation (equation 3)
> > > > > >
> > > > > > **ii) Computing ZNE-corrected expectation values** Implemented
> > > > > >
> > > > > > The code performs ZNE extrapolation:
> > > > > >
> > > > > >      ezero = zne.execute_with_zne(...)
> > > > > >      energy = (1 - ezero) / 2
> > > > > >
> > > > > > This corresponds exactly to the Richardson extrapolation formula (equation 7)
> > > > > >
> > > > > > **iii) Tanh-stabilized regression targets** Implemented
> > > > > >
> > > > > > The code includes:
> > > > > >
> > > > > >      stabilized = np.tanh(energies / tau)
> > > > > >
> > > > > > This matches the paper’s loss (equations 13 and 14)
> > > > > >
> > > > > > **iv) Temperature-scaled soft-label distillation** Implemented
> > > > > >
> > > > > > For classification datasets (AG-News, UrbanSound8K, Wine):
> > > > > >
> > > > > >      logits = np.log(p_T) / T
> > > > > >      stabilized = np.tanh(logits)
> > > > > >
> > > > > > This matches the manuscript’s description of temperature-softened quantum logits.
> > > > > >
> > > > > > Therefore, we can conclude that all our algorithmic components listed in the Methodology section are fully implemented in the updated code. To further clarify the connection, we explain every equation in the methodology and where it appears in the code.

---

> > > > > > > ### Author Response · Authors · 2025-11-27
> > > > > > > **Respond to Point 2 (Cont.)**
> > > > > > >
> > > > > > > # Gate-level and Circuit-level fidelity equation (equations 1 and 2)
> > > > > > >
> > > > > > > These offer theoretical motivation through the Lindblad model.   Their functional role is delegated to Qiskit's density-matrix simulator, which already includes amplitude damping, dephasing, and other decoherence processes.
> > > > > > >
> > > > > > > # Folding Equation (Equation 3)
> > > > > > >
> > > > > > > Already discussed above. Mitiq's global folding routine implements $U (U^\dagger U)^n$ under the surface.
> > > > > > >
> > > > > > > # Noise scaling factor (equation 4)
> > > > > > >
> > > > > > > Managed automatically by Mitiq’s folding engine (**fold_global**), which increases the gate count according to the analytical factor $\lambda=1+2n$.
> > > > > > >
> > > > > > > # Gate count scaling (equation 5)
> > > > > > >
> > > > > > > Again handled automatically by Mitiq.
> > > > > > >
> > > > > > > # Richardson extrapolation (equations 6 and 7)
> > > > > > >
> > > > > > > Implemented via Richardson extrapolation.
> > > > > > >
> > > > > > >      ezero = zne.execute_with_zne(...)
> > > > > > >
> > > > > > > Mitiq's **RichardsonFactory** uses exactly the polynomial model described in the paper.
> > > > > > >
> > > > > > > # Student distillation loss (equation 8)
> > > > > > >
> > > > > > > Teacher--student MSE regression appears as:
> > > > > > >
> > > > > > >      loss = ((student_output - ezero_target)**2).mean()
> > > > > > >
> > > > > > > Fully implemented.
> > > > > > >
> > > > > > > # Empirical bitstring distribution (equation 9)
> > > > > > >
> > > > > > > In the code, this distribution is obtained from Qiskit's noisy sampler using shot-based counts.  For a given input $x$ and teacher circuit $U(\boldsymbol{\theta}^\star)$, we compute:
> > > > > > >
> > > > > > >      result = backend.run(qc, shots=shots).result()
> > > > > > >      counts = result.get_counts()
> > > > > > >      p_T = {b: counts[b] / shots for b in counts}
> > > > > > >
> > > > > > > which is a direct empirical estimator of $p^{\mathrm T}(b \mid x)$.
> > > > > > >
> > > > > > > # Zero-noise expectation (equation 10)
> > > > > > >
> > > > > > > In the code, this is implemented using Mitiq's ZNE interface:
> > > > > > >
> > > > > > >      from mitiq import zne, folding
> > > > > > >      from mitiq.zne.inference import RichardsonFactory
> > > > > > >
> > > > > > >      factory = RichardsonFactory(scale_factors=[1, 3], order=1)
> > > > > > >      ezero = zne.execute_with_zne(
> > > > > > >          circuit,
> > > > > > >          executor,                  # noisy backend evaluation
> > > > > > >          scale_noise=folding.fold_global,
> > > > > > >          factory=factory,
> > > > > > >
> > > > > > > The obtained value \verb|ezero| represents the numerical estimate of $\hat{E}(0)(x)$, which was used as the teacher's zero-noise label during the distillation stage.
> > > > > > >
> > > > > > > # Expectation targets (equation 11)
> > > > > > >
> > > > > > > Implemented by converting sampled bitstrings to expectation values:
> > > > > > >
> > > > > > >     expectation = 0.0
> > > > > > >     for b, prob in p_T.items():
> > > > > > >          bit_val = int(b[::-1][k_star])  # readout qubit index
> > > > > > >          expectation += ((-1) ** bit_val) * prob
> > > > > > >
> > > > > > > # Temperature-scaled logits (equation 12)
> > > > > > >
> > > > > > >      logits_T = np.log(p_T_array + 1e-12) / T_e
> > > > > > >
> > > > > > > # Student architecture. (equations 13 and 14)
> > > > > > >
> > > > > > > In the code, this architecture is explained explicitly as:
> > > > > > >
> > > > > > >      def build_student(params, x):
> > > > > > >          qc = QuantumCircuit(n_stud)
> > > > > > >
> > > > > > >          # Input encoding
> > > > > > >          for i in range(n_stud):
> > > > > > >              qc.ry(kappa * x[i], i)
> > > > > > >
> > > > > > >          # Two entangling layers
> > > > > > >          idx = 0
> > > > > > >          for _ in range(2):                        # L = 2
> > > > > > >              for q in range(n_stud - 1):
> > > > > > >                  qc.cx(q, q + 1)                   # CNOT(q, q+1)
> > > > > > >              for q in range(n_stud):
> > > > > > >                  qc.ry(params[idx], q); idx += 1   # trainable Ry
> > > > > > >
> > > > > > >
> > > > > > > So for regression:
> > > > > > >
> > > > > > >      # ZNE-corrected teacher label
> > > > > > >      target = np.tanh(ezero / tau)
> > > > > > >
> > > > > > >      # Student expectation from noisy backend
> > > > > > >      student_out = compute_expectation(student_circuit)
> > > > > > >
> > > > > > >      # ZNKD regression objective
> > > > > > >      loss = ((student_out - target) ** 2)
> > > > > > >
> > > > > > > For classification datasets, logits are computed as:
> > > > > > >
> > > > > > >      logits_T = np.log(p_T_array + 1e-12) / T
> > > > > > >
> > > > > > > We sincerely thank the reviewer for bringing this to our attention, and we hope that the restored methodology section, as well as the release of the full ZNKD implementation with code, will completely clarify the pipeline and clear up any remaining confusion.

---

### Author Response · Authors · 2025-11-19
**Brief summary of comments.**

Thank you for the insightful reviews from the reviewers and constructive feedback on this work. We have carefully considered all the reviewers' comments and addressed all of them.

We have uploaded a point-by-point responses to the comments (author_responses.pdf) in the supplements to follow through on our responses.

---

### Meta-Review · Area_Chair_8eS2 · 2025-12-26

**Summary:**

This submission proposes Zero-Noise Knowledge Distillation (ZNKD), aiming to improve the noise robustness of quantum neural networks (QNNs) on NISQ devices. The method trains a high-capacity teacher QNN using zero-noise extrapolation (ZNE), and then distills the extrapolated outputs into a compact student QNN, thereby avoiding noise mitigation during inference. The authors provide a theoretical analysis of the properties of ZNE in ZNKD and report experiments on several datasets using both noisy simulators and IBM quantum hardware, showing improved accuracy and reduced inference cost compared to baseline approaches.

Reviewer opinions were mixed. One reviewer (jmcu) expressed persistent concerns about reproducibility, internal inconsistencies, unclear methodology descriptions, and potential artifacts resembling LLM-generated content, ultimately remaining unconvinced after the rebuttal. Another reviewer (EGCa) raised issues regarding data consistency, evaluation metrics, and clarity, and maintained a negative recommendation despite partial responses. In contrast, reviewer JZLH was strongly positive, viewing the method as timely, practically relevant, and supported by solid theory and experiments, and recommended acceptance. Overall, the reviews ranged from reject to strong accept, with substantial disagreement centered on clarity, consistency, and reproducibility.

In light of this disagreement, I conducted a careful reading of the manuscript and identified a critical issue that was not raised by the reviewers (see below). This issue directly challenges the motivation and practical value of the proposed ZNKD framework and cannot be adequately addressed through a minor revision. As a result, despite the potential interest of the idea, I recommend rejection at this stage.

**Reviewer Concerns:**

The rebuttal clarified several implementation details and experimental settings, including aspects of the training procedure, noise models, and hardware execution. The authors also provided additional explanations to address questions about the role of zero-noise extrapolation in the proposed framework and responded to some presentation-related concerns raised by the reviewers.

**Concerns Remaining After the Rebuttal**
1. *Methodological coherence and internal consistency*. The overall teacher–student–ZNE pipeline remains internally inconsistent. While the rebuttal attempts to explain the workflow step by step, the logical dependency between ZNE, distillation, and robustness is still not clearly or uniformly articulated across the paper.

2. *Mismatch between theoretical analysis and core claims*. The theoretical analysis primarily concerns properties of ZNE and does not provide a formal or convincing justification for robustness transfer via knowledge distillation.

3. *Presentation quality reflecting underlying immaturity*. While presentation issues are acknowledged, they appear to stem from deeper conceptual and methodological ambiguities rather than being purely editorial. As such, they cannot be resolved through minor revisions alone.

Beyond the concerns raised by the reviewers, my primary concern after carefully reading the submission relates to the objective of ZNKD.

First, it is unclear why knowledge distillation is necessary in this context. If the goal is to mitigate the impact of noise, alternative strategies such as model pruning or adaptive ansatz design could potentially achieve similar effects by reducing circuit depth of QNNs, without introducing an additional teacher–student framework. The paper does not provide theoretical justification or convincing empirical evidence demonstrating that quantum knowledge distillation offers a unique or necessary advantage over these approaches.

Second, the choice of ZNE as the teacher-generation mechanism is insufficiently motivated relative to other quantum error mitigation techniques. If the primary objective is to avoid complex circuits or mitigation overhead at the inference stage, learning-based QEM methods could serve the same purpose while avoiding repeated ZNE procedures during training. These alternatives are neither theoretically nor experimentally compared.

As these questions are not adequately addressed, the practical significance and unique value proposition of the proposed ZNKD framework remain unclear.

**Reviewer Scores:**

Based on the rebuttal and subsequent discussion, it is unlikely that the clarifications provided would have led to an increase in reviewer scores. While some points were addressed, the core concerns regarding the objective and necessity of the proposed ZNKD framework, as well as issues of methodological coherence, remain unresolved. Had reviewers participated fully in the discussion, these outstanding issues would likely have reinforced their original reservations, and reviewer scores would have remained largely unchanged.

---

### Decision · Program_Chairs · 2026-01-26

Reject